# Triacylglycerol mobilization underpins mitochondrial stress recovery

Zakery N. Baker[1], Yunyun Zhu[2,3], Rachel M. Guerra[1], Andrew J. Smith[1], Aline Arra[4], Lia R. Serrano[2,3], Katherine A. Overmyer[2,3,5], Shankar Mukherji[4], Elizabeth A. Craig[6], Joshua J. Coon [2,3,5,7] & David J. Pagliarini [1,8,9,10] ✉

Mitochondria are central to myriad biochemical processes, and thus even their moderate impairment could have drastic cellular consequences if not rectified. Here, to explore cellular strategies for surmounting mitochondrial stress, we conducted a series of chemical and genetic perturbations to *Saccharomyces cerevisiae* and analysed the cellular responses using deep multiomic mass spectrometry profiling. We discovered that mobilization of lipid droplet triacylglycerol stores was necessary for strains to mount a successful recovery response. In particular, acyl chains from these stores were liberated by triacylglycerol lipases and used to fuel biosynthesis of the quintessential mitochondrial membrane lipid cardiolipin to support new mitochondrial biogenesis. We demonstrate that a comparable recovery pathway exists in mammalian cells, which fail to recover from doxycycline treatment when lacking the ATGL lipase. Collectively, our work reveals a key component of mitochondrial stress recovery and offers a rich resource for further exploration of the broad cellular responses to mitochondrial dysfunction.

Mitochondria are home to diverse metabolic, biosynthetic and signalling processes. Given this, even moderate disruptions to mitochondrial function can lead to pervasive cellular consequences. Mutations in more than 400 genes have been causally linked to primary mitochondrial diseases[1], which have a collective incidence rate of 1:2,000–1:5,000 (ref. 2). Furthermore, several human diseases including Parkinson's disease[3], amyotrophic lateral sclerosis[4], diabetes mellitus[5] and Alzheimer's disease[6] exhibit hallmarks of secondary mitochondrial dysfunction. Determining the mechanisms by which cells adapt to and overcome mitochondrial stress may be critical to treating these diseases therapeutically. Additionally, this knowledge may assist the strategic impairment of mitochondria, such as in efforts to thwart apoptosis evasion by cancer cells—a therapeutic strategy used for more than 50 years[7,8].

The cellular response pathways that restore mitochondrial function under stress conditions remain an active area of investigation. Early examinations of mitochondrial DNA (mtDNA) loss in *Saccharomyces cerevisiae* established a retrograde signalling pathway that alters nuclear gene expression to rescue critical metabolic functions and extend the replicative lifespan[9–11]. Since these seminal studies, work in cultured mammalian cells and *Caenorhabditis elegans* has revealed signalling pathways and gene expression changes that occur during inter- and intracellular mitochondrial unfolded protein responses[12–15]. Beyond changes in gene expression, cells can elicit mitochondrial fragmentation[16,17] or stress-induced mitochondrial hyperfusion and elongation in response to mitochondrial stress, which can confer stress resistance[16,18,19].

[1]Department of Cell Biology and Physiology, Washington University School of Medicine, St. Louis, MO, USA. [2]Department of Chemistry, University of Wisconsin–Madison, Madison, WI, USA. [3]National Center for Quantitative Biology of Complex Systems, Madison, WI, USA. [4]Department of Physics, Washington University, St. Louis, MO, USA. [5]Morgridge Institute for Research, Madison, WI, USA. [6]Department of Biochemistry, University of Wisconsin–Madison, Madison, WI, USA. [7]Department of Biomolecular Chemistry, University of Wisconsin–Madison, Madison, WI, USA. [8]Department of Biochemistry and Molecular Biophysics, Washington University School of Medicine, St. Louis, MO, USA. [9]Department of Genetics, Washington University School of Medicine, St. Louis, MO, USA. [10]Howard Hughes Medical Institute, Washington University School of Medicine, St. Louis, MO, USA. ✉e-mail: pagliarini@wustl.edu

Although these and other studies have greatly advanced our understanding of mitochondrial stress-related responses[20–23], they were often conducted using singular and severe perturbations that may obscure important nuanced responses across a gradient of stressor severity. This can hinder the exploration of certain response pathways, such as stress-induced mitochondrial hyperfusion, that require metabolically active organelles[18]. Additionally, prior work has focused primarily on protein and gene effectors, marginalizing the role of lipids and metabolites, with few notable exceptions[24,25]. Multiomic mass spectrometry profiling is an effective method to address these potential limitations because it can measure proteins, lipids and metabolites across multiple, diverse stress conditions. This technique has been used extensively to link protein function to novel biological phenomenon[26–28].

Here, we present results from a multiomic mass spectrometry screen examining *S. cerevisiae* strains following perturbations to mitochondrial proteostasis of varying severity. We identified a consistent molecular signature unique to strains that recover from mitochondrial stress. Specifically, we observed a repletion of mitochondrial content dependent on the mobilization of triacylglycerol (TAG) stores. This requisite mobilization was independent of fatty acid oxidation but dependent on TAG lipases (Tgl3-5p), which liberated acyl groups for nascent cardiolipin biosynthesis. We further demonstrate that this mechanism for overcoming mitochondrial stress is conserved in mammalian cells and that deletion of the TAG lipase gene *ATGL* sensitizes cultured cells to treatment with doxycycline (DOX). Collectively, these discoveries expand our understanding of a fundamental biological stress response and suggest that modulating lipid mobilization pathways may help cells contend with the moderate mitochondrial dysfunction observed in pathological conditions.

## Results

### Multiomic profiling of mitochondrial stress and recovery

To investigate how *S. cerevisiae* respond to and recover from certain mitochondrial stress, we performed a multiomic mass spectrometry analysis of 14 yeast strains (2 wild-type (WT) and 12 experimental) with distinct genetic and chemical perturbations to mitochondrial proteostasis. Rather than ablate gene function through gene deletion, we introduced previously documented, disruptive point mutations in target genes using CRISPR–Cas9 (see Supplementary Table 1 for detailed strain descriptions)[20,29–37]. This allowed us to include essential genes (*ssc1*, *hsp10* and *mas1*) or genes required for respiratory competency (*mrpl34*, *rrf1* and *mrh4*). Furthermore, we examined constitutive overexpression of genes known to disrupt mitochondrial proteostasis (*cox4* and *mDhfr*) and exposure to pharmacological compounds that inhibit mitochondrial translation (doxycycline (DOX) and chloramphenicol (CAM)). To control for vehicle effects with pharmacologically treated samples, we also included a dimethylsulfoxide (DMSO)-treated WT (WTV) as a vehicle control.

We grew each experimental and control condition in biological triplicate in respiratory media and collected samples at one or two timepoints: a set early (E) timepoint shared by all strains and, for those strains with a growth lag, a late (L) timepoint at which their optical density (OD) matched that of the WT strain's early timepoint (Fig. 1a). Five strains, denoted 'group 1', exhibited a negligible growth lag within 20 min of WT and thus were only collected at the E timepoint (Fig. 1b and Extended Data Fig. 1a,e). Among the others, three had a mild growth lag (group 2; Extended Data Fig. 1b) and three had a more severe growth lag (group 3; Extended Data Fig. 1c), each of which were also collected at the L timepoint (Fig. 1c). Finally, a single strain (group 4; Extended Data Fig. 1d) did not survive the diauxic shift and was thus collected only at the E timepoint. Collectively, these strains offer an opportunity to examine the cellular and molecular responses to a graded range of mitochondrial stress.

We analysed all collected samples using a custom liquid chromatography–mass spectrometry (LC–MS) processing protocol that allows for the extraction of proteins, lipids and polar metabolites from a single sample[38]. This method resulted in the measurement of 4,489 proteins, 2,713 lipid species (308 identified) and 1,578 metabolites (112 identified), with biological replicates matching closely on principal component analysis (Fig. 1d and Extended Data Fig. 1f). Hierarchical clustering separated the 12 experimental yeast strains and conditions collected at the E timepoint into four distinct clades (Fig. 1e). These clades closely matched the four respiratory growth groups (Fig. 1b), suggesting that they may share common deficiency and recovery signatures. Indeed, our group 3 strains, which had severe yet recoverable growth defects, exhibited marked loss of mitochondrial and OXPHOS proteins at the E timepoint that rebounded as the strains recovered, along with increases in mtDNA levels and a rescue of OXPHOS, suggesting an increase in mitochondrial biogenesis (Fig. 1f–h and Extended Data Fig. 1g–j). Interestingly, the group 4 *ssc1* strain displayed a comparable set of mitochondria-related changes at the E timepoint (Extended Data Fig. 1k). In fact, group 3 and group 4 strains possessed a very similar global respiration deficiency response—a universal response to loss of respiratory capacity[26] (Extended Data Fig. 1l). Despite its similar response, the *ssc1* strain did not recover, perhaps indicating a failure to induce restorative pathways. Moreover, expression of defined retrograde stress-response pathway markers that may aid recovery were not induced in any of these strains (Extended Data Fig. 1m,n). This is perhaps due to our selected parental yeast strain (W303), which has been shown to lack a robust retrograde response[11]. Collectively, these data suggest that group 3 strains mount a successful cellular response to mitochondrial stress that may be distinct from the currently appreciated response pathways.

### TAG-derived cardiolipin production coincides with recovery

To further investigate pathways relevant to the group 3 recovery, we examined our multiomic data for molecular changes that distinguish these strains from group 4. The most striking difference between these groups was a marked decrease in group 3 TAG species, which remained largely unaffected in the sicker *ssc1* strain (Fig. 2a). These changes in TAG abundance occurred without notable changes to any other lipid class or an overall downshift in total lipid abundance (Extended Data Fig. 2a–d). Except for slight increases in select species with low levels of saturation, the changes occurred in all TAG species regardless of chain length or saturation level (Extended Data Fig. 2e–g). The TAG phenotype was confirmed using BODIPY staining (Fig. 2b and Extended Data Fig. 2h) and Nile red fluorescence (Fig. 2c and Extended Data Fig. 2i), demonstrating that the decrease was robust and specific to the affected strains.

Diminished TAG levels could result from compromised TAG biosynthesis and/or increased TAG mobilization. To assess the former, we performed a labelling experiment in which strains subjected to chronic DOX treatment were pulsed with heavy [$^{13}C_1$]oleic acid for 1 h before collection to label newly synthesized TAGs. DOX-treated yeast again demonstrated overall depletion of their unlabelled (light) TAGs (Fig. 2d); however, they rapidly produced new heavy-labelled TAGs following the [$^{13}C_1$]oleic acid pulse (Fig. 2e). This observation suggests that the TAG loss in group 3 strains probably occurs through an increase in TAG mobilization, with liberated acyl groups potentially being shuttled into other lipid species.

To monitor potential destinations for TAG-derived acyl chains, we performed a separate labelling experiment in which WT cells were pulsed with [$^{13}C_1$]oleic acid for 1 h followed by an acute 3 h treatment with DOX or vehicle (DMSO) control (Fig. 2f,g). The heavy oleic acid treatment effectively labelled TAGs and phospholipid species (Extended Data Fig. 2j–m), enabling us to observe subsequent DOX-induced changes. Following the 3 h treatment, DOX-treated cells exhibited a significant elevation of labelled cardiolipin over vehicle-treated cells, despite the former's reduced growth rate (Fig. 2f,g). By contrast, no significant increases were observed for heavy phosphatidylethanolamine or heavy

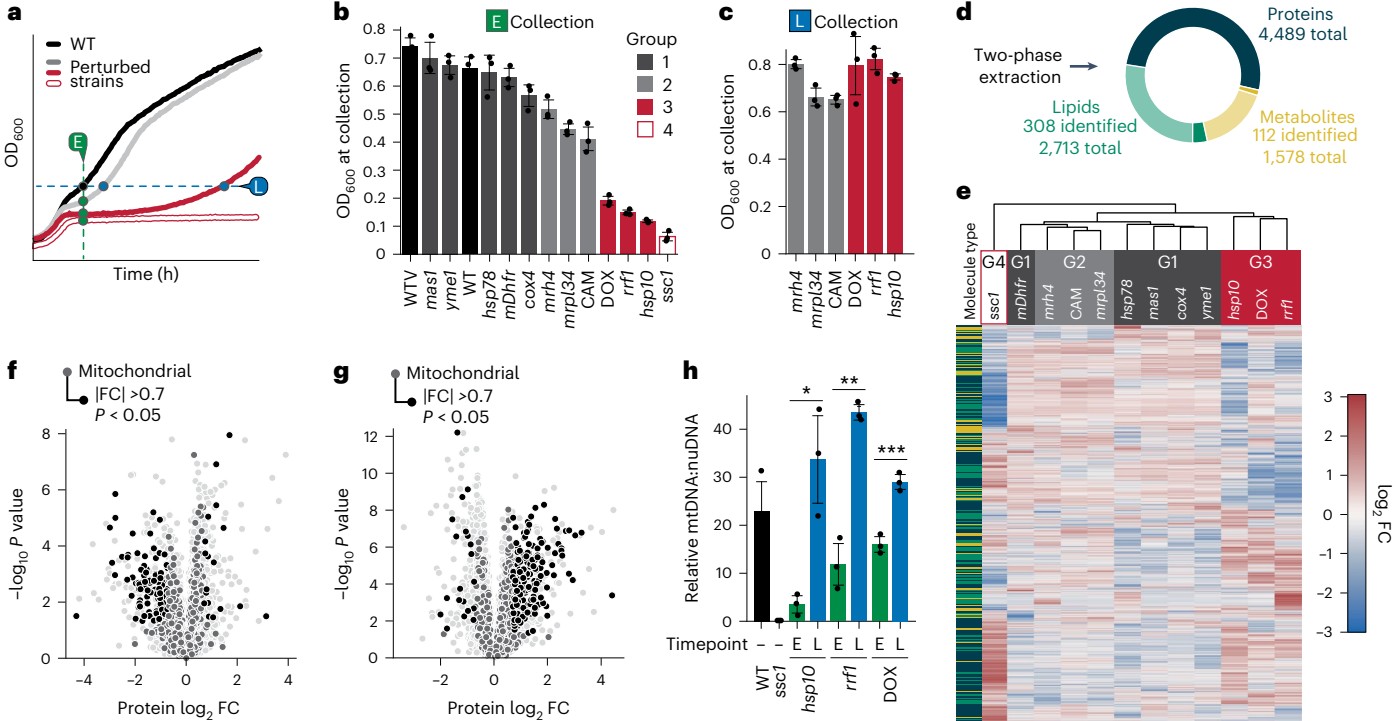

**Fig. 1 | A multiomic mass spectrometry screen to identify requirements for overcoming mitochondrial dysfunction. a**, A schematic of the collection timeline for multiomic mass spectrometry screening. All 14 strains (2 WT and 12 experimental) were inoculated in YPG respiratory media and incubated for 24 h until the first early collection timepoint after the diauxic shift (E, green dots). Strains with an appreciable growth defect were collected a second time (L, blue dots) when each respective strain had reached the optical density (OD) of the WT at the E timepoint. **b,c**, The OD (600 nm wavelength) at collection for each of the 14 strains at the E timepoint ($n = 3$) (**b**) or the six growth deficient strains at the second (L) timepoint ($n = 3$) (**c**). **d**, Breakdown of the >8,500 biomolecules quantified in each strain by class. **e**, Hierarchical clustering of the experimental strains in the screen (group 1 (G1), group 2 (G2), group 3 (G3), group 4 (G4)). Strains were clustered based on the average abundance FC ($n = 3$) for all biomolecules quantified (proteins, dark green; lipids, light green;

metabolites, yellow)). **f**, Relative protein abundances for all group 3 strains ($n = 9$) collected at the E timepoint compared with WT ($n = 3$) versus statistical significance. **g**, Relative protein abundances for all group 3 strains ($n = 9$) collected at the L timepoint compared with all group 3 strains ($n = 9$) collected at the E timepoint versus statistical significance. In **f** and **g**, non-mitochondrial proteins are coloured light grey and mitochondrial proteins are coloured dark grey. Mitochondrial proteins that are significantly changed ($|FC| > 0.7$, $P < 0.05$; two-sided Student's $t$-test) are highlighted black. **h**, The relative abundance of mtDNA in group 3 and *ssc1* strains at both in the E and L timepoints as measured by a ratio of mtDNA to nuclear DNA (nuDNA). Strains were grown in growth conditions identical to the original screen ($n = 3$, *$P = 0.01$, **$P = 6.37 \times 10^{-4}$, ***$P = 1.21 \times 10^{-3}$; two-sided Student's $t$-test). For all experiments, the error bars are the s.d., centre values represent the mean and $n$ is the number of independent biological replicates.

phosphatidylcholine (Fig. 2g). This suggests that, following mitochondrial stress, available acyl groups are preferentially shuttled into nascent cardiolipin biosynthesis, potentially to support mitochondrial biogenesis. Consistently, along with their increased mitochondrial protein abundance at the L timepoint of the multiomic screen (Fig. 1g), group 3 strains exhibited considerable elevation of the four most abundant *S. cerevisiae* cardiolipin species[39] to levels higher even than their healthy WT counterparts (Fig. 2h).

Biosynthesis of the mitochondria-specific cardiolipin occurs entirely within the organelle itself[40]. To determine whether disruption of this biosynthetic pathway would influence efficient recovery from mitochondrial dysfunction, we generated yeast deficient in the cardiolipin synthase Crd1p and tested their growth under DOX treatment. Consistent with previous work[41], deletion of *CRD1* resulted in the total loss of cardiolipin in cells both treated and untreated with DOX (Fig. 2i and Extended Data Fig. 2n). Likewise, consistent with past reports on *CRD1* (ref. 42), we observed that the loss of cardiolipin occurred concurrently with a sharp increase in the abundance of the cardiolipin precursor phosphatidylglycerol (PG) (Extended Data Fig. 2n). Although this strain did display a slight growth defect in glycerol, the increase in PG presumably allows the *crd1Δ* strain to maintain respiratory growth (Fig. 2j). The increase in PG did not, however, enable this strain to efficiently recover from DOX treatment like its WT counterpart

(Fig. 2k). These data further suggest that cardiolipin is essential for recovery from mitochondrial stress and that the increase in cardiolipin we observe in our group 3 yeast strains is an adaptive response that promotes efficient recovery. Interestingly, we found that, in addition to their lack of recovery, *crd1Δ* yeast did not mobilize TAG stores in response to DOX treatment (Fig. 2l).

## Lipid droplets provide TAGs for stress adaptation

These considerable lipid profile changes, coupled with the fact that our multiomic data lack spatial resolution, prompted us to examine broader cellular phenotypes. To do so, we measured organellar changes using confocal microscopy in fluorescently tagged 'rainbow' yeast treated with DOX[43]. Consistent with our multiomic data, the mitochondrial volume fraction of DOX-treated yeast was negatively skewed at the earlier timepoint and partially recovered by the late postrecovery timepoint. (Fig. 3a–c). By contrast, the endoplasmic reticulum exhibited a more normal distribution at all timepoints measured (Fig. 3d,e). Interestingly, the volume of lipid droplets closely matched that of mitochondria (Fig. 3f,g), further indicating a role for lipid mobilization in stress recovery and suggesting that lipid droplets are indeed the source of the exhausted TAG species.

Guided by this observation, we revisited our proteomics data and performed a correlation analysis with all measured proteins, lipids and metabolites to identify proteins correlated with the TAGs most affected

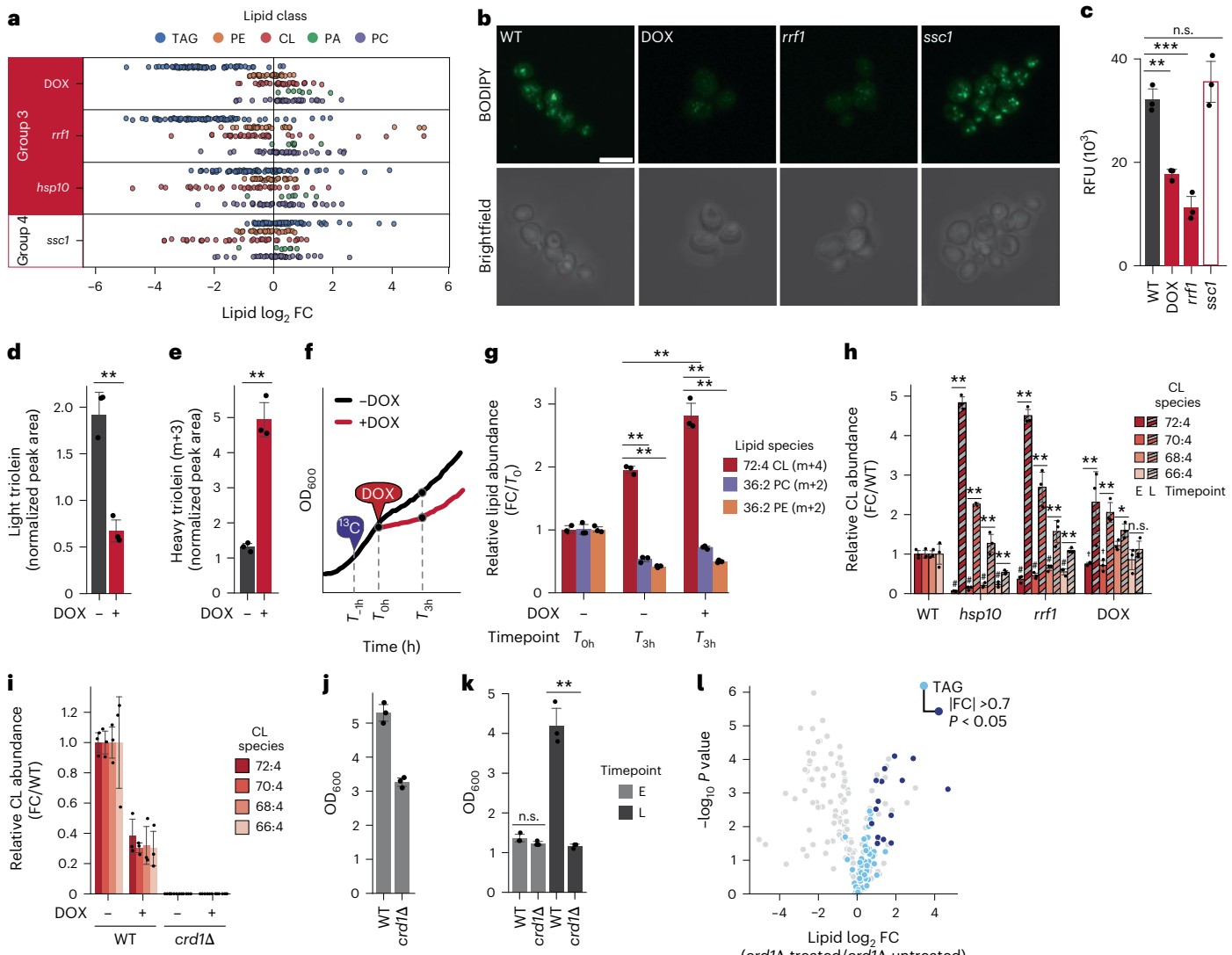

**Fig. 2 | Yeast strains that recover from mitochondrial stress mobilize TAG stores to facilitate cardiolipin biosynthesis. a**, The relative abundances ($\log_2$ FC) of select lipid species for group 3 and *ssc1* strains at the E timepoint compared with WT from the multiomic screen ($n = 3$). CL, cardiolipin; PA, phosphatidic acid; PC, phosphatidylcholine; PE, phosphatidylethanolamine; TAG, triacylglycerol. **b**, Representative confocal microscopy images of BODIPY (green) stained yeast strains grown in respiratory YPG media taken at 100× magnification. Scale bar, 10 μm. **c**, Relative fluorescent signal from Nile red staining of select group 3 and *ssc1* yeast strains grown in respiratory media ($n = 3$, \*\*$P = 8.61 \times 10^{-4}$, \*\*\*$P = 5.85 \times 10^{-4}$; two-sided Student's *t*-test). n.s., not significant. **d**,**e**, The normalized abundance of triolein (**d**) or de novo synthesized [$^{13}$C$_3$]triolein (**e**) in cells treated with 400 μM DOX or DMSO vehicle control ($n = 3$, in **d**, \*\*$P = 1.62 \times 10^{-3}$ and in **e**, \*\*$P = 5.01 \times 10^{-4}$; two-sided Student's *t*-test). Where indicated, cells were treated with 0.002% (v/v) of [$^{13}$C$_1$]oleic acid. **f**, A schematic for the acute DOX labelling experiment. WT cells grown in YPG respiratory media were inoculated and incubated for 23 h and treated for 1 h with [$^{13}$C$_1$]oleic acid, followed by a 3 h acute DOX treatment and collection. **g**, The normalized abundance of de novo synthesized [$^{13}$C$_4$]72:4 CL (red), [$^{13}$C$_2$]36:2 PC (purple) and [$^{13}$C$_2$]36:2 PE (orange) in cells after pulse-chase growth. Abundance is normalized to the average $T_0$ values

($n = 3$). **h**, The normalized abundance of the four most abundant species of CL in group 3 strains at both the E (solid) and, when specified, L (striped) timepoints from the multiomic screen. CL species are denoted by colour. Abundance is normalized to the average WT values ($n = 3$). **i**, The normalized abundance of the four most abundant species of CL in *crd1*Δ yeast compared with WT in the presence or absence of DOX. Cells were grown for 24 h in respiratory YPG media. Abundance is normalized to the average WT values ($n = 3$). **j**,**k**, Growth assay of WT and *crd1*Δ yeast treated with DMSO vehicle control (**j**) or 400 μM DOX (**k**) ($n = 3$, \*\*$P = 5.91 \times 10^{-4}$; two-sided Student's *t*-test). Growth was measured after 24 h (**j** and **k**, timepoint E) or 48 h (**k**, timepoint L) in respiratory YPG media. **l**, The relative lipid abundance versus statistical significance for *crd1*Δ yeast treated with DOX compared with DMSO vehicle control. Cells were grown for 24 h in respiratory YPG media. Quantified TAG species are coloured light blue, with TAG species that are significantly changed (|FC| >0.7, $P < 0.05$; two-sided Student's *t*-test) highlighted dark blue. All other lipid species are coloured grey. For all experiments, the error bars are the s.d., centre values represent the mean and $n$ is the number of independent biological replicates. For samples with more than four comparisons, \*$P < 0.05$, \*\*$P < 0.01$; two-sided Student's *t*-test. For all lipid experiments, abundances were normalized to CoQ$_8$ internal standard.

by mitochondrial stress (Fig. 3h). Of these, the partially characterized lipid droplet protein Pln1p (also known as Pet10p) was most positively correlated with all TAGs measured, despite its modest absolute protein or transcript abundance changes in affected strains (Extended Data Fig. 3a–c). Pln1p has been described as a yeast perilipin and is thought to reside on the surface of nascent lipid droplets and have a role in TAG

biogenesis[44]; however, its role in TAG mobilization is unknown. Given its positive correlation with TAG levels, we reasoned that overexpressing *PLN1* could disrupt TAG catabolism observed under mitochondrial stress. Indeed, overexpression of *PLN1* prevented the reduction of TAGs under DOX treatment and resulted in an overall increase of TAG species (Fig. 3i,j and Extended Data Fig. 3d).

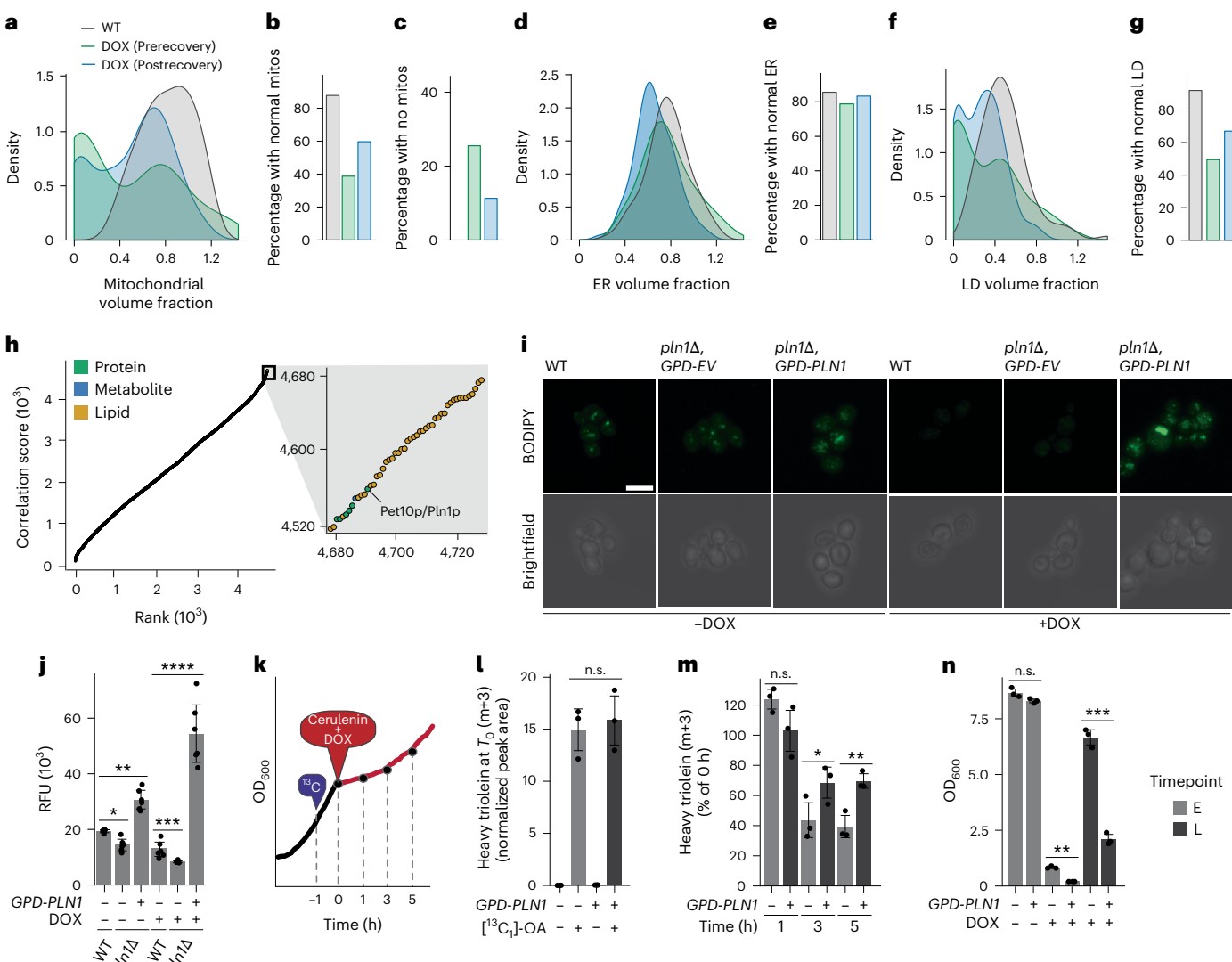

**Fig. 3 | Lipid dropletss are linked to recovery from mitochondrial stress.**
**a**, Fluorescent microscopy-derived cellular volume fractions of mitochondria for cells treated with vehicle or 200 µM DOX. Cells were grown in YPG respiratory media and imaged at 24 h (prerecovery) and again when the OD of the treated strain had reached the OD of the untreated (postrecovery). **b,c**, The percentage of cells from **a** that contained normal volume of mitochondria (mitos) (**b**) or no detectable mitochondria (**c**). **d**, Cellular volume fractions of endoplasmic reticulum (ER). **e**, The percentage of cells from **c** and **d** that contained a normal volume of ER. **f**, Cellular volume fractions of lipid droplets. **g**, The percentage of cells from **c** and **d** that contained a normal volume of lipid droplets (LDs). In **a**, **d** and **f**, density is plotted as the estimated kernel density based on the counts of the measured population ($n$ = 90, 90 and 100 for untreated mitochondria, ER and lipid droplets, respectively; $n$ = 98, 90 and 97, for treated prerecovery mitochondria, ER and lipid droplets, respectively; $n$ = 186, 186 and 185 for postrecovery mitochondria, ER and lipid droplets, respectively). In **b**, **e** and **g**, volume fractions were considered 'normal' if they fell within 1.5 s.d. of the mean volume fraction in WT cells. **h**, Correlation analysis among all individual, quantified biomolecules from multiomic screen and decreased TAG species. Biomolecules were rank ordered according to their average Spearman correlation coefficient between themselves and each individual TAG species. The inset shows the top 50 most positively correlated biomolecules with the decreased TAG species. **i**, Representative confocal microscopy images of BODIPY (green) stained WT or $pln1\Delta$ yeast strains containing either an empty

vector ($GPD$-$EV$) or the $PLN1$ gene ($GPD$-$PLN1$). Cells were grown for 24 h in SDG respiratory media treated with either 400 µM DOX or DMSO vehicle control. Images were taken at 100× magnification. Scale bar, 10 µm. **j**, Relative fluorescent signal from Nile red staining of WT or $pln1\Delta$ yeast strains expressing either empty vector or $GPD$-$PLN1$. Cells were grown for 24 h in SDG respiratory media treated with either 400 µM DOX or DMSO vehicle control ($n$ = 6, $^{*}P$ = 4.66 × 10$^{-4}$, $^{**}P$ = 2.19 × 10$^{-5}$, $^{***}P$ = 1.83 × 10$^{-3}$, $^{****}P$ = 5.92 × 10$^{-6}$; two-sided Student's $t$-test). **k**, A schematic for the TAG mobilization experiment. WT cells, expressing empty vector or $GPD$-$PLN1$, were inoculated and incubated for 23 h in YPG respiratory media then treated for 1 h with [$^{13}$C$_1$]oleic acid, followed by an acute 400 µM DOX and 10 mg l$^{-1}$ cerulenin treatment. Cells were collected at 1, 3 and 5 h after DOX/cerulenin inoculation. **l**, Normalized abundance of de novo incorporated [$^{13}$C$_3$]triolein after pulse treatment at $T_0$ ($n$ = 3). **m**, Normalized abundance of [$^{13}$C$_3$]triolein at 1, 3 and 5 h timepoints post-DOX/cerulenin treatment to track TAG mobilization. The abundance is normalized to the average corresponding $T_0$ samples ($n$ = 3, $^{*}P$ = 0.08, $^{**}P$ = 8.19 × 10$^{-3}$; two-sided Student's $t$-test). **n**, Growth assay of WT yeast strains expressing empty vector or $GPD$-$PLN1$. Cells were treated with 400 µM DOX or DMSO control and growth was measured after 24 h (E timepoint) or 44 h (L timepoint) in YPG respiratory media ($n$ = 3, $^{**}P$ = 3.97 × 10$^{-5}$, $^{***}P$ = 8.85 × 10$^{-5}$; two-sided Student's $t$-test). For all experiments, error bars are the s.d., centre values represent the mean and $n$ is the number of independent biological replicates. For all lipid experiments, abundances are given as peak area normalized to a CoQ$_8$ internal standard.

To directly measure the effect of *PLN1* overexpression on TAG catabolism, we performed a modified [$^{13}$C$_1$]oleic acid pulse-chase experiment (Fig. 3k). As in our previous heavy labelling experiments, we cultured yeast in respiratory media and treated them with [$^{13}$C$_1$]oleic acid for 1 h before collecting and measuring the abundance of newly synthesized heavy [$^{13}$C$_1$]-labelled triolein (Fig. 3k). Overexpression

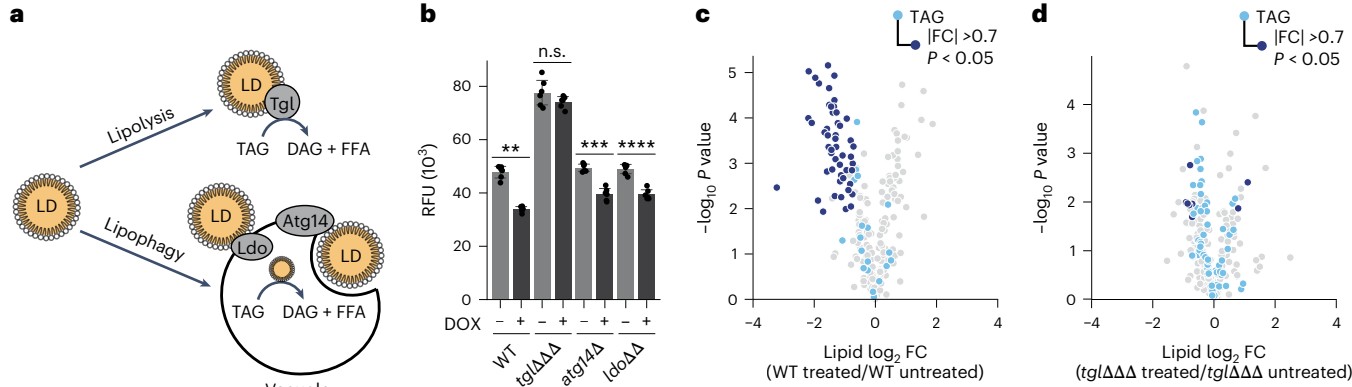

**Fig. 4 | TAG mobilization during mitochondrial stress requires *tgl* lipases.**
**a**, A schematic illustrating the two main pathways for TAG mobilization in yeast. TAGs stored in lipid droplets are accessed through either lipolysis using the Tgl3-5p lipases (top) or lipophagy involving direct consumption of the lipid droplet into the vacuole and digestion of TAGs requiring Ldo45/16 and Atg14 (bottom). DAG, diacylglycerol; FFA, free fatty acid. **b**, The relative fluorescent signal from Nile red staining of WT, *TGL* triple deletion (*tgl*ΔΔΔ), *atg14*Δ, and *ldo*ΔΔ) yeast grown in YPG respiratory media (*n* = 6). Cells were treated with either 400 μM DOX or a DMSO vehicle control. The error bars represent the s.d., centre values

represent the mean (***P* = 1.37 × 10⁻⁷, ****P* = 5.05 × 10⁻⁶, *****P* = 5.67 × 10⁻⁶; two-sided Student's *t*-test). **c,d**, The relative lipid abundance versus statistical significance for WT (**c**) or *tgl*ΔΔΔ (**d**) yeast treated with either 400 μM DOX or a DMSO vehicle control (*n* = 3). Cells were grown for 24 h in YPG respiratory media. In **c** and **d**, quantified TAG species are coloured light blue, with TAG species that are significantly changed (|FC| >0.7, *P* < 0.05; two-sided Student's *t*-test) highlighted dark blue. All other lipid species are coloured grey. For all experiments, *n* is the number of independent biological replicates.

of *PLN1* did not have a significant effect on the abundance of heavy [¹³C₃]-labelled TAGs, suggesting that is does not drastically increase the rates of TAG biosynthesis (Fig. 3l). Following the 1 h labelling with [¹³C₁] oleic acid, we next treated the strains with cerulenin, a potent inhibitor of TAG biosynthesis[45], and DOX to induce TAG mobilization, and collected yeast after 1, 3 and 5 h to measure the rate at which the heavy [¹³C₃]triolein was mobilized (Fig. 3k). By the 3 h timepoint, TAG levels in WT yeast were already trending lower than the *PLN1* overexpressing strain. This difference widened at 5 h, with WT yeast having 43% lower levels of heavy [¹³C₃]triolein compared with the *PLN1* overexpressing strain (Fig. 3m). Fittingly, without the ability to mobilize TAG stores, yeast overexpressing *PLN1* displayed a significant growth defect in respiratory media when treated with DOX (Fig. 3n and Extended Data Fig. 3e,f). Together, these analyses highlight the importance of lipid droplet-derived TAG mobilization for mitochondrial stress recovery that is, in part, regulated by the poorly characterized protein Pln1p.

**Efficient stress recovery requires TAG mobilization**
Given our observation that TAG mobilization is key for mitochondrial recovery, we next explored the mechanism by which acyl chains are derived from lipid droplets. TAG catabolism in yeast occurs via one of two pathways (Fig. 4a). The most well studied is the gradual lipolysis of TAG molecules at the lipid droplet via the TAG lipases Tgl3-5p[46,47]. The second pathway for TAG mobilization is lipophagy, where lipid droplets are engulfed within the vacuole and rapidly degraded[48]. Lipophagy requires the lipid droplet–vacuole contact site Ldo proteins (Ldo45/16p)[49] and canonical autophagy proteins such as Atg14p[50], Atg15p[48] and Atg1p[51]. To determine the mechanism for our observed decrease in intracellular TAG abundance, we generated yeast deletion mutants defective in each of these pathways (*tgl3*Δ*tgl4*Δ*tgl5*Δ (*tgl*ΔΔΔ) for lipolysis, and both *ldo45*Δ/*16*Δ (*ldo*ΔΔ) *and atg14*Δ for lipophagy). We then performed an acute 3 h DOX treatment to observe TAG mobilization. Despite the reduced duration of DOX treatment compared with our initial analysis (Fig. 1a), WT cells displayed reduced TAG levels as measured by either Nile red florescence (Fig. 4b) or LC–MS/MS (Fig. 4c). Of the three deletion strains tested, only the triple *tgl*ΔΔΔ mutant preserved TAG levels after the 3 h DOX treatment, while both the *ldo*ΔΔ and *atg14*Δ strains exhibited TAG depletion comparable to WT cells (Fig. 4b,d and Extended Data Fig. 4a,b). Other canonical autophagy mutants such as *atg1*Δ and *atg15*Δ also had no effect on TAG

mobilization (Extended Data Fig. 4c–e). Together, these data indicate that Tgl3-5p-driven lipolysis is the main pathway for TAG catabolism employed by yeast cells following this mitochondrial stress.

To further explore the role of lipolysis in enabling cellular recovery from mitochondrial stress, we compared the ability of WT and *tgl*ΔΔΔ mutant strains to recover from prolonged DOX treatment. Both strains exhibited a substantial growth defect after 24 h in DOX; however, unlike WT, *tgl*ΔΔΔ mutant yeast were unable to recover growth by the later stage (Fig. 5a). Interestingly, yeast lacking the TAG biosynthetic gene diacylglycerol-acyl transferase (*dga1*Δ), which fail to accumulate TAGs under respiratory conditions, or quadruple knockout strains lacking all neutral lipid synthesis enzymes (*dga1*Δ*lro1*Δ*are1*Δ*are2*Δ (ΔΔΔΔ)), also failed to overcome DOX treatment (Fig. 5a and Extended Data Fig. 5a–c). Deletion of the alternative TAG synthesis enzyme in yeast, *LRO1*, alone did not affect recovery (Extended Data Fig. 2c). This further implicates an essential role for TAG biogenesis and subsequent mobilization in overcoming mitochondrial stress and excludes the possibility that TAG toxicity from failed catabolism is driving the delayed growth phenotype in our *TGL* triple mutant strain.

Our analyses above demonstrate that TAG mobilization is key to cellular recovery from mitochondrial stress, at least in part by supporting new cardiolipin production. A potential second role for TAG lipolysis could be to fuel fatty acid beta-oxidation (FAO). To test this, we also generated an FAO-deficient yeast strain by deleting the fatty-acyl coenzyme A oxidase gene, *POX1. pox1*Δ yeast, which showed increased TAG levels before treatment (Extended Data Fig. 5c), demonstrated robust recovery and TAG mobilization following DOX treatment (Fig. 5a and Extended Data Fig. 5d). These experiments indicate that increased FAO was not supporting recovery and further highlight the importance of new lipid biosynthesis, such as cardiolipin, for mitochondrial membrane biogenesis. Consistent with this hypothesis, unlike WT yeast, *tgl*ΔΔΔ yeast failed to fully increase their cardiolipin levels after prolonged DOX treatment (Fig. 5b) and were unable to rescue mitochondrial protein abundance, despite having WT levels of mtDNA (Fig. 5c–g and Extended Data Fig. 5f). Furthermore, supplementation with oleic acid, at a sufficient level for labelling TAG species, was not able to rescue the growth defect (Extended Data Fig. 5g,h). Together, these data indicate that TAG mobilization, specifically through Tgl3-5p-driven lipolysis, is essential for overcoming DOX-induced mitochondrial stress.

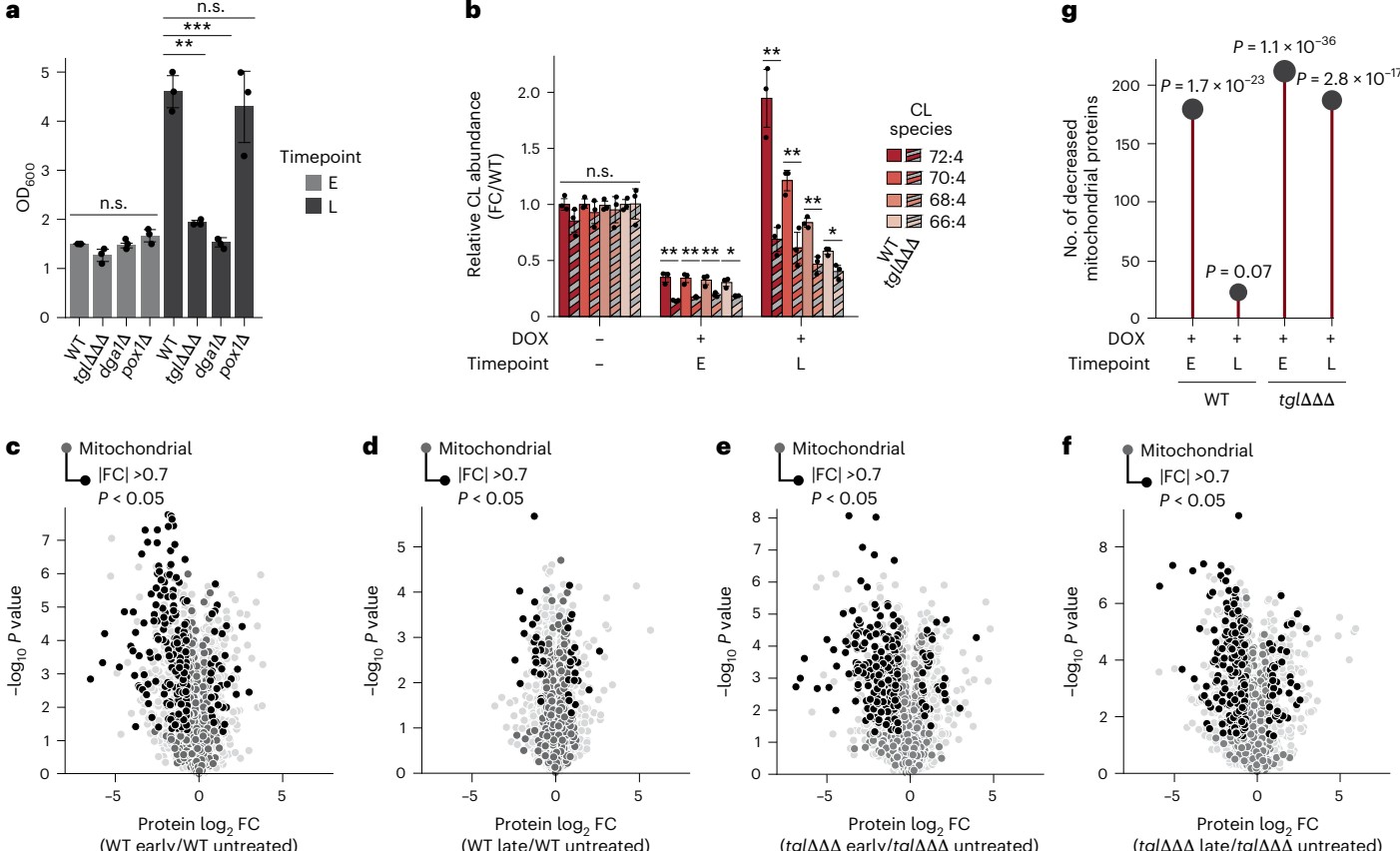

**Fig. 5 | Tgl-dependent TAG mobilization is essential for yeast to overcome mitochondrial stress. a**, Growth assay of WT, *TGL* triple deletion (*tgl*ΔΔΔ), *dga1*Δ and *pox1*Δ yeast treated with 400 μM DOX. Growth was determined after 24 h (E timepoint) or 48 h (L timepoint) in YPG respiratory media ($n = 3$, **$P = 2.17 \times 10^{-4}$, ***$P = 3.35 \times 10^{-4}$; two-sided Student's *t*-test). **b**, Normalized abundance of the four most abundant species of cardiolipin in WT (solid colours) or *TGL* triple deletion (*tgl*ΔΔΔ, striped colours) yeast treated with 400 μM DOX or DMSO control. Cells were grown for 24 h (E timepoint) or 48 h (L timepoint) in YPG respiratory media ($n = 3$). The abundance is normalized to the average WT values. **c,d**, Relative protein abundances for WT yeast treated with 400 μM DOX or DMSO control versus statistical significance: cells were grown for 24 h (WT early) (**c**) or 48 h (WT late) (**d**) in YPG respiratory media. The abundance is given by a log₂ transformed FC compared with the WT DMSO-treated vehicle at 24 h ($n = 3$). **e,f**, The relative protein abundances for *TGL* triple deletion (*tgl*ΔΔΔ) yeast treated with 400 μM DOX of DMSO control versus statistical significance: cells were grown for 24 h (*tgl*ΔΔΔ early) (**e**) or 48 h (*tgl*ΔΔΔ late) (**f**) in YPG respiratory media. The abundance is given by a log₂-transformed FC compared with the *tgl*ΔΔΔ DMSO-treated vehicle control at 24 h ($n = 3$). In **c–f**, non-mitochondrial proteins are coloured light grey and mitochondrial proteins are coloured dark grey. Mitochondrial proteins that are significantly changed (|FC| >0.7, $P < 0.05$; two-sided Student's *t*-test) are highlighted black. **g**, The number of significantly decreased mitochondrial proteins (FC <−0.7, $P < 0.05$; two-sided Student's *t*-test) as quantified in **c–f** with calculated enrichment. The enrichment is calculated using the Fisher exact test. For all experiments, the error bars are the s.d., centre values represent the mean and $n$ is the number of independent biological replicates. For analyses with more than four comparisons, *$P < 0.05$, **$P < 0.01$; two-sided Student's *t*-test.

## TAG processing drives stress recovery in mammalian cells

Finally, we sought to determine whether mammalian cells also rely on TAG mobilization to recover from mitochondrial stress. To test whether mammalian cells would recover from DOX treatment, we grew HAP1 cells in galactose-based media, in which cells require active mitochondrial respiration for survival, and treated with DOX over a 6 day time course. DOX-treated HAP1 cells exhibited a clear growth lag following the swap into galactose media, but eventually grew to confluency, mirroring the growth pattern observed in our group 3 yeast strains (Fig. 6a). We next collected cells both pre- and post-recovery to mirror our yeast collections. Unlike the yeast system, treatment with DOX did not result in the global loss of mitochondrial proteins, but instead elicited a more specific decrease in nuclear- and mtDNA-encoded OXPHOS subunits (Extended Data Fig. 6a). Similar to the yeast strains, these mitochondrial proteins recovered in abundance over time (Fig. 6b).

To examine the activity of TAG mobilization in this recovery phenotype, we performed untargeted lipidomics on the cells grown throughout the galactose time course. While DOX-treated cells had higher TAG abundance at day 2 (Extended Data Fig. 6b), they consistently depleted

their TAG stores between days 2 and 5 (Fig. 6c). We next tested whether the mammalian functional homologue of the yeast *TGL* genes, *ATGL*, was required for this TAG reduction and cellular growth recovery. Indeed, DOX-treated *ATGL* knockout cells had increased TAG abundance by day 5, indicating that *ATGL* was at least partially responsible for the observed TAG mobilization (Fig. 6d). Remarkably, we observed that knockout of *ATGL* was synthetically lethal with DOX treatment, validating the importance of TAG mobilization in overcoming mitochondrial stress in mammalian cells (Fig. 6e,g). Comparable to the yeast model, mobilized acyl groups from catabolized TAGs seemingly were not consumed via FAO, as disruption of carnitine palmitoyltransferase II (CPT2)—an enzyme essential for normal FAO—caused no further growth defect with DOX treatment (Fig. 6f,g). Collectively, these data confirm that, in both mammalian and yeast systems, TAG mobilization is essential for the recovery from certain mitochondrial stress.

## Discussion

Mitochondria are essential for many cellular processes, and yet cells have a considerable capacity to tolerate mitochondrial defects. This

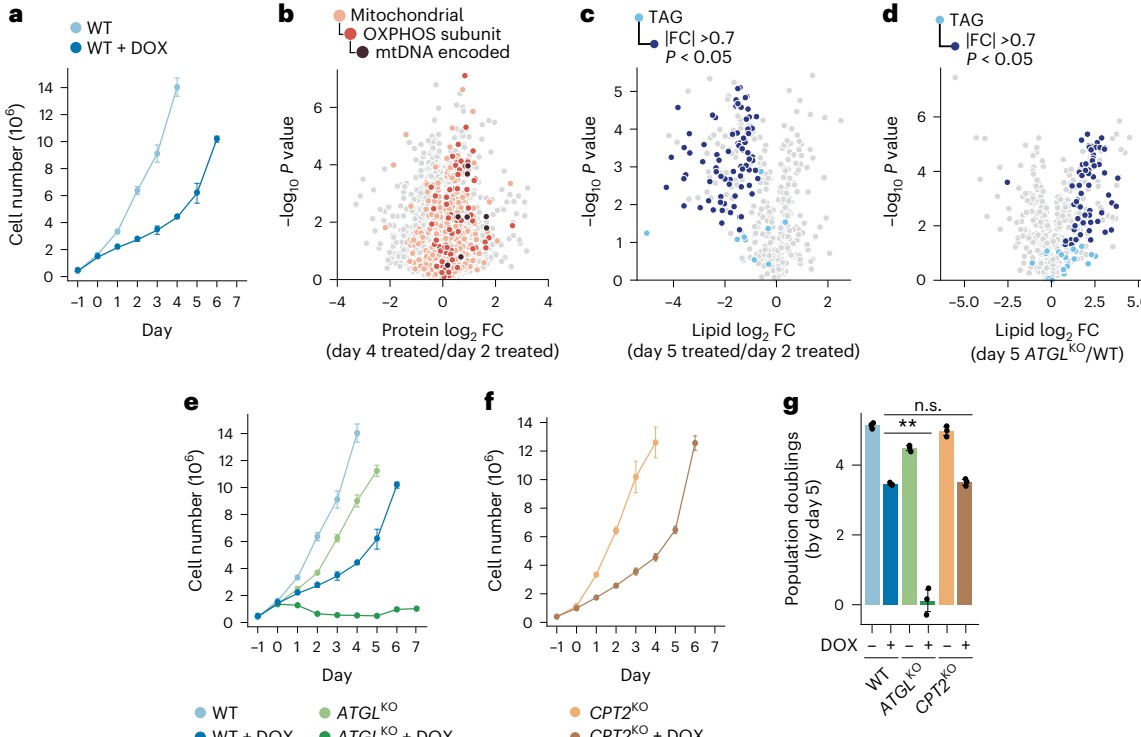

**Fig. 6 | TAG mobilization is required for mammalian cells to overcome mitochondrial stress. a**, A growth assay of WT HAP1 cells treated with 10 μM DOX (dark blue) or DMSO control (light blue). Cells were plated into glucose-containing media for 24 h before being swapped into galactose media (day 0) to measure respiratory growth over 4–6 days (n = 3). **b**, The relative protein abundances for DOX-treated WT HAP1 cells from **a** collected on day 2 or 4 versus statistical significance. The abundance is the $\log_2$ transformed FC compared with the day 2 DOX-treated sample. Non-mitochondrial proteins (grey), mitochondrial proteins (light orange), OXPHOS (dark orange) and mtDNA-encoded proteins (dark brown) are highlighted by colour (n = 3; two-sided Student's t-test). **c**, The relative lipid abundance versus statistical significance for DOX-treated WT HAP1 cells from **a** collected on day 2 or 5 versus statistical significance. **d**, The relative lipid abundance versus statistical significance for *ATGL* knockout HAP1 cells (*ATGL*^KO) grown in galactose media and treated with 10 μM DOX for

5 days compared with 5-day-treated WT cells. In **c** and **d**, the quantified TAG species are coloured light blue with TAG species that are significantly increased (FC > 0.7, P < 0.05; two-sided Student's t-test) (**d**) or decreased (FC < −0.7, P < 0.05; two-sided Student's t-test) (**c**) highlighted dark blue. All other lipid species are coloured grey (n = 3). **e**, Growth assay of WT HAP1 (from **a**) or *ATGL*^KO cells treated with 10 μM DOX (dark blue or green, respectively) or DMSO vehicle control (light blue or green, respectively) grown in the same conditions as **a** (n = 3). **f**, Growth assay of *CPT2* knockout HAP1 cells (*CPT2*^KO) treated with 10 μM DOX (tan) or DMSO vehicle control (peach) grown in the same conditions as **a** (n = 3). **g**, The growth phenotypes at day 5 of WT, *ATGL*^KO and *CPT2*^KO HAP1 cells under vehicle or DOX treatment as depicted in **a**, **e** and **f**. Growth was calculated as the number of population doublings after 5 days (n = 3, **P = 1.14 × 10⁻⁴; two-sided Student's t-test). For all experiments, the error bars are the s.d., centre values represent the mean and n is the number of independent biological replicates.

tolerance is perhaps most evident by the extreme mutational load required to elicit an observable phenotype in patients with mtDNA heteroplasmy[52]. In turn, this indicates that cells invoke restorative response pathways to buffer mitochondrial stresses. Here, we designed a study to probe how yeast overcome a range of mitochondrial stressors with varying severity and identified a common and essential response for surmounting these perturbations. We found that cells faced with notable stress can rescue mitochondrial content by inducing mitochondrial biogenesis. Critically, this increase in mitochondrial biogenesis requires the capacity to mobilize intracellular TAG stores as a means to provide acyl groups to support mitochondrial lipid biosynthesis. Yeast strains that lacked the ability to use TAGs became sensitive to DOX treatment. Furthermore, we show that the necessity for TAG mobilization in overcoming mitochondrial stress is conserved in mammalian cells, suggesting that there is, perhaps, an underappreciated role for the regulation of lipid availability in mitochondrial disease.

This study is part of a growing narrative that lipid homeostasis becomes dysregulated during mitochondrial stress. While numerous groups have measured the abundance of lipid species during stress conditions, the results are conflicting. In *C. elegans*, previous work showed a decrease in the abundance of TAG species after treatment with DOX, potentially caused by an increase in TAG lipolysis[25]. By contrast, knockdown of mtHsp70 in *C. elegans* resulted in the induction

of a mitochondria-to-cytosolic stress response, characterized by a concomitant increase in TAG abundance[24]. Interestingly, these results match our DOX-treated and *ssc1* (yeast homologue of mtHsp70) yeast strains, respectively, in which only the DOX-treated strain exhibited depleted TAG species. Collectively, these studies point to the dysregulation of lipid metabolism during mitochondrial stress; however, their conflicting nature suggest these response pathways are more nuanced then previously realized and require further investigation.

How the lipidome changes in mammalian cells challenged with mitochondrial stress has also been an active area of investigation. Patients with primary mitochondrial disease commonly have markedly increased plasma TAG abundance[53] and mouse models with genetically induced mitochondrial dysfunction in enterocytes or hepatocytes will accumulate lipids in these tissues[54–56]. Similarly, cultured cells treated with antimycin A or rotenone, two inhibitors of the mitochondrial electron transport chain, displayed elevated TAG species due to an increase in TAG biosynthesis and decrease in beta-oxidation[57–59]. Increases in TAG abundance may perhaps lead to the incorrect assumption that TAG mobilization is not important under stress conditions. However, our results demonstrate that, despite the increase in TAG abundance in DOX-treated HAP1 cells, their inability to mobilize TAG stores leads to their demise. This suggests that increased TAG biogenesis may be tipping the scales to favour TAG accumulation despite increased reliance on mobilization.

The role of the lipolysis in overcoming mitochondrial stress in mammalian cells remains to be determined. The increase in cardiolipin abundance and the essentiality of the cardiolipin biosynthetic pathway that we observe in yeast treated with DOX may point to changes in mitochondrial architecture. It has long been appreciated that mammalian cells undergo changes to their mitochondrial network after CAM[16] or DOX[17] treatment. Furthermore, energy stress caused by glucose starvation will alter mitochondrial dynamics and result in mitochondrial elongation with an increase in cristae surface area[19,60]. Presumably such drastic changes in mitochondrial membrane abundance would involve a considerable injection of lipid species into the system, although this has yet to be fully explored. It is tempting to speculate that lipolysis of TAG stores could be involved in this process. Transfer of the acyl-tails from TAGs to phospholipids could then be used to supply lipids needed for rapid mitochondrial membrane expansion, akin to that seen in autophagosome formation[61]. Strains that lack this ability may then lack the required space for mitochondrial protein expression, resulting in the failure to rescue mitochondrial protein abundance observed in our *tglΔΔΔ* strains, even though they maintain WT levels of mtDNA.

The mechanism of drawing from TAG stores to feed membrane expansion is well established in yeast. Following the exit from stationary the phase, rapid growth requires the activity of *TGL3* or *TGL4* to provide phospholipid substrates for membrane biosynthesis[46]. Similarly, spore membrane formation and sporulation efficiency also require these proteins[46]. Our observations that the *tglΔΔΔ* mutant could not efficiently recover from mitochondrial stress and exogenous free fatty acids did not rescue this phenotype suggest that these proteins are also essential for mitochondrial stress recovery. Interestingly, Tgl3-5p proteins in *S. cerevisiae* contain both lipase and acyltransferase domains[46,47], with the latter being most critically important for rescuing spore formation[46]. Determining the contributions of the acyltransferase domain in overcoming mitochondrial stress is an intriguing future direction. Furthermore, as the mammalian TAG lipases do not contain acyltransferase domains, this may indicate that there are additional acyltransferase proteins that could participate in mitochondrial recovery. It would be interesting to assess whether any of the multiple, poorly characterized acyltransferases that localize to mitochondria could have a role in this process.

Remaining open questions concern the signal that initiates TAG mobilization and why certain strains seemingly lack the ability to mount a mobilization response. One possibility is the signal requires mtDNA. This model would be in line with our observation that the *ssc1* strain—the only strain with no detectable mtDNA—lacked the ability to mobilize TAGs under these circumstances. An alternative possibility is that the signal requires a baseline level of OXPHOS function before being activated. Given that a baseline level of OXPHOS function is required to maintain mtDNA levels, these models are difficult to separate. Interestingly, the *crd1Δ* mutant that lacked all cardiolipin also failed to mobilize TAG species, suggesting that this mitochondrial phospholipid may have an unappreciated role in activating TAG mobilization.

Our deep multiomic analyses enabled us to draw further connections between proteins and TAG abundance. In particular, we linked the lipid droplet protein Pln1p to TAG mobilization during mitochondrial stress. The overexpression of *PLN1* prevented efficient lipolysis required for surmounting mitochondrial stress, the first direct evidence that Pln1p has a role in lipolysis. Mammalian cells have five distinct perilipins (PLIN1–PLIN5), whereas Pln1p is the only known perilipin in *S. cerevisiae* expressed under normal growth conditions[62]. Pln1p is involved in lipid droplet biogenesis, where it binds to nascent lipid droplets at an early stage and facilitates their expansion, akin to the function of the mammalian PLIN3 (ref. 44). Additionally, Pln1p has been found localized to lipid droplet–mitochondrial contact sites, suggesting it may have overlapping functions with PLIN5 (ref. 63). It is plausible that Pln1p performs more general, widespread perilipin functions rather than its more specialized mammalian counterparts. Fitting with this possibility, both PLIN2 and PLIN3, but not PLIN1, were able to rescue the lipid droplet biogenesis phenotype observed in the *pln1Δ* yeast strain[44]. The role of Pln1p in lipolysis is less clear. Deletion of *PLN1* destabilizes lipid droplets[44], therefore localized removal of Pln1p may be required for efficient lipolysis. It cannot be ruled out that Pln1p may have a more active role in lipolysis inhibition. Future studies on Pln1p will elaborate on these functions, as well as determine what role, if any, mammalian perilipins play in overcoming mitochondrial stress.

Finally, our results provide the first indication that TAG mobilization may be a valid therapeutic target for human diseases involving mitochondrial stress. Increasing mitochondrial biogenesis to overcome mitochondrial stress has been proposed to treat mitochondrial diseases[64,65] and neurological conditions[66,67]. Our results would indicate that the bioavailability of acyl groups may become rate limiting under these conditions and that potentially increasing phospholipid availability could be considered to help aid in treatment. Furthermore, many cancers are reliant on mitochondrial function[7] and increased metabolic stress in these cell lines makes their mitochondria a target for chemotherapeutic agents[8,68]. Here, combinatorial therapies that target both mitochondrial function and lipid availability may represent an intriguing means to boost drug efficacy.

In total, our dataset provides the quantification of over 8,500 biomolecules for each strain used in the study, comprising comprehensive coverage of the expressed yeast proteome, many major lipid classes, and metabolites from the most well-known metabolic pathways, all within the context of mitochondrial stress. Beyond the biological insights we provide here, this dataset will provide a rich community resource for further exploration of the multifaceted mitochondrial stress-response pathways.

## Data and material availability

All MS data (proteomics, lipidomics and metabolomics) have been deposited in Massive with the primary accession codes MSV000092267 (multiomic screen) and MSV000095028 (follow-up experiments). Source data are provided with this paper. The following databases were used in the searching of MS files: Biocyc, Human Metabolome Database, Kyoto Encyclopedia of Genes and Genomes, mzCloud, MassBank, MitoCarta and Uniprot. All other data supporting the findings of this study are available from the corresponding author on reasonable request.

### Reporting summary

Further information on research design is available in the Nature Portfolio Reporting Summary linked to this article.

## Online content

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

## Methods

### Yeast strain generation and culture conditions

The *S. cerevisiae* haploid strain W303 (MATa leu2 trp1 can1 ura3 ade2 his3) was used and cultured under standard laboratory conditions. Single (*gene*Δ), double (*gene*Δ*gene*Δ) and triple (*gene*Δ*gene*Δ*gene*Δ) deletion strains (Supplementary Table 1) were generated using homologous recombination where open reading frames were replaced with the kanMX6, His3MX6, HygMX6, Leu2MX6 and/or Trp1 cassettes transformed using standard heat shock conditions and confirmed via PCR genotyping[69,70]. Point mutants in the relevant genes were generated with a combined CRISPR–Cas9 vector[71] (Addgene, plasmid 81191), using 500 bp donor DNA containing the relevant point mutations. For overexpression strains, plasmids with arms homologous to the inert HO locus containing the Ura3 cassette (Addgene, plasmid 51663) were first digested using the NotI restriction enzyme (NEB) and integrated into the genome though homologous recombination as above[72]. DOX (200 µM, Biogems) and CAM (300 µM, Sigma-Aldrich) treated strains were treated with their respective compound throughout the secondary growth.

For the multiomic screen, strains from glycerol stocks were first struck out YPD plates consisting of 1% (w/v) yeast extract ('Y') (Research Products International), 2% (w/v) peptone ('P') (Research Products International), 2% (w/v) dextrose ('D') (Fisher) and 2% (w/v) agar (Sigma-Aldrich) and allowed to grow for 48 h at 30 °C. Preliminary starter cultures were inoculated from individual colonies in 3 ml YPD media and incubated for 24 h (30 °C, 230 rpm). Cell density was measured at $OD_{600}$ and $1.25 × 10^6$ cells from each starter culture were used to inoculate 50 ml of respiratory YPG media (1% (w/v) Y, 2% (w/v) P, 0.1% D and 3% glycerol ('G')) in a sterile 250 ml Erlenmeyer flask. Samples were incubated (30 °C, 230 rpm) and $1 × 10^8$ cells were collected at 24 h, a timepoint that corresponds to early respiratory growth. Strains with a noticeable growth lag (>20 min) were continuously monitored and a subsequent $1 × 10^8$ cells were collected at an OD that was at least 90% of the average WT OD at the 24 h timepoint. The samples were collected by centrifugation (4,000*g*, 5 min, room temperature). The supernatant was removed and the cells washed with 1 ml of sterile water. Cells were pelleted again (12,000*g*, 1 min, room temperature) and the supernatant was removed. Cell pellets were snap frozen in liquid nitrogen ($LN_2$) and stored at −80 °C until analysis.

### HAP1 cell culture conditions

HAP1 WT and knockout cells (Horizon Discovery; Supplementary Table 5) were cultured in Iscove's modified Dulbecco media (Thermo) with 10% heat-inactivated FBS (Biotechne) and 1× penicillin–streptomycin (Thermo) at 37 °C and 5% $CO_2$. For galactose growth assays, $5 × 10^6$ cells per plate were seeded in triplicate into 10 cm dishes with 10 µM DOX (Biogems) or vehicle control and incubated overnight to allow cells to adhere to the plate. Cells were washed with 1× Dulbecco's phosphate-buffered saline (DPBS) and medium was replaced with glucose-free Dulbecco's modified Eagle media (Thermo) supplemented with 25 mM galactose, 10% dialysed FBS (Biotechne) 1× penicillin–streptomycin and 10 µM DOX or DMSO vehicle control. Cells were collected at the indicated timepoint and cell counts were acquired with the TC20 Automated Cell Counter (Bio-Rad). Population doubling was calculated as $\log_2$(final density/seeding density). For proteomic and lipidomic analysis $2 × 10^6$ cells were collected at the indicated timepoints, snap frozen in $LN_2$ and stored at −80 °C until analysis.

### Respiratory growth assays

Starter cultures (YPD, 3 ml) were inoculated with individual colonies and incubated overnight (30 °C, 230 rpm, 14–16 h). For plate reader-based assays, cells were pelleted and resuspended in respiratory media (YPG) at a density of $5 × 10^6$ cells per ml. Then, 100 µl of the resuspended cells were transferred to a sterile 96-well round-bottom plate (Thermo) with a Breathe-Easy cover seal (Diversified Biotech). Cultures were incubated (30 °C, 1140 rpm) in an Epoch2 plate reader (BioTek) with $OD_{600}$ measured every 10 min. Growth rates and lag times were calculated

using Gen5 v3.02.2 software (BioTek), excluding timepoints before the diauxic shift and during the stationary phase growth. For larger culture growth assays, $2.5 × 10^4$ cells per ml were used to inoculate 50 ml of YPG culture in a sterile 250 ml Erlenmeyer flask. Samples were incubated (30 °C, 230 rpm, 14–16 h) and then the OD was measured at the desired timepoints using a NanoDrop One spectrophotometer (Thermo Scientific). Where indicated, 200 µg ml⁻¹ DOX was dissolved into media either immediately before inoculation for chronic treatment or spiked in at the desired timepoint for acute treatments. For experiments involving overexpression of *PLN1* in synthetic deficient (SD) media, growth times were as above but cells were grown in SD media (0.67% yeast nitrogen base, 0.2% Ura− drop-out mix) containing either fermentative (2% D) or respiratory (3% G and 0.1% D) carbon sources.

### [¹³C₁]oleic acid heavy labelling conditions

For initial labelling experiments, 3 ml starter and 50 ml secondary growth cultures were inoculated as in 'Yeast strain generation and culture conditions' section with WT cells either treated with DOX (400 µM) or DMSO vehicle. [¹³C₁]oleic acid (Cambridge Isotope Labs) was added to the growth media at a concentration of 30 µM, 1 h before collection (23 h after inoculation). Then, $1 × 10^8$ cells were collected at 24 h growth time. For subsequent experiments involving acute DOX treatment, WT cells were grown as above, but after 1 h of ¹³C labelling, 400 µM DOX was spiked into media, either with or without 10 mg l⁻¹ cerulenin for the pulse or pulse chase respectively. Next, $1 × 10^8$ cells were collected at the time of the DOX/cerulenin injection ($T_0$) and again either 1, 3 or 5 h later ($T_{1–5h}$). Cells for both experiments were centrifuged (4,000*g*, 5 min, 4 °C) and washed with 1 ml of water. Cells were then snap frozen in $LN_2$ and stored at −80 °C until analysis.

### Biomolecule extraction for multiomic screen

Samples were grouped into multiple batches and were randomized to mitigate batch effects on the overall study. The detailed extraction procedures were as follows: all reagents were chilled on ice, and samples were maintained at ≤4 °C during the extraction procedure. A 5-mm-diameter stainless metal bead (Qiagen) was first added to each sample. Next, 500 µl of M1 (75% (v/v) methyl *tert*-butyl ether (MTBE; Sigma-Aldrich), 25% (v/v) LC–MS grade methanol (Fisher) was added and tubes were vortexed for 2 min. Then, 325 µl of M2 (75% (v/v) filtered water, 25% (v/v) methanol) was added to each tube. Samples were vortexed briefly then snap frozen in $LN_2$ and thawed on ice three times to facilitate cell breakage. Samples were transferred to a bead beater and shaken at 1/25 s frequency for 5 min, and this process was done three times. The samples were then centrifuged (12,500*g*, 10 min, 4 °C). For downstream lipid analysis, 200 µl of the organic layer (upper phase) was transferred to a glass autosampler vial and dried by vacuum centrifugation. For downstream metabolomic analysis, 200 µl of the aqueous layer (lower phase) was transferred to a glass autosampler vial and dried by vacuum centrifugation. The remaining protein pellets were kept on ice until protein digestion.

Once dried, organic extracts intended for lipid analysis were resuspended in 100 µl of 65% (v/v) isopropyl alcohol (IPA; Fisher), 30% (v/v) acetonitrile (ACN; Fisher) and 5% (v/v) sterile water and vortexed for 20 s before analysis by LC–MS. Aqueous extracts intended for metabolomic analysis were resuspended in 50 µl of 50% (v/v) ACN and 50% (v/v) sterile water and also vortexed for 20 s before analysis by LC–MS.

### Proteomics LC–MS data acquisition and analysis for multiomic screen

Protein pellets were washed with 1 ml ACN and centrifuged (1,000*g*, 3 min, 4 °C). The supernatant ACN was aspirated and pellets were allowed to sit for 10–15 min at room temperature or vacuum dried briefly to allow for evaporation of the liquid remaining in the tube. Next, 300 µl lysis buffer (8 M urea (Sigma-Aldrich) with 100 mM tris(2-carboxyethyl)phosphine (Sigma-Aldrich), 40 mM

2-chloroacetamide (Sigma-Aldrich) and 100 mM Tris (Sigma-Aldrich, pH 8.0)) was added to each sample and vortexed until the protein pellets were fully dissolved. Then, 5 µg LysC (Wako Chemicals) was added to each sample with a protein:enzyme ratio of 70:1 and digestion was proceeded overnight at room temperature. Each sample was diluted with 100 mM Tris to reach a final concentration of 2 M urea. Trypsin (Promega) was added at 70:1 protein:enzyme ratio and digestion proceeded for 6 h at room temperature. Desalting was carried out with 96-well desalting plates (10 mg per well, StrataTM-X 33 µm Polymeric Reversed phase, Phenomenex). A blank well between any two samples was reserved to avoid cross contamination. Desalting started with equilibrating the desalting wells with 1 ml of 100% ACN, followed by 1 ml of 0.2% (v/v) formic acid (FA; Thermo Scientific). The acidified peptide mixture was loaded to the 96-well desalting plate, followed by 2 ml 0.2% (v/v) FA wash. Peptides were eluted into a 96-well collection plate with 600 µl 80% (v/v) ACN with 0.2% (v/v) FA. Peptides were dried by vacuum centrifugation and stored at −80 °C until resuspension with 0.2% (v/v) FA. After resuspension, peptide concentration was measured using a quantitative colorimetric peptide assay according to the manufacturer's protocols (Pierce, Thermo Scientific).

Peptides were separated on an in-house prepared high-pressure reversed-phase C18 column. In brief, a 75–360 µm inner–outer diameter bare-fused silica capillary was packed, with 1.7 µm diameter, 130 Å pore size, bridged ethylene hybrid C18 particles (Waters) under high pressure of 25,000 psi to a final length of ~40 cm (ref. [73]). The column was installed onto a Thermo Ultimate 3000 nano LC and heated to 50 °C for all runs. Mobile phase buffer A was composed of water with 0.2% (v/v) FA. Mobile phase B was composed of 70% (v/v) ACN with 0.2% (v/v) FA. Samples were separated with a 90 min LC method: peptides were loaded onto the column for 9 min at 0.30 µl per min. Mobile phase B was increased from 0% to 10% in 9 min, then increased to 55% B by 74 min, and increased to 100% B by 75 min and held for 4 min at 100% B, then decreased to 0% B by 80 min and allowed to equilibrate for 10 min 0% B.

Eluting peptides were ionized by electrospray ionization and analysed on a Thermo Orbitrap Eclipse. Survey scans of precursors were taken from 300 to 1,350 $m/z$ at 240,000 resolution (at 200 $m/z$). The maximum injection time was set to 50 ms and the automatic gain control (AGC) target was 250%. Tandem MS was performed using an isolation window of 0.5 Th with a dynamic exclusion time of 10 s. Selected precursors were fragmented by higher energy collisional dissociation using a normalized collision energy (NCE) level of 25%. The MS2 AGC target was set to $3 \times 10^4$ ions with a maximum injection time of 14 ms. The scan range was 150–1,350 $m/z$. Scans were taken using the Turbo speed setting and only peptides with a charge state of +2 or greater were selected for fragmentation.

LC–MS files for proteomics were searched in Maxquant (version 1.5.5.5) against the downloaded *S. cerevisiae* proteome database from Uniprot. Original outputs from Maxquant were inspected and potential contaminant proteins, protein groups that contain proteins identified with decoy peptide sequence and those identified only with a modification site were removed. Label-free quantification intensities were used as the quantification metric. To replace missing values, log2 transformation and imputation in Perseus was performed. The parameters for imputation were the default settings, with width: 0.3, down shift: 1.8 and mode: separately for each column.

### Proteomics for HAP1 cells

Cell pellets were resuspended in 200 µl 2% SDS containing cOmplete Protease Inhibitor Cocktail (Sigma-Aldrich) and heated at 95 °C for 5 min. Nucleic acids were sheared with 2 µl benzonase (Sigma-Aldrich) and samples were incubated on ice for 15 min. Protein content was quantified using the BCA assay (Pierce, Thermo) and 100 µg of protein was alkylated and reduced in digestion solution (10 mM tris(2-carboxyethyl)phosphine, 40 mM 2-chloroacetamide and 100 mM Tris (pH 8.0)) for 30 min at room temperature. Protein was subjected to single-pot, solid-phase-enhanced sample preparation (SP3) to remove

detergent by incubating with magnetic carboxylated SpeedBeads (Sigma). After 1 h incubation to allow protein binding, beads were washed with 80% (v/v) ethanol and allowed to dry. The beads were resuspended in 100 µl of 100 mM Tris (pH 8.0) and trypsin (Promega) was added to each sample in an estimated 50:1 protein:enzyme ratio to digest at 37 °C for 16 h. The supernatant containing tryptic peptides was collected and acidified with trifluoroacetic acid (Sigma-Aldrich) to a final pH of 2. Peptides were desalted by solid-phase extraction cartridges (Phenomenex) and dried under vacuum.

Samples were resuspended in 0.2% FA and subjected to LC–MS analysis. LC separation was performed using the Thermo Ultimate 3000 RSLC nano system. A 15 cm EASY-Spray PepMap RSLC C18 column (Thermo, 150 mm × 75 µm, 3 µm) was used at 300 nl per min flow rate with an Acclaim PepMap C18 HPLC trap column (Thermo, 20 mm × 75 µm, 3 µm) for sample loading. For each sample run, the temperature was held at 35 °C for a 120 min gradient that consisted of 4% B for 5 min and increased to 30% B over 100 min, followed by 5 min at 99% B and back to 4% B for equilibration for 10 min. Mobile phase A consisted of 0.1% FA in water, and mobile phase B consisted of 0.1% FA in 80% (v/v) ACN and 20% (v/v) water.

MS detection was performed with a Thermo Exploris 240 Orbitrap mass spectrometer with an EASY-Spray source operating in positive mode. The source voltage was 1.8 kV, the ion transfer tube temperature was set to 275 °C and the RF lens at 70%. Full MS spectra were acquired from $m/z$ 350 to 1,400 at the Orbitrap resolution of 60,000, with a normalized AGC target of 300% ($3 \times 10^6$). Data-dependent acquisition was performed with a 3 s duty cycle with a charge state of 2–6, an isolation window width of 2 and an intensity threshold of $5 \times 10^3$. Dynamic exclusion was 20 s with the exclusion of isotopes. Other settings for data-dependent acquisition were an Orbitrap resolution of 15,000 and higher energy collisional dissociation energy of 30%. Raw files were analysed by the SequestHT Search Engine incorporated in Proteome Discoverer v.2.5.0.400 software against human databases downloaded from Uniprot. Label-free quantification was enabled in the searches.

### Lipidomics LC–MS data acquisition and analysis for multiomic screen

Extracted lipids were separated on an Acquity CSH C18 column (100 mm × 2.1 mm × 1.7 µm particle size; Waters) at 50 °C using the following gradient: 2% mobile phase B from 0 to 2 min, increased to 30% B over the next 1 min, increased to 50% B over the next 1 min, increased to 85% over the next 14 min, increased to 99% B over the next 1 min, then held at 99% B for the next 7 min (400 µl per min flow rate). Column re-equilibration of 2% B for 1.75 min occurred between samples. For each analysis 10 µl per sample was injected by the autosampler. Mobile phase A consisted of 10 mM ammonium acetate (Sigma-Aldrich) in 70% (v/v) ACN, 30% (v/v) water with 250 µl l⁻¹ acetic acid (Sigma-Aldrich). Mobile phase B consisted of 10 mM ammonium acetate in 90% (v/v) IPA and 10% (v/v) ACN with 250 µl l⁻¹ acetic acid.

The LC system (Vanquish Binary Pump, Thermo Scientific) was coupled to a Q-Exactive Orbitrap mass spectrometer through a heated electrospray ionization (HESI II) source (Thermo Scientific). Source and capillary temperatures were 300 °C, the sheath gas flow rate was 25 units, the aux gas flow rate was 15 units, the sweep gas flow rate was 5 units, the spray voltage was |3.5 kV| for both positive and negative modes, and the S-lens RF was 90.0 units. MS was operated in a polarity switching mode, with alternating positive and negative full-scan MS and MS2 (top 2). Full-scan MS were acquired at 17,500 resolution (at 200 $m/z$) with $1 \times 10^6$ AGC target, max ion accumulation time of 100 ms and a scan range of 200–1,600 $m/z$. MS2 scans were acquired at 17,500 resolution (at 200 $m/z$) with $1 \times 10^5$ AGC target, max ion accumulation time of 50 ms, 1.0 $m/z$ isolation window, stepped NCE at 20, 30 and 40, and 10.0 s dynamic exclusion.

LC–MS files for lipidomics were processed using Compound Discoverer 3.1 (Thermo Scientific) and LipiDex[74]. All peaks with a 1.4–23 min

retention time and 100–5,000 Da MS1 precursor mass were aggregated into compound groups using a 10 ppm mass tolerance and 0.4 min retention time tolerance. Peaks were excluded if peak intensity was less than $2 \times 10^6$, peak width was greater than 0.75 min, signal-to-noise ratio was less than 1.5 or intensity was <3-fold greater than the blank. MS2 spectra were searched against an in silico generated spectral library[75]. Spectra matches with a dot product score >500 and reverse dot product score >700 were retained for further analysis. Lipid MS/MS spectra that contained <75% interference from co-eluting isobaric lipids, eluted within a 3.5 median absolute retention time deviation of each other and were found within at least four processed files were used for identification at the individual fatty acid substituent levels of structural resolution. If individual fatty acid substituents were unresolved, then identifications were made with the sum of the fatty acid substituents. Lipid identifications were filtered with our in-house developed Degreaser module within LipidDex2 (v0.1.0)[76], based on retention time modelling. The retention time tolerance used was 0.5 min. Unreliable identifications were discarded. Further filtering based on instrument quality control (QC) coefficient variance was carried out and only features with below 30% QC coefficient variance were kept for further analysis.

### Follow-up lipidomics experiments

Lipids from cell pellets in follow-up experiments were extracted using MTBE (Sigma-Aldrich). Frozen cell pellets were resuspended in 225 µl 100% LC-grade methanol (Fisher), containing 1 µM $CoQ_8$ (Avanti Lipids) as an internal standard. Glass beads (100 µl, 0.5 mm; BioSpec) were then added and the samples were vortexed using a Vortex Genie for 10 min (3,000 rpm, 4 °C) to lyse the cells. Next, 187.5 µl of water and 750 µl of MTBE were added to each sample and the tubes were vortexed again for 3 min (3,000 rpm, 4 °C). To separate the layers, the samples were centrifuged for 3 min (1,000$g$, 4 °C). The organic (top) layer was removed into a separate microcentrifuge tube and a new 750 µl of MTBE was added. Organic extraction was repeated a second time with the second MTBE layer added to the first. Samples were dried by vacuum centrifugation and resuspended in 50 µl of 20 mM ammonium acetate in 78% (v/v) methanol, 20% IPA (Sigma-Aldrich) and 2% water.

LC instrumentation and separation conditions were identical to those used for the multiomic screen. MS acquisition was performed by a Thermo Exploris 240 Orbitrap mass spectrometer. Samples were ionized by a HESI II source (Thermo Scientific) kept at a vaporizer temperature of 350 °C. The sheath gas was set to 50 units, auxiliary gas to 8 units and sweep gas to 1 unit. For untargeted discovery lipidomics, the MS was operated in polarity switching mode with the spray voltage set to 3,500 V for positive mode and 2,500 V for negative mode. The inlet ion transfer tube temperature was kept at 325 °C with 70% RF lens. Full MS1 scans were acquired at 22,500 resolution (at 200 $m/z$), a max ion accumulation time of 100 ms and with a scan range of $m/z$ 200–1,600. MS2 scans (top 3) were acquired at 30,000 resolution (at 200 $m/z$), max ion accumulation time of 50 ms, a 1.0 $m/z$ isolation window, stepped NCE at 20, 30 and 40, and a 10.0 s dynamic exclusion. Automatic gain control (AGC) targets were set to standard mode for both MS1 and MS2 acquisitions. Raw files were analysed as described above for the multiomic screen, apart from requiring the identified lipids to be found in at least four processed files and removal of the secondary QC coefficient variance filter. For targeted analyses, chromatography and mass spectrometer conditions were the same as untargeted analyses, except runs for quantification were acquired in only full MS1 scans in either positive or negative mode depending on the target. List of targeted compounds, polarity, targeted $m/z$ and retention times are given in Supplementary Table 6. Compound identification was first validated using previously published MS2 fragmentation patterns[77] and/or analytical standards (Avanti Lipids). Peak integration was performed on MS1 peaks that had at least >10 measurements across peak width using Tracefinder 5.1 (Thermo Scientific). For all analyses, the peak area was then normalized to the area of the $CoQ_8$ internal standard.

### Metabolomics LC–MS data acquisition and analysis for multiomic screen

Polar metabolites were separated on a Sequant ZIC-pHILIC HPLC column (100 mm × 2.1 mm × 5 µm particle size) at 50 °C using the following gradient: 95% mobile phase B from 0–2 min, decreased to 30% B over the next 16 min, held at 30% B for 8 min, then increased to 95% B over the next 1 min, then held at 95% B for the next 8 min. The flow rate was 130 µl per min. For each analysis, 2 µl per sample was loaded onto the column. Mobile phase A consisted of 10 mM ammonium acetate in 10% (v/v) ACN and 90% (v/v) water with 0.1% (v/v) ammonium hydroxide (Sigma-Aldrich). Mobile phase B consisted of 10 mM ammonium acetate in 95 (v/v) ACN and 5% (v/v) water with 0.1% ammonium hydroxide.

The LC system (Vanquish Binary Pump, Thermo Scientific) was coupled to a Q-Exactive HF Orbitrap mass spectrometer through a HESI II source (Thermo Scientific). Source and capillary temperatures were 350 °C, the sheath gas flow rate was 45 units, the aux gas flow rate was 15 units, sweep gas flow rate was 1 units, spray voltage was 3.0 kV for both positive and negative modes and the S-lens RF was 50.0 units. The MS was operated in a polarity switching mode, with alternating positive and negative full-scan MS and MS2 (top 10). Full-scan MS were acquired at 60,000 resolution (at 200 $m/z$) with a $1 \times 10^6$ AGC target, max ion accumulation time of 100 ms and a scan range of 70–900 $m/z$. MS2 scans were acquired at 45,000 resolution (at 200 $m/z$) with $1 \times 10^5$ AGC target, max ion accumulation time of 100 ms, 1.0 $m/z$ isolation window, stepped NCE at 20, 30 and 40, and a 30.0 s dynamic exclusion.

LC–MS files for metabolomics were processed using Compound Discoverer 3.3 (Thermo Scientific) in a discovery mode workflow. All peaks between 0 and 22 min retention time and 0–5,000 Da MS1 precursor mass were grouped into distinct chromatographic profiles (that is, compound groups) and aligned against the reference file (the QC file running in the middle of all the files). Profiles not reaching a minimum peak intensity of $5 \times 10^4$, a maximum peak width of 3 min, a signal-to-noise ratio of 1.5 and a fivefold intensity increase over blanks were excluded from further processing. Profiles having fewer than five points across the peak were also excluded. Element compositions were predicted with 5 ppm mass tolerance based on MS1 precursor mass. Precursors were matched to compounds by searching against databases including Biocyc, Human metabolome database and Kyoto Encyclopedia of Genes and Genomes. MS/MS spectra were searched against mzCloud (Thermo Scientific) containing 19,503 unique molecular compositions, mzVault libraries including in-house curated MS2 spectra of 151 standards, 598 polar compounds from Bamba lab, the Fiehn lab HILIC library of 3,061 entries, the KI-GIAR zic HILIC library of 814 entries and six other libraries from MassBank of North America. The resulting features were filtered based on the peak quality rating and only the features that had a peak rating greater than 4.0 (on a scale of 0–10) in at least 20 samples were kept for further analysis. Compound annotation was done manually by examining the formula composition, MS2 spectrum and retention time similarities to library entries. Only features with instrument QC coefficient variance below 30% were kept for further analysis.

### Multiomic data integration and visualization

All data analysis and visualizations were generated using Python (v3.9). Principal component analysis was performed using the sklearn Python module with the default settings. Hierarchical clustering was calculated using the seaborn Python module set to the Euclidean distance and $z$-transformed. Gene Ontology term enrichment was calculated using the ShinyGO 0.80 web interface with a false discovery rate cutoff of 0.05 (ref. [78]). To label mitochondrial protein volcano plots and perform enrichment tests, yeast proteins were deemed mitochondrial if they were present in the high-confidence yeast mitochondrial proteome[79]. Human proteins were deemed mitochondrial if present in MitoCarta 3.0 (ref. [80]). Proteins were classified as up- or downregulated if the absolute value of the fold change was >0.7 and the $P$ was <0.05. Enrichment was determined by a Fisher exact test. Correlation analysis was performed using all FCs

for each molecule under all experiment conditions. To determine the correlation score for a set of molecules, the Spearman correlation coefficient was calculated for every individual pairwise comparison between the molecules within that set and all others. Each molecule was then rank ordered based on the correlation coefficient and the correlation score was determined by taking the average rank for each molecule.

To train the support vector machine (SVM) model for the respiration deficient (RD) response, data and classifications of RD and respiration competent strains were compiled from previous work[26]. Only overlapping measurements from both datasets were considered for training. SVM was generated using the sklearn module from Python (kernel='rbf', probability=True, gamma=.03) with 70% of the data used for training and 30% withhold for testing. SVM model was deemed to accurately predict RD strains on the test set (accuracy: 0.98; precision: 0.90; recall: 1.0). SVM was then used to predict probability of RD in indicated strains from the current multiomic dataset.

## mtDNA measurement

Yeast were cultured and collected using conditions described in 'Yeast strain generation and culture conditions' section. Total DNA (genomic and mitochondrial) was extracted using previously described methods[27], then diluted to 100 ng μl$^{-1}$. Quantitative PCR (qPCR) was performed using the following reaction: 10 μl Power SYBR Green PCR Master Mix (Thermo), 1 μl DNA and 250 nM forward and reverse primers. Previously documented mtDNA and genomic DNA targeting primers were used[81]. For the qPCR cycle, an initial 2 min incubation at 50 °C was followed by 2 min denaturing at 95 °C. Then, 40 cycles of 95 °C for 15 s, 60 °C for 15 s and 72 °C for 1 min were performed. qPCR data were collected using QuantStudio Real-Time PCR software v1.2 (Applied Biosciences). mtDNA abundance was calculated using the $\Delta\Delta^{Ct}$ method and the relative level of mtDNA abundance was calculated and normalized to the expression level of actin[82]. The primers for qPCR are listed in Supplementary Table 7.

## *PLN1* mRNA measurement using qPCR

Yeast were cultured and collected using the conditions described in 'Yeast strain generation and culture conditions' section. Total RNA was extracted using the Mastepure Yeast RNA Purification kit (Lucigen), according to the manufacturer's protocols. Then, 2 μg of RNA was converted into cDNA using the SuperScript III kit (Thermo Scientific) with the random hexamer primers and standard reverse transcription reaction conditions. qPCR was performed using the following reaction: 10 μl Power SYBR Green PCR Master Mix (Thermo), 1 μl of a 1:20 dilution of cDNA and 250 nM forward and reverse primers. For the qPCR cycle, an initial 2 min incubation at 50 °C was followed by 2 min denaturing at 95 °C. Then, 40 cycles of 95 °C for 15 s, 60 °C for 15 s and 72 °C for 1 min were performed. qPCR data were collected using QuantStudio Real-Time PCR software v1.2 (Applied Biosciences). *PLN1* abundance was calculated using the $\Delta\Delta^{Ct}$ method and normalized to the expression level of actin. The primers for qPCR are listed in Supplementary Table 7.

## OCR measurement

The oxygen consumption rate (OCR) was measured using the MitoXpress Xtra Oxygen Consumption Assay kit (Agilent), according to the manufacturer's protocols. In brief, yeast were cultured as described in 'Yeast strain generation and culture conditions' section. Then, $5 \times 10^5$ cells were transferred into a clear, 96-well plate (Thermo) and the volume was adjusted to 100 μl. Next, 10 μl of MitoXpress Xtra probe was added to each well and the liquid was topped with four drops of mineral oil. Changes in fluorescence intensity were measured using a Cytation 3 plate reader (BioTek, 30 °C, 1,140 rpm, excitation: 380 nm and emission: 650 nm) over a 120 min period. The OCR was calculated using Gen5 v3.02.2 software (BioTek) measuring the Δrelative fluorescent units (RFU) over the linear time frame for each sample.

## Fluorescence microscopy

For imaging neutral lipids inside lipid droplets, cells were first grown as described in 'Yeast strain generation and culture conditions' section. Then, $1 \times 10^8$ cells were pelleted by centrifugation (3,000g, 5 min) and washed twice with 1 ml of 1× DPBS solution (Gibco). Cells were then incubated in DPBS containing 2 μM BODIPY 493/503 (Thermo) for 15 min at 30 °C. Cells were then centrifuged (3,000g, 5 min) and washed twice in 1× DPBS before imaging. Samples were mounted on Superfrost Plus slides (Fisher) with Fluoromount-G (SouthernBiotech). Yeast were imaged on a Zeiss LSM 880 II Airyscan FAST confocal microscope using a 100× objective. Fluorescent and differential interference contrast czi images were processed using ImageJ software (v2.9.0/1.53t)

## Nile red fluorescence

The Nile red staining protocol was adapted from previously described methods[83]. Cells were grown as described in 'Yeast strain generation and culture conditions' section. First, $2 \times 10^7$ cells were pelleted by centrifugation (500g, 2 min) and washed twice with 1 ml of 1× DPBS solution (Gibco). The pellet was resuspended in 1 ml of DPBS and 250 μl was transferred into a black 96-well clear-bottom plate (Thermo), along with 25 μl of 50% (v/v) DMSO and 50% (v/v) DPBS. Finally, 25 μl of 60 μg ml$^{-1}$ Nile red stain (Sigma-Aldrich) in acetone was added to each well. The RFU was measured rapidly using a Cytation 3 plate reader (BioTek, 30 °C, 1,140 rpm, excitation: 485 nm, emission: 535 nm, automatic gain) for a 10 min period. RFU values were calculated using Gen5 v3.02.2 software (BioTek) by averaging the fluorescent signal intensity over the 10 min time frame and subtracting background wells where 250 μl of DPBS was substituted for sample.

## Imaging mitochondrial volume density

*S. cerevisiae* diploid yeast expressing six fluorescently labelled, organelle-specific proteins (Supplementary Table 8) were used for imaging. The diploid strain was generated by mating two parental WT W303 haploid strains (EY2795 and EY2796) engineered to express three fluorescent organelle-specific marker proteins, using standard mating protocols.

Imaging was performed using a Nikon Ti2 microscope equipped with laser scanning A1-HD25 confocal scan head, a LUN4 4-line solid state laser illumination system and an A1-DUS spectral detector. Organelle images were captured in z-stacks (step size of 0.2 μm) with hyperspectral confocal microscopy at four different optical configurations each with a laser power of 5% (Supplementary Table 9). The spectral detector was set to have a resolution of 6.0 nm per bin. To distinguish overlapping fluorescence emission spectra, red and cyan fluorescent protein (CFP) channels were subject to linear unmixing using the in-built Nikon Elements software to acquire single channel z-stacks for the two organelles captured in each channel. Organelles were segmented by inputting corresponding single channel z-stacks into the interactive machine learning tool ilastik (v1.3.3)[84]. Using the ilastik pixel classification workflow, organelle and background classes were determined by user-defined, hand-drawn labels for each class. Ilastik then assigned each pixel a probability of belonging to either class. Using a threshold of 0.5, the probability map was distinguished into background and organelle, giving a binary image. To segment cells, an average intensity z-projection of the endoplasmic reticulum binary ilastik images was taken using ImageJ[85] and input into the convolutional neural network YeaZ (v1.0.3)[86].

A custom MATLAB code was used to binarize the cell and organelle images and estimate volume fractions. Cell size was estimated by treating cells as prolate ellipsoids. Average intensity-based cell size measurement was verified to be correlated to a brightfield image-derived cell-size estimate (correlation coefficient of 0.984, $P = 1.334 \times 10^{-74}$). The cell mask was applied to the organelle images over all z-planes and organelles were grouped into objects. To estimate the volume fraction, the pixels of each organelle within a cell were counted and converted into physical units and then normalized to the estimated cell size. Kernel density estimation was calculated using the seaborn Python module.

## Statistics and reproducibility

All experiments were performed in at least biological triplicate, unless otherwise stated. No statistical methods were used to predetermine sample sizes but our sample sizes are similar to those reported in previous publications[26,27,87]. Data distribution was assumed to be normal, but this was not formally tested. Randomization was used for MS injection to minimize batch effects. No other randomization was used for experimental groups. Data collection and analysis were not performed blind to the conditions of the experiments. No data were excluded from the analyses. For all assays, quantification and statistics were derived from $n = 3$ independent biological replicates unless specified in the legends. All statistical analysis was performed using Excel or Python. All results are presented as the arithmetic mean ± s.d. $P$ values were calculated using either unpaired, two-sided Student's $t$-test, or Fisher enrichment tests as specified in the methods and legends. $P$ values less than 0.05 were considered significant.

## Code availability

Custom MATLAB code used to binarize cell data is available without restrictions via GitHub (https://github.com/alinearra/CellandOrganelleAnalysis).

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

## Acknowledgements

We thank the Pagliarini Laboratory for their feedback and discussion throughout the duration of this study. This work was supported by National Institutes of Health awards R35GM131795 (to D.J.P.), R35GM118119 and P41GM208538 (to J.J.C.), R35GM127009 (to E.A.C) and R35GM142704 (to S.M.), as well as funds from the BJC Investigator Program (to D.J.P.). D.J.P. is an investigator of the Howard Hughes Medical Institute. This article is subject to HHMI's Open Access to Publications policy. HHMI lab heads have previously granted a nonexclusive CC BY 4.0 license to the public and a sublicensable license to HHMI in their research articles. Pursuant to those licenses, the author-accepted manuscript of this article can be made freely available under a CC BY 4.0 license immediately upon publication.

## Author contributions

Z.N.B., E.A.C. and D.J.P. conceived of the project and its design. Y.Z., Z.N.B., L.R.S. and K.A.O performed sample preparation, MS and data analyses for the initial screen. A.A. and S.M. performed volume fraction measurements. Z.N.B., R.M.G., A.J.S., J.J.C. and D.J.P. performed all other experimentation, MS and data analyses. Z.N.B. and D.J.P. wrote the original draft. All authors contributed to editing and revising of the final manuscript. D.J.P., E.A.C., J.J.C. and S.M. provided supervision and funding.

## Competing interests

J.J.C. is a consultant for Thermo Fisher Scientific, 908 Devices, and Seer. The remaining authors declare no competing interests.

## Additional information

**Extended data** is available for this paper at https://doi.org/10.1038/s41556-024-01586-6.

**Correspondence and requests for materials** should be addressed to David J. Pagliarini.

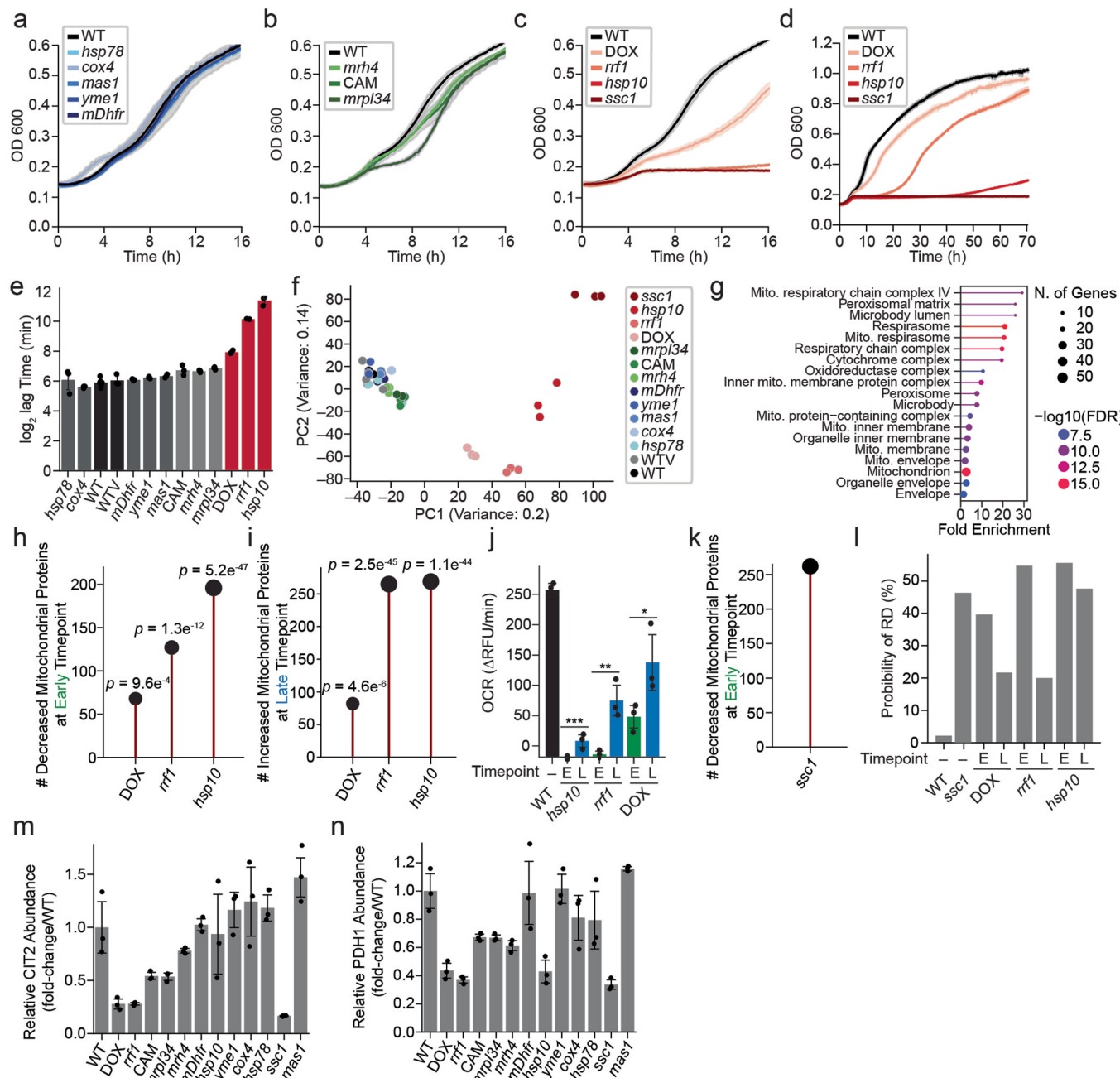

**Extended Data Fig. 1 | A multiomic mass spectrometry screen to identify requirements for overcoming mitochondrial disfunction. a-d,** Growth assays of WT and experimental yeast strains used in multiomic screen. Strains were grouped based on growth rate: WT-like growth (Group 1) (**a**), slight growth defect (Group 2) (**b**), large growth defect (Group 3) (**c**), and unrecoverable growth defect (Group 4, *ssc1*) (**d**). Growth was measured continuously in a 100 µl culture in YPG media (n = 3) **e,** log₂ transformed lag time calculated for WT and all experiment strains from growth assays (**a-d**) (n = 3). **f,** Principal component analysis (PCA) derived from the normalized abundances of all biomolecules for WT controls and all experimental strains used in multiomic screen (n = 3). **g,** Enriched GO-terms for proteins that had significantly decreased abundance (FC < -0.7, *p* < 0.05, two-sided Student's t-test) in combined Group 3 strains (n = 9) compared to the WT (n = 3) at the early time point. Size of ball represents number of proteins containing the respective GO term, color represents the false discovery rate (FDR), and the bar size represents the fold enrichment of the GO term over the expected frequency. **h,i,** Number of significantly decreased (FC < -0.7, *p* < 0.05, two-sided Student's t-test) mitochondrial proteins for Group 3 strains at the early time point (n = 9) (**h**) or increased mitochondrial proteins (FC > 0.7, *p* < 0.05, two-

sided Student's t-test) at the later time point (n = 9) (**i**) as quantified in Fig. 1f, g, with calculated enrichment. **j,** Oxygen consumption rate (OCR) of WT and Group 3 strains at the early and later time points. OCR was measured as a change in relative fluorescence over a linear period of either 60-, 90-, or 120-minutes depending on rate of change (n = 3, *P = 0.06, **P = 8.42×10⁻³, ***P = 1.92×10⁻², two-sided Student's t-test). **k,** Number of significantly decreased (FC < -0.7, *p* < 0.05, two-sided Student's t-test) mitochondrial proteins for the *ssc1* strain with calculated enrichment (n = 3). **l,** Probability of respiratory deficiency in WT, *ssc1*, and Group 3 strains at either the early (E) or later (L) time point. Probability was calculated by a support vector machine model trained on data obtained from a previous large-scale multiomic screen containing both respiratory competent and deficient yeast strains. **m,l,** Relative protein abundance of common retrograde response proteins Cit2p (**m**) and Pdh1p (**n**) in WT and experimental yeast strains (n = 3). For all experiments error bars are standard deviation, centre values represent the mean and n= number of independent biological replicates. All enrichments are calculated using a Fisher exact test. Numerical data are available in Source data.

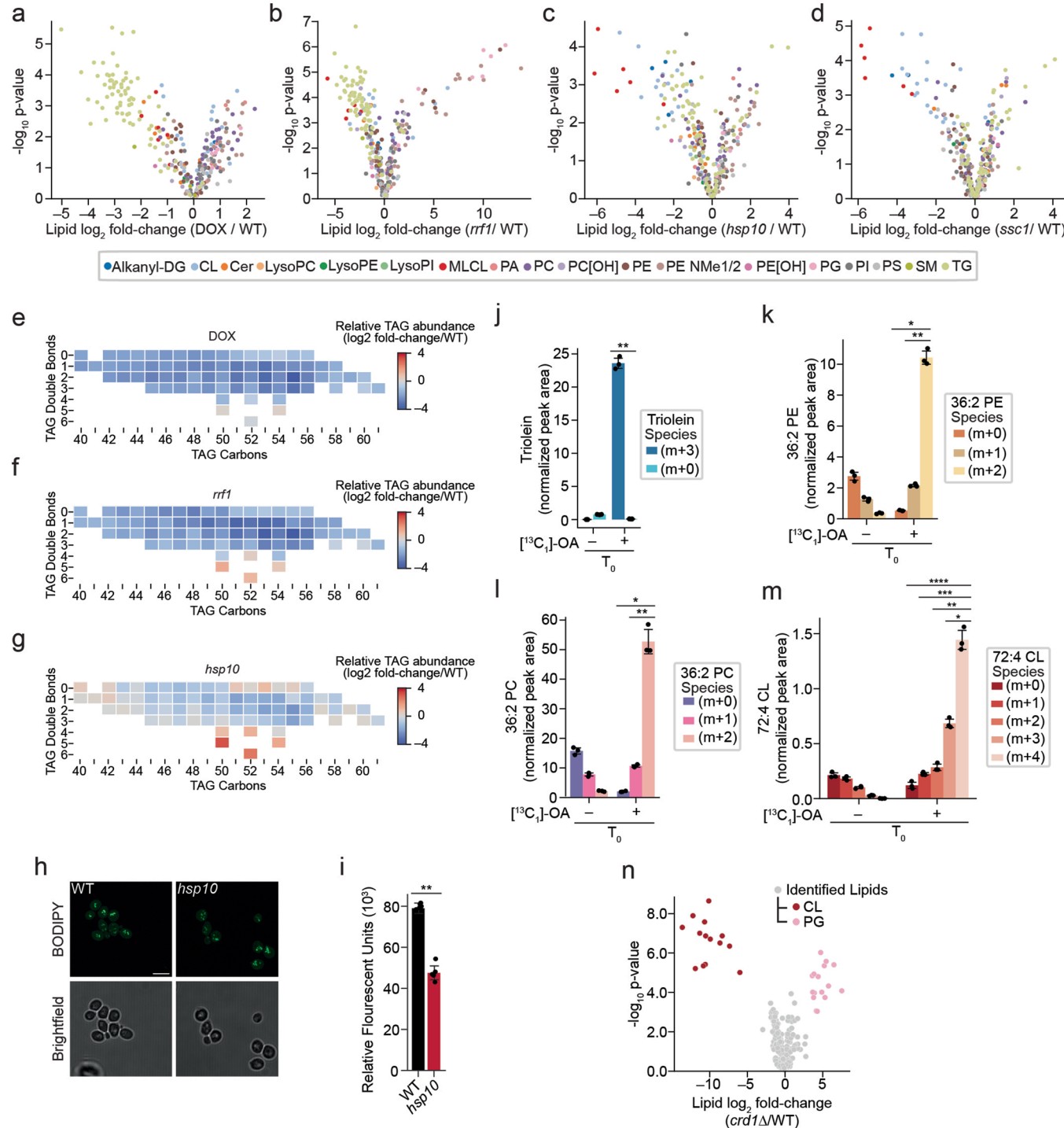

**Extended Data Fig. 2 | See next page for caption.**

**Extended Data Fig. 2 | Yeast strains that recover from mitochondrial stress mobilise TAG stores to facilitate cardiolipin biosynthesis. a-d**, Relative lipid abundance versus statistical significance for DOX-treated (**a**), *rrf1* (**b**), *hsp10* (**c**), and *ssc1* (**d**) yeast strains collected at the early time point. Abundance is given by the $\log_2$ transformed fold-change for each strain compared to the WT. Points are color-coded according to lipid class. (n = 3) **e-g**, Relative $\log_2$ fold change abundance of all quantified TAG species of DOX-treated (**e**), *rrf1* (**f**), or *hsp10* (**g**) yeast strains compared to WT. Saturation and length of TAG species is given by the number of double bonds or total number of carbons in the acyl-tails, respectively (n = 3). **h**, Representative confocal microscopy images of WT and *hsp10* BODIPY (green) stained yeast strains taken at 60X magnification. Cells were grown for 24 hours in YPG respiratory media. Scale bar represents 10 μm. **i**, Relative fluorescent signal from Nile red staining of *hsp10* and WT yeast strains. Cells were grown for 24 hours in YPG respiratory media (n = 6, **P = $1.40 \times 10^{-8}$, two-sided Student's t-test). **j**, Normalized relative abundance of light Triolein (light blue, m + 0) or heavy [$^{13}C_1$]-Triolein (dark blue) in WT cells (from Fig. 2f) pulsed with [$^{13}C_1$]-oleic acid for one hour (n = 3, ***P = $1.70 \times 10^{-6}$, two-sided Student's t-test). **k**, Normalized relative abundance of

light 36:2 Phosphatidylethanolamine (PE) (orange, m + 0), [$^{13}C_1$]-36:2 PE (tan, m + 1), or [$^{13}C_2$]-36:2 PE (beige, m + 2) in WT cells (from Fig. 2f) pulsed with [$^{13}C_1$]-oleic acid for one hour (n = 3, *P = $5.55 \times 10^{-6}$, **P = $1.12 \times 10^{-5}$, two-sided Student's t-test). **l**, Normalized relative abundance of light 36:2 Phosphatidylcholine (PC) (purple, m + 0), [$^{13}C_1$]-36:2 PC (pink, m + 1), or [$^{13}C_2$]-36:2 PC (peach, m + 2) in WT cells (from Fig. 2f) pulsed with [$^{13}C_1$]-oleic acid for one hour (n = 3, *P = $6.39 \times 10^{-5}$, **P = $1.35 \times 10^{-4}$, two-sided Student's t-test). **m**, Normalized relative abundance of light 72:4 Cardiolipin (CL) (dark red, m + 0), [$^{13}C_1$]-72:4 CL (red, m + 1), [$^{13}C_2$]-72:4 CL (orange, m + 2), [$^{13}C_3$]-72:4 CL (light orange, m + 3), or [$^{13}C_4$]-72:4 CL (light peach, m + 4) in WT cells (from Fig. 2f) pulsed with [$^{13}C_1$]-oleic acid for one hour (n = 3, *P = $3.39 \times 10^{-4}$, **P = $5.61 \times 10^{-5}$, ***P = $3.85 \times 10^{-5}$, ****P = $3.26 \times 10^{-5}$, two-sided Student's t-test). **n**, Relative lipid abundance versus statistical significance for *crd1Δ* versus WT yeast. Cells were grown for 24 hours in YPG respiratory media (n = 3). Quantified cardiolipin species are colored in red and phosphatidylglycerol species highlighted in pink. All other lipid species are colored in grey. For all experiments error bars are standard deviation, centre values represent the mean and n= number of independent biological replicates. Numerical data are available in Source data.

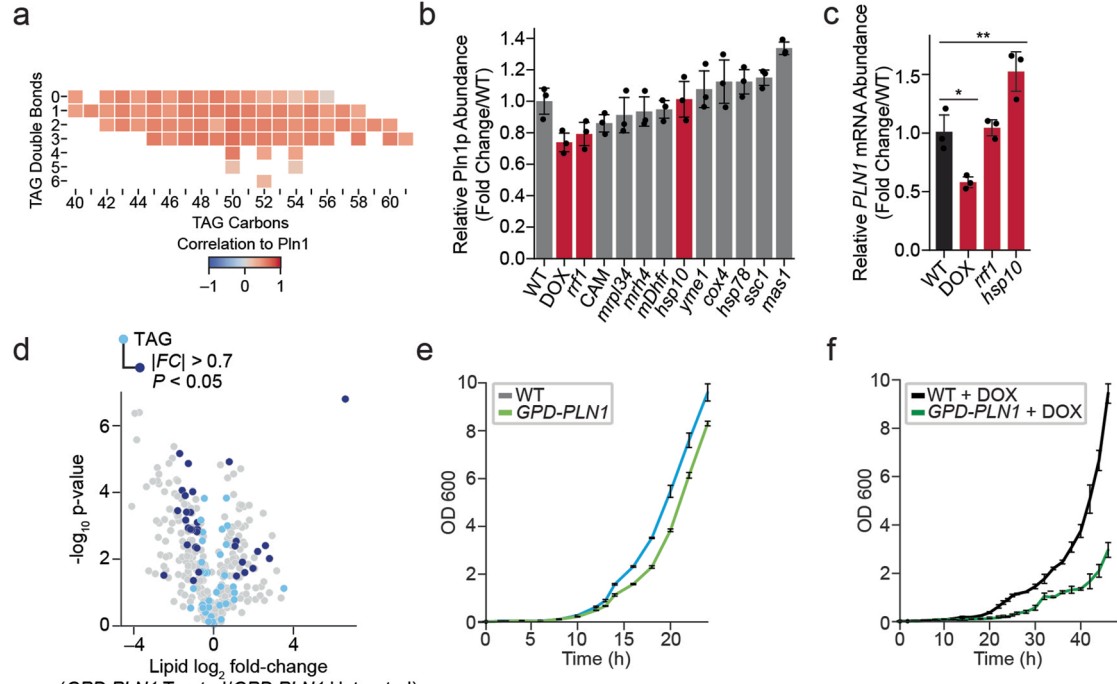

**Extended Data Fig. 3 | Pln1p has a role TAG mobilization and recovery from mitochondrial stress. a**, Spearman correlation of log$_2$ transformed Pln1p abundance and all quantified species of triacylglyceride (TAG) from multiomic screen. Saturation and length of TAG species is given by the number of double bonds or total number of carbons in the acyl-tails respectively. **b-c**, Relative protein abundance at the early time point of multiomic screen (**b**) or mRNA (**c**) abundance of Pln1p in WT and experimental yeast strains. Red bars indicate Group 3 strains. for **c**, cells were grown for 24 hours in YPG respiratory media (n = 3, *P = 0.03, **P = 0.02, two-sided Student's t-test). **d**, Relative lipid abundance versus statistical significance for *pln1Δ* yeast overexpressing GPD-*PLN1* treated with DOX or a DMSO control. Cells were

grown for 24 hours in SD (Ura-) respiratory media. Abundance is given as a log$_2$ transformed fold-change compared the vehicle control. Quantified TAG species are colored in light blue with TAG species that are significantly changed ( | FC | > 0.7, *p* < 0.05, two-sided Student's t-test) highlighted in dark blue. All other lipid species are colored in grey (n = 3). **e-f**, Full growth assay of WT expressing the *GPD-PLN1* gene as depicted in Fig. 3n. or an empty vector control. Cells were treated with either DMSO control (**e**) or DOX (**f**) (n = 3). Growth was determined in cells grown in YPG + G418 respiratory media over the course of 24 (**e**) to 48 (**f**) hours. For all experiments error bars are standard deviation, centre values represent the mean and n= number of independent biological replicates. Numerical data are available in Source data.

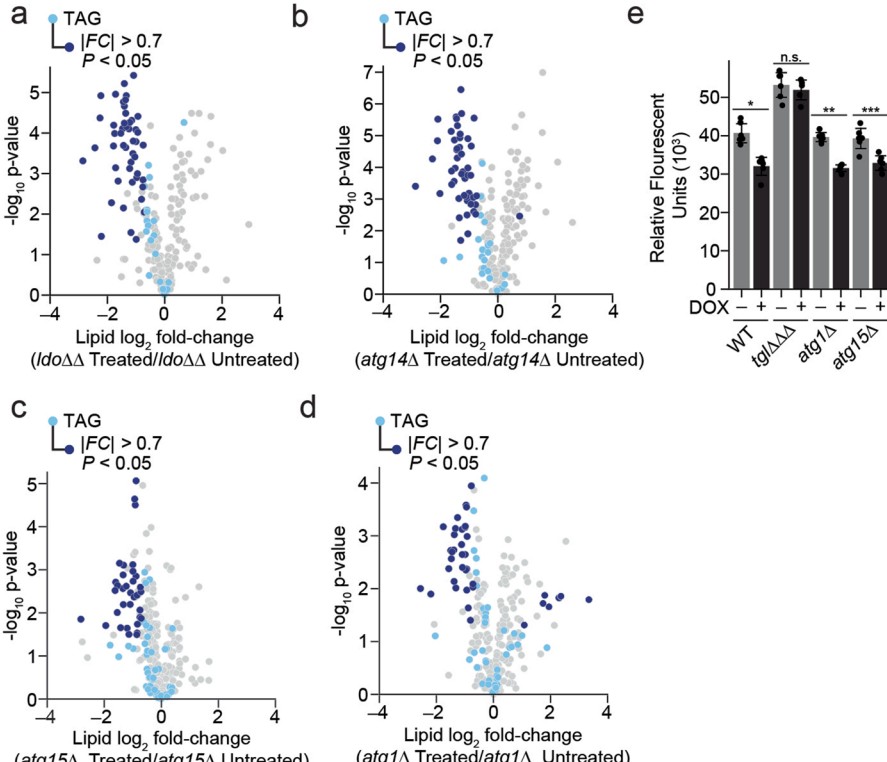

**Extended Data Fig. 4 | Triacylglyceride mobilisation during mitochondrial stress requires *tgl* lipases. a-d**, Relative lipid abundance versus statistical significance for *ldo45Δldo16Δ* (*ldoΔΔ*) **a**), *atg14Δ* (**b**), *atg15Δ* (**c**), or *atg1Δ* (**d**) yeast. Cells were grown for 24 hours in YPG respiratory media. Abundance is given as a log$_2$ transformed fold-change compared to the same strain treated with vehicle control. For (**a-d**), quantified TAG species are colored in light blue with TAG species that are significantly changed (|FC| > 0.7, $p$ < 0.05, two-sided Student's t-test) highlighted in dark blue. All other lipid species are colored in grey (n = 3). **e**, Relative fluorescent signal from Nile red staining of WT, *TGL* triple deletion (*tgl3Δtgl4Δtgl5Δ, tglΔΔΔ*), *atg1Δ*, and *atg15Δ* yeast (n = 6, *P = 2.20×10$^{-4}$, **P = 2.15×10$^{-7}$, ***P = 1.31×10$^{-3}$, n.s.= not significant, two-sided Student's t-test). Cells were grown for 24 hours in YPG respiratory media and were treated with either DOX or a DMSO vehicle control. For all experiments error bars are standard deviation, centre values represent the mean and n= number of independent biological replicates. Numerical data are available in Source data.

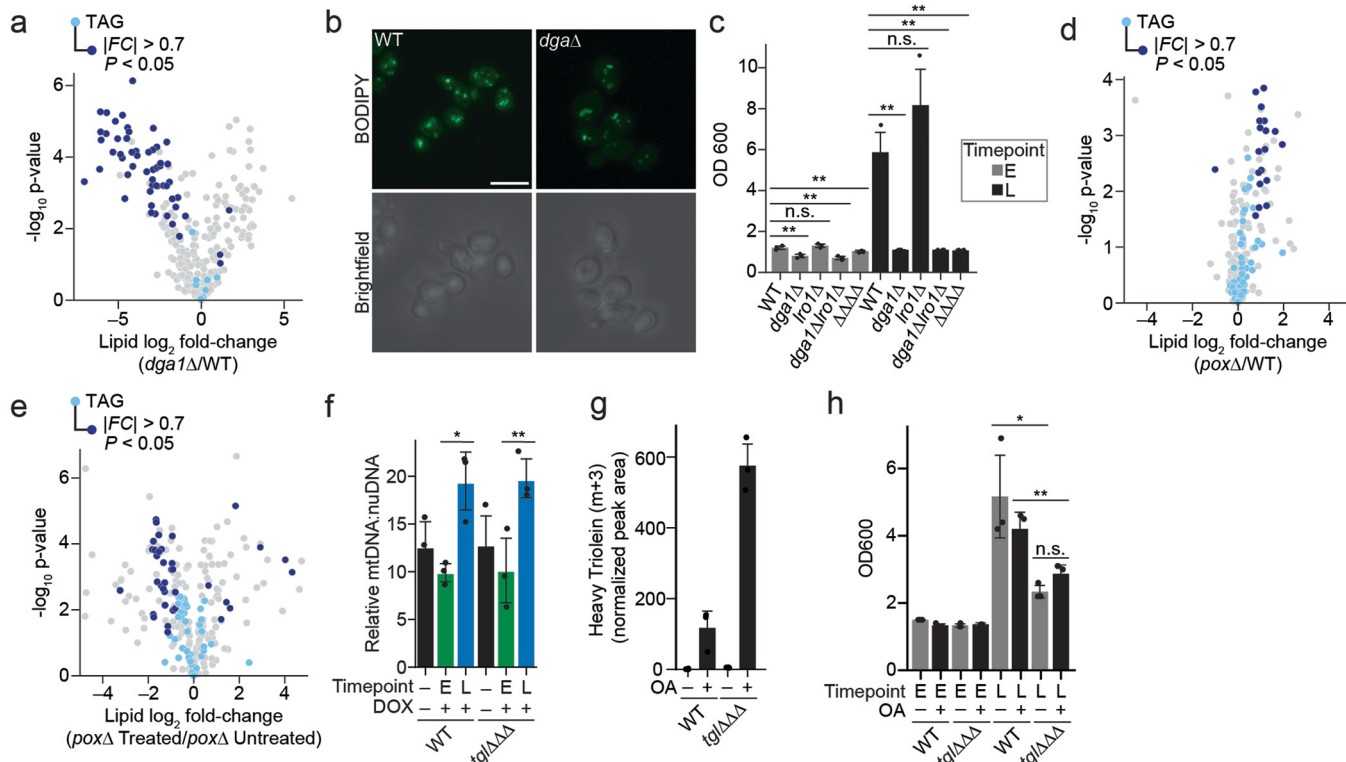

**Extended Data Fig. 5 | Tgl-dependent triacylglyceride mobilisation is essential for yeast to overcome mitochondrial stress. a**, Relative lipid abundance versus statistical significance for *dga1Δ* yeast. Cells were grown for 24 hours in YPG respiratory media. Abundance is given as a log₂ transformed fold-change compared to WT cells (n = 3). **b**, Representative confocal microscopy images of BODIPY (green) stained WT or *dga1Δ* yeast strains. Images were taken at 100X magnification. Scale bar represents 10 μm. **c**, Growth assay of WT, *dga1Δ, lro1Δ, dga1Δlro1Δ* double deletion, and *dga1Δlro1Δare1Δare2Δ* quadruple deletion (ΔΔΔΔ) yeast treated with DOX. Growth was determined after 24 hours (Time point E) or 48 hours (Time point L) in YPG respiratory media (n = 3, *P < 0.05, **P < 0.01, n.s.= not significant, two-sided Student's t-test). **d**, Relative lipid abundance versus statistical significance for *pox1Δ* yeast compared to a WT strain (**d**) or treated with DOX and compared to *pox1Δ* yeast treated with a vehicle (**e**). Cells were grown for 24 hours in YPG respiratory media (n = 3). For (**a**), (**c**), and (**d**) quantified TAG species are colored in light blue with TAG species that are significantly changed (|FC| > 0.7, p < 0.05, two-sided Student's t-test)

highlighted in dark blue. All other lipid species are colored in grey.
**f**, Relative abundance of mitochondrial DNA (mtDNA) as measured by a ratio of mtDNA to nuclear DNA (nuDNA) for WT or *TGL* triple deletion (*tgl3Δtgl4Δtgl5Δ, tglΔΔΔ*) yeast treated with DOX or vehicle control after 24 hours (Time point E) or 48 hours (Time point L). Cells were grown in YPG respiratory media (n = 3, *P = 0.01, **P = 0.03, two-sided Student's t-test). **g**, Normalized abundance of de-novo synthesized [¹³C₃]-Triolein (m + 3) in WT or *TGL* triple deletion (*tgl3Δtgl4Δtgl5Δ, tglΔΔΔ*) after 1 hr treatment with 0.0002% [¹³C₁]-oleic acid or vehicle (n = 3). Abundances were normalized to CoQ₈ internal standard.
**h**, Growth assay of WT or *TGL* triple deletion (*tgl3Δtgl4Δtgl5Δ, tglΔΔΔ*) yeast treated with 0.0002% [¹³C₁]-oleic acid or vehicle. Growth was determined after 24 hours (Time point E) or 48 hours (Time point L) in YPG respiratory media, (n = 3, *P = 0.03, **P = 0.02, n.s.= not significant, two-sided Student's t-test). For all experiments error bars are standard deviation, centre values represent the mean and n= number of independent biological replicates. Numerical data are available in Source data.

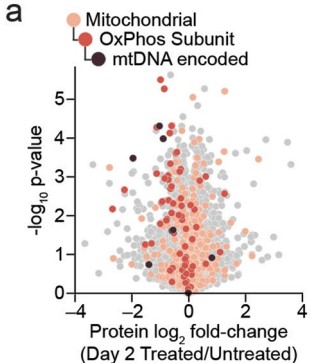

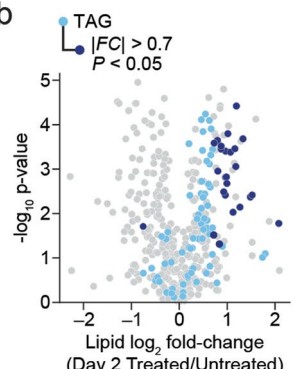

**Extended Data Fig. 6 | Triacylglyceride mobilisation is required for mammalian cells to overcome mitochondrial stress. a**, Relative protein abundances for WT HAP1 cells grown in galactose media treated with 10 μM DOX collected on Day 2 versus statistical significance. Abundance is given by a log$_2$ transformed fold-change compared to the vehicle treated control with non-mitochondrial proteins (grey), mitochondrial proteins (light orange), OxPhos (dark orange) and mtDNA encoded proteins (dark brown) highlighted by colour. (n = 3) **b**, Relative lipid abundance versus statistical significance for WT

HAP1 cells grown in galactose, treated with 10 μM DOX DOX for 2 days compared to a vehicle treated control. Quantified TAG species are colored in light blue with TAG species that are significantly increased (FC > 0.7, $p$ < 0.05, two-sided Student's t-test) or decreased (FC < −0.7, $p$ < 0.05, two-sided Student's t-test) highlighted in dark blue. All other lipid species are colored in grey (n = 3). For all experiments, n= number of independent biological replicates. Numerical data are available in Source data.

# Reporting Summary

## Statistics

For all statistical analyses, confirm that the following items are present in the figure legend, table legend, main text, or Methods section.

| n/a | Confirmed | |
|---|---|---|
| ☐ | ☒ | The exact sample size (*n*) for each experimental group/condition, given as a discrete number and unit of measurement |
| ☐ | ☒ | A statement on whether measurements were taken from distinct samples or whether the same sample was measured repeatedly |
| ☐ | ☒ | The statistical test(s) used AND whether they are one- or two-sided *Only common tests should be described solely by name; describe more complex techniques in the Methods section.* |
| ☒ | ☐ | A description of all covariates tested |
| ☒ | ☐ | A description of any assumptions or corrections, such as tests of normality and adjustment for multiple comparisons |
| ☐ | ☒ | A full description of the statistical parameters including central tendency (e.g. means) or other basic estimates (e.g. regression coefficient) AND variation (e.g. standard deviation) or associated estimates of uncertainty (e.g. confidence intervals) |
| ☐ | ☒ | For null hypothesis testing, the test statistic (e.g. *F*, *t*, *r*) with confidence intervals, effect sizes, degrees of freedom and *P* value noted *Give P values as exact values whenever suitable.* |
| ☒ | ☐ | For Bayesian analysis, information on the choice of priors and Markov chain Monte Carlo settings |
| ☒ | ☐ | For hierarchical and complex designs, identification of the appropriate level for tests and full reporting of outcomes |
| ☒ | ☐ | Estimates of effect sizes (e.g. Cohen's *d*, Pearson's *r*), indicating how they were calculated |

*Our web collection on statistics for biologists contains articles on many of the points above.*

## Software and code

Policy information about availability of computer code

| | |
|---|---|
| Data collection | Quantitative PCR data was collected using QuantStudio Real-Time PCR software v1.2 (Applied Biosciences). Growth assay, OCR, Florescent data was collected using Gen5 v3.02.2 (BioTek). GO term enrichments were determined using ShinyGO v0.80 (http://bioinformatics.sdstate.edu/go/) Florescent Microscopy images were taken using Zenn v.3.6 (Zeiss) Volume density imaging was acquired using Elements v5.21 (Nikon) |
| Data analysis | LC-MS targeted lipidomics was analyzed by Tracefinder 5.1 (Thermo) LC-MS untargeted lipidomics was analyzed by Lipidex v1.0 LC_MS proteomic files were analyzed using either MaxQuant v1.5.5.5 or Proteome Discoverer v2.5 (Thermo) LC-MS Metabolomics was analyzed using Compound Discoverer (v3.1 & v3.3) (Thermo) Florescent images were analyzed using ImageJ software v2.9.01.53t Volume density imaging was analyzed using Ilastik (v.1.3.3) and YeaZ (v1.0.3) Statistical analysis and all graphing was performed using Python v3.9 Cell and organelle analysis was performed using custom code (Github: https://github.com/alinearra/CellandOrganelleAnalysis) |

For manuscripts utilizing custom algorithms or software that are central to the research but not yet described in published literature, software must be made available to editors and reviewers. We strongly encourage code deposition in a community repository (e.g. GitHub). See the Nature Portfolio guidelines for submitting code & software for further information.

## Data

Policy information about <u>availability of data</u>

All manuscripts must include a <u>data availability statement</u>. This statement should provide the following information, where applicable:

- Accession codes, unique identifiers, or web links for publicly available datasets
- A description of any restrictions on data availability
- For clinical datasets or third party data, please ensure that the statement adheres to our <u>policy</u>

> All mass spectrometry data (proteomics, lipidomics, and metabolomics) have been deposited in Massive with the primary accession codes MSV000092267 (multiomic screen) and MSV000095028 (follow up experiments). Source data, including all mass spectrometry data tables, have been provided in Source Data. The following databases were used in the searching of mass spectrometry files (Biocyc (https://www.biocyc.org/), Human Metabolome Database (https://hmdb.ca/), KEGG (https://www.genome.jp/kegg/), mzCloud (https://www.mzcloud.org/), MassBank (https://massbank.eu/MassBank/), MitoCarta (https://personal.broadinstitute.org/scalvo/MitoCarta3.0/human.mitocarta3.0.html), Uniprot (https://www.uniprot.org/)). All other data supporting the findings of this study are available from the corresponding author on reasonable request.

## Research involving human participants, their data, or biological material

Policy information about studies with <u>human participants or human data</u>. See also policy information about <u>sex, gender (identity/presentation), and sexual orientation</u> and <u>race, ethnicity and racism</u>.

| Reporting on sex and gender | N/A |
|---|---|
| Reporting on race, ethnicity, or other socially relevant groupings | N/A |
| Population characteristics | N/A |
| Recruitment | N/A |
| Ethics oversight | N/A |

Note that full information on the approval of the study protocol must also be provided in the manuscript.

# Field-specific reporting

Please select the one below that is the best fit for your research. If you are not sure, read the appropriate sections before making your selection.

☒ Life sciences      ☐ Behavioural & social sciences      ☐ Ecological, evolutionary & environmental sciences

For a reference copy of the document with all sections, see nature.com/documents/nr-reporting-summary-flat.pdf

# Life sciences study design

All studies must disclose on these points even when the disclosure is negative.

| Sample size | No statistical methods were used to predetermine sample size. All experiments were performed in at least biological triplicate. Samples sizes were chosen based on the generally accepted standard for the minimum number of replicates needed to obtain conclusive evidence for these types of experiments reported in previous publications. |
|---|---|
| Data exclusions | No data were excluded from these analyses. |
| Replication | All attempts at experimental replication were successful. All experiments were performed in at least biological triplicate, as indicated in the figure legends. |
| Randomization | Randomization was used for mass spectrometry injection to minimize batch effects. No other randomization was used for experimental groups as all other measurements were quantitatively made by instruments or made using computational analyses as is standard and previously reported. |
| Blinding | Blinding of experimental groups was not relevant as experimental measurements were generated by automated measurements or computational analyses. |

# Reporting for specific materials, systems and methods

We require information from authors about some types of materials, experimental systems and methods used in many studies. Here, indicate whether each material, system or method listed is relevant to your study. If you are not sure if a list item applies to your research, read the appropriate section before selecting a response.

## Materials & experimental systems

| n/a | Involved in the study |
|-----|----------------------|
| ☒ | Antibodies |
| ☐ | ☒ Eukaryotic cell lines |
| ☒ | Palaeontology and archaeology |
| ☒ | Animals and other organisms |
| ☒ | Clinical data |
| ☒ | Dual use research of concern |
| ☒ | Plants |

## Methods

| n/a | Involved in the study |
|-----|----------------------|
| ☒ | ChIP-seq |
| ☒ | Flow cytometry |
| ☒ | MRI-based neuroimaging |

## Eukaryotic cell lines

Policy information about cell lines and Sex and Gender in Research

| | |
|---|---|
| Cell line source(s) | HAP1 wild type - Horizon Discovery C631<br>HAP1 CPT2 KO - Horizon Discovery HZGHC003795c011<br>HAP1 ATGL KO - Horizon Discovery HZGHC004666c014 |
| Authentication | HAP1 cell lines were authenticated by Horizon Discovery using PCR amplification and Sanger sequencing. |
| Mycoplasma contamination | All cell lines were negative for mycoplasma contamination as tested using a commercial test kit. |
| Commonly misidentified lines<br>(See ICLAC register) | No commonly misidentified cell lines were used in this study. |

## Plants

| | |
|---|---|
| Seed stocks | N/A |
| Novel plant genotypes | N/A |
| Authentication | N/A |

