## [Peer Review File · Nature Cell Biology]

Triacylglycerol mobilization underpins mitochondrial stress recovery

Corresponding Author: Professor David Pagliarini

Version 0:

Decision Letter:

*Please delete the link to your author homepage if you wish to forward this email to co-authors.

Dear Professor Pagliarini,

Thank you again for submitting your manuscript, "Triacylglycerol mobilization underpins mitochondrial stress recovery", to the journal; it has now been seen by 3 referees, who are experts in lipid cell biology, lipid droplets (Referee #1); lipidomics, metabolomics (Referee #2); and mitochondria stress responses (Referee #3). As you will see from their comments (attached below), they found the work of potential interest but have raised substantial concerns that in our view would need to be addressed thoroughly before we can consider publication in Nature Cell Biology.

As per our standard editorial process, we have now discussed the referees' comments to identify key referee points that should be addressed with priority. To guide the scope of the revisions, I have listed these points below. As you know, our standard revision period is six months and we're happy to discuss if you have any questions or concerns.

In our view, the reviewers brought up important concerns and identified conclusions that need strengthening and clarifying. We recommend dedicating efforts in revision to address the reviews as follows:

1- The reviewers asked for additional tests of the importance of TAG as lipid species for the survival stress response and of the mitochondrial phenotypes of stress and recovery:

Rev#1 points #1, #3

Rev#2 "It appears that overall TG species were mobilized during stress and under all studied conditions, but was there also a shift of fatty acyl chain length or total degrees of unsaturation?"

Rev#3 "Fig 1h might suggest that the *ssc1* mutant has no or very low amounts of mtDNA. It can explain why this strain could not recover and grow on glycerol. This should be tested.

This result also raises an important question – why this mutant does not affect TAGs. I am not sure this question was attempted to be addressed and it should be at least referred to in the discussion. Is it a direct effect of mtDNA loss or maybe just a complete block of respiration and inability to make enough ATP to execute a response?"

Fig 1h- The mutants seem to have more mtDNA. This result seems to be in line with the authors' hypothesis that the mutants have higher rates of mitochondrial biogenesis and therefore higher mitochondrial mass. This should be referred to in the manuscript. The authors could also test the mitochondrial mass in at least some of the mutants in addition to DOX treatment."

"Fig 5 e-f – This is a very important result that positions TAG catabolism upstream of mitochondrial gene upregulation. The authors should test whether TAG catabolism is also necessary for mtDNA replication using the same strains and the assay used in fig 1h."

2- Rev#3 felt that the Pln1 data needed to be explored further:

Rev#3 "Fig 3 h-k- I am not sure the authors can conclude that Pln1 is important for TAG mobilization without direct evidence especially since it was previously shown to mediate TAG biogenesis. Another interpretation of the results is that Pln1 overexpression increases TAG biogenesis and changes the balance of stored versus free TAG (less TAG is available for CL biosynthesis). Pln1 deletion might have no impact due to compensation by other factors."

"Fig 3h- Are the transcription levels of Pln1 different between wt and DOX/group 3 mutants?"

3- Rev#1 in point #4 asked for deeper analyses of the mechanism by lipids are derived from LDs, which we agree are needed to support the current claims.

4- Please also strengthen the analyses showing the importance of cardiolipin as per Rev#1 point #2.

5- All other referee concerns pertaining to technical concerns (including about the mass spectrometry approaches and analyses, as per Rev#2), strengthening existing data, providing controls, methodological details, clarifications and textual changes, should also be addressed.

6- Finally, please pay close attention to our guidelines on statistical and methodological reporting (listed below) as failure to do so may delay the reconsideration of the revised manuscript. In particular, please provide:

We would be happy to consider a revised manuscript that would satisfactorily address these points, unless a similar paper is published elsewhere, or is accepted for publication in Nature Cell Biology in the meantime.

- ensure that it conforms to our format instructions and publication policies (see below and <https://www.nature.com/nature/for-authors>).

- provide a point-by-point rebuttal to the full referee reports verbatim, as provided at the end of this letter.

- provide the completed Reporting Summary (found here <https://www.nature.com/documents/nr-reporting-summary.pdf>). This is essential for reconsideration of the manuscript will be available to editors and referees in the event of peer review. For more information see <http://www.nature.com/authors/policies/availability.html> or contact me.

Nature Cell Biology is committed to improving transparency in authorship. As part of our efforts in this direction, we are now requesting that all authors identified as 'corresponding author' on published papers create and link their Open Researcher and Contributor Identifier (ORCID) with their account on the Manuscript Tracking System (MTS), prior to acceptance. ORCID helps the scientific community achieve unambiguous attribution of all scholarly contributions. You can create and link your ORCID from the home page of the MTS by clicking on 'Modify my Springer Nature account'. For more information please visit please visit www.springernature.com/orcid.

This journal strongly supports public availability of data. Please place the data used in your paper into a public data repository, or alternatively, present the data as Supplementary Information. If data can only be shared on request, please explain why in your Data Availability Statement, and also in the correspondence with your editor. Please note that for some data types, deposition in a public repository is mandatory - more information on our data deposition policies and available repositories appears below.

Link Redacted

*This url links to your confidential home page and associated information about manuscripts you may have submitted or be

reviewing for us. If you wish to forward this email to co-authors, please delete the link to your homepage.

We hope that you will find our referees' comments and editorial guidance helpful. Please do not hesitate to contact me if there is anything you would like to discuss. Thank you again for considering the journal for your work.

Best wishes,

Melina

Melina Casadio, PhD
Senior Editor, Nature Cell Biology
ORCID ID: <https://orcid.org/0000-0003-2389-2243>

Reviewers' Comments:

Reviewer #1:

Remarks to the Author:

This is an interesting study dissecting how budding yeast mount responses to mitochondrial stress. Omics based screening of multiple yeast strains with defects in mitochondrial homeostasis identified triglyceride synthesis as an important pathway for stress response. TG mobilization via Tgl lipases was found to be important for mitochondrial stress response. This lipid mobilization is proposed to fuel mitochondrial biogenesis (specifically cardiolipin synthesis), rather than other lipids or bioenergetics at peroxisomes, to enable adaptation and survival.

Overall this is an interesting and informative study that links triglyceride stores and lipid droplets to mitochondrial stress adaptation. The study nicely combines large-scale omics screening with pathway analysis and functional outputs. The implication that lipids mobilized from TAG are channeled into cardiolipin synthesis is intriguing but requires additional experiments to more fully test. There are also additional control experiments needed to allow some of the conclusions to be stated.

General concerns:

1) A key question is whether TAG is a necessary lipid species for the DOX survival response. Would providing exogenous fatty acids directly (without their storage as TAG) functionally replace TG stores? For example, does feeding the tgl-triple deletion yeast strain oleic acid during DOX treatment rescue similarly to yeast that have TGIs?

2) Cardiolipin is proposed to be the major mitochondrial lipid species that TAG derived lipids are channeled into. Further investigation of how loss of Crd1p impacts DOX and mito stress survival would strengthen this claim.

3) The TAG synthesis enzyme Dga1p was found to be important for yeast to survive DOX treatment (ED, Fig 5). However there are two TAG synthesis enzymes in yeast, Dga1 and Lro1p. Is Lro1p similarly as important as Dga1p? What occurs in the double dga1/lro1 KO, or in yeast that lack all four neutral lipid synthesis enzymes (dga1/lro1/are1/are2) that cannot form droplets at all?

4) Tgl lipases are proposed to be the major TAG mobilization pathway versus micro-lipophagy at the yeast vacuole, which was blocked by Atg1p and Atg15p deletion. Both these Atg deletion may have pleiotropic effects since Atg1p also governs general macroautophagy, and Atg15p is a vacuolar lipase and its loss may perturb general vacuole function. To further investigate micro-lipophagy in this pathway, depleting factors more specifically linked to micro-lipophagy like Atg14p, Vac8p, or the Ldo16/45p proteins would be necessary.

Reviewer #2:

Remarks to the Author:

This is a well-written manuscript describing triglyceride changes under mitochondrial stress in yeast.

It appears that overall TG species were mobilized during stress and under all studied conditions, but was there also a shift of fatty acyl chain length or total degrees of unsaturation?

The S-lens conditions are very high for the lipidomic analysis. An s-lens of 90 will cause significant source fragmentation which could cause misidentification of lipids. This could be observed across many lipid classes. Was this presence of source fragments accounted for? Were ergosterol esters identified in the lipids? This is a class of lipids that would be highly altered based on the source conditions and the ester group could easily fragment. There is some research on the role of ergosterol esters in stress response, so identification of them could be important, however this stress response could be more heat derived and may be out of the scope of this work.

It is strange that lipids and proteomic were conducted with 2 different methods each. Was there a reason for the change in

methods? What was the overall correlation? Did the methods find the same number of lipids and proteins?

Please add to the text the level of identification confidence based on the metabolomics standard initiative. Metabolites were identified by MS/MS primarily since authentic standards were not analyzed. There was very little discussion of any metabolite changes which is a bit surprising since metabolites should be altered under stress especially ATP and other nucleotide phosphates. It might be that the extraction method did not retain this class of compounds.

There are a few errors in the methods where the superscript was not added correctly for AGC target values.

Reviewer #3:

Remarks to the Author:

In their manuscript entitled "Triacylglycerol mobilization underpins mitochondrial stress recovery" Baker et al. investigated mechanisms that aid recovery from mitochondrial respiration stress. The authors used a multiomic approach to compare the response and recovery of various yeast strains and conditions that disturb mitochondrial functions. They have identified few mutants/drugs that exhibit a growth defect when grown in non-fermentable carbon source. These mutants/drugs could not undergo diauxic shift in the same timeframe as wild type but could eventually recover and grow. This study explores the cellular requirements for the recovery of these respiratory defective mutants. The work demonstrated that TAG catabolism and CL biosynthesis are key to the respiration and growth recovery of the mutants. Finally, the authors provide evidence that a similar mechanism exist in mammals.

This is a very interesting study that includes an extensive amount of data and broad range of methods to support the hypotheses. It is highly relevant to mitochondrial diseases with decreased oxphos capacity. The quality of the data is excellent and the analyses are very convincing. The manuscript is well-written, but the story is difficult to follow in some of the sections and is often missing details in the main text and figure legends. The take-home messages are novel but they are not always clearly delivered. For example, not enough emphasis is placed on the important message that TAG conversion to CL is required for mitochondrial protein upregulation and therefore for all aspects of mitochondrial biogenesis. Overall, this study will be of interest for the broad scientific community if the following comments could be addressed.

Comments:

"To investigate how *S. cerevisiae* respond to and recover from certain mitochondrial stress"- I am not sure how accurate this statement is. It might be easier to the reader if this is presented in a more straightforward way with the question of how cells compensate for suboptimal mitochondrial oxphos capacity. Since the study focuses on the ability to adapt to conditions in which mitochondrial respiration is essential, I think it will be beneficial for the reader to present the question as such. Similarly- "Collectively, these data demonstrate that Group 3 strains mount a successful cellular response to mitochondrial stress distinct from the currently appreciated response pathways." I agree with this statement but this is too early to conclude this before showing the rest of the data. It can still be that these strains are identical to wild type but just slow to react.

Fig 1h might suggest that the *ssc1* mutant has no or very low amounts of mtDNA. It can explain why this strain could not recover and grow on glycerol. This should be tested.

This result also raises an important question – why this mutant does not affect TAGs. I am not sure this question was attempted to be addressed and it should be at least referred to in the discussion. Is it a direct effect of mtDNA loss or maybe just a complete block of respiration and inability to make enough ATP to execute a response?

Fig 1h- The mutants seem to have more mtDNA. This result seems to be in line with the authors' hypothesis that the mutants have higher rates of mitochondrial biogenesis and therefore higher mitochondrial mass. This should be referred to in the manuscript. The authors could also test the mitochondrial mass in at least some of the mutants in addition to DOX treatment.

Line 118- "Moreover, expression of defined retrograde stress response pathway markers that may aid recovery were not induced in any of these strains (Extended Data Fig. 1m,n), perhaps due to our selected parental yeast strain (W303)" What were the growth conditions in this analysis? Early or late collections? What does it mean "due to our selected parental yeast strain"?

How do the authors explain the low levels of *cit2* in the mutants compared to wt?

Fig 2- Did the lipid analysis include phosphatic acid (PA)? If yes it would be helpful to add the results given that PA is a precursor of CL.

Fig 2B- The authors should include *hsp10* BODIPY imaging in the figure.

Fig 3 h-k- I am not sure the authors can conclude that *Pln1* is important for TAG mobilization without direct evidence especially since it was previously shown to mediate TAG biogenesis. Another interpretation of the results is that *Pln1* overexpression increases TAG biogenesis and changes the balance of stored versus free TAG (less TAG is available for CL biosynthesis). *Pln1* deletion might have no impact due to compensation by other factors.

Fig 3k – please add growth curves for this experiment

Fig 3h- Are the transcription levels of *Pln1* different between wt and DOX/group 3 mutants?

Fig 5 e-f – This is a very important result that positions TAG catabolism upstream of mitochondrial gene upregulation. The authors should test whether TAG catabolism is also necessary for mtDNA replication using the same strains and the assay used in fig 1h.

Fig 6g and h- I assume that galactose medium was used here? It would be helpful to add the parental wild type strain to this graph (in the presence and absence of DOX) to verify that the untreated KO mutants do not have a growth defect. If there is a defect the authors should confirm that the combination of mutants with DOX is not additive.

Minor comments:

It would be helpful to add references for the mutants used in this study. For example, to provide context to the claim that constitutive overexpression of *cox4* and *mDhfr* disrupts mitochondrial proteostasis.

Fig. 1 and S1 present data from 2 strains that are not mentioned in the text: *hsp78* and *yme1*

The figures and figure legends are often missing details that could help evaluate and interpret the data and the conclusion. Moreover, some small details are missing. A few examples:

Fig 1B- What is WTV?

Fig 1B and C- it would be helpful to add a color key to the legend

Fig 1 legend –

“Mitochondrial proteins that are significantly changed ($|FC| > 0.7$, $p < 0.05$) (g) are highlighted in black.” Why is this statement relevant only to g? black labels are depicted in fig f too.

Also, in g and f- the color key in the graph is unclear. What is “L” from the left of the black circle? This is relevant to other figures.

Please specify the conditions used in each experiment. For example, what were the conditions in figure 3i? how long were the cells grown in glycerol, how long was the DOX treatment for?

Fig 2g- the authors should discuss the decrease in PE and PC

Fig 3J and k- typo in the legend, replace f and g with j and k

Lines 206- Isn't *dga1* included in Fig 5a as well?

Fig 5f –f is missing from the figure.

Sup table 2- It would be helpful to include gene names in the table.

ABSTRACT AND MAIN TEXT – please follow the guidelines that are specific to the format of your manuscript, as listed in

our Guide to Authors (http://www.nature.com/ncb/pdf/ncb_gta.pdf) Briefly, Nature Cell Biology Articles, Resources and Technical Reports have 3500 words, including a 150 word abstract, and the main text is subdivided in Introduction, Results, and Discussion sections. Nature Cell Biology Letters have up to 2500 words, including a 180 word introductory paragraph (abstract), and the text is not subdivided in sections.

Methods should be written concisely, but should contain all elements necessary to allow interpretation and replication of the results. As a guideline, Methods sections typically do not exceed 3,000 words. The Methods should be divided into subsections listing reagents and techniques. When citing previous methods, accurate references should be provided and any alterations should be noted. Information must be provided about: antibody dilutions, company names, catalogue numbers and clone numbers for monoclonal antibodies; sequences of RNAi and cDNA probes/primers or company names and catalogue numbers if reagents are commercial; cell line names, sources and information on cell line identity and authentication. Animal studies and experiments involving human subjects must be reported in detail, identifying the committees approving the protocols. For studies involving human subjects/samples, a statement must be included confirming that informed consent was obtained. Statistical analyses and information on the reproducibility of experimental results should be provided in a section titled "Statistics and Reproducibility".

All Nature Cell Biology manuscripts submitted on or after March 21 2016 must include a Data availability statement as a separate section after Methods but before references, under the heading "Data Availability". For Springer Nature policies on data availability see <http://www.nature.com/authors/policies/availability.html>; for more information on this particular policy see <http://www.nature.com/authors/policies/data/data-availability-statements-data-citations.pdf>. The Data availability statement should include:

- Accession codes for primary datasets (generated during the study under consideration and designated as "primary accessions") and secondary datasets (published datasets reanalysed during the study under consideration, designated as "referenced accessions"). For primary accessions data should be made public to coincide with publication of the manuscript. A list of data types for which submission to community-endorsed public repositories is mandated (including sequence, structure, microarray, deep sequencing data) can be found here <http://www.nature.com/authors/policies/availability.html#data>.
- Unique identifiers (accession codes, DOIs or other unique persistent identifier) and hyperlinks for datasets deposited in an approved repository, but for which data deposition is not mandated (see here for details <http://www.nature.com/sdata/data-policies/repositories>).
- At a minimum, please include a statement confirming that all relevant data are available from the authors, and/or are included with the manuscript (e.g. as source data or supplementary information), listing which data are included (e.g. by figure panels and data types) and mentioning any restrictions on availability.
- If a dataset has a Digital Object Identifier (DOI) as its unique identifier, we strongly encourage including this in the Reference list and citing the dataset in the Methods.

We recommend that you upload the step-by-step protocols used in this manuscript to protocols.io. More details can be found at <https://www.protocols.io/help/publish-articles>.

All imaging data should be accompanied by scale bars, which should be defined in the legend.

Cropped images of gels/blots are acceptable, but need to be accompanied by size markers, and to retain visible background signal within the linear range (i.e. should not be saturated). The boundaries of panels with low background have to be demarked with black lines. Splicing of panels should only be considered if unavoidable, and must be clearly marked on the figure, and noted in the legend with a statement on whether the samples were obtained and processed simultaneously. Quantitative comparisons between samples on different gels/blots are discouraged; if this is unavoidable, it should only be performed for samples derived from the same experiment with gels/blots were processed in parallel, which needs to be stated in the legend.

The total number of Supplementary Figures (not including the “unprocessed scans” Supplementary Figure) should not exceed the number of main display items (figures and/or tables (see our Guide to Authors and March 2012 editorial <http://www.nature.com/ncb/authors/submit/index.html#supinfo>; <http://www.nature.com/ncb/journal/v14/n3/index.html#ed>). No restrictions apply to Supplementary Tables or Videos, but we advise authors to be selective in including supplemental data.

GUIDELINES FOR EXPERIMENTAL AND STATISTICAL REPORTING

REPORTING REQUIREMENTS – We are trying to improve the quality of methods and statistics reporting in our papers. To that end, we are now asking authors to complete a reporting summary that collects information on experimental design and reagents. The Reporting Summary can be found here <https://www.nature.com/documents/nr-reporting-summary.pdf>. If you would like to reference the guidance text as you complete the template, please access these flattened versions at <http://www.nature.com/authors/policies/availability.html>.

We strongly recommend the presentation of source data for graphical and statistical analyses as a separate Supplementary Table, and request that source data for all independent repeats are provided when representative experiments of multiple independent repeats, or averages of two independent experiments are presented. This supplementary table should be in Excel format, with data for different figures provided as different sheets within a single Excel file. It should be labelled and numbered as one of the supplementary tables, titled “Statistics Source Data”, and mentioned in all relevant figure legends.

Version 1:

Decision Letter:

Our ref: NCB-A54496A

11th November 2024

Dear Dr. Pagliarini,

Thank you for submitting your revised manuscript "Triacylglycerol mobilization underpins mitochondrial stress recovery" (NCB-A54496A) and for your patience with the re-review process. Your revision has now been seen by the original referees and their comments are below. The reviewers find that the paper has been strengthened in revision, and therefore we'll be happy in principle to publish it in Nature Cell Biology, pending minor revisions to comply with our editorial and formatting

guidelines.

Please note that the current version of your manuscript is in a PDF format; could you please email us a copy of the file in an editable format (Microsoft Word or LaTeX) as we can not proceed with PDFs at this stage? Many thanks in advance.

Once we have the Word file, we will begin performing detailed checks on your paper and will send you a checklist detailing our editorial and formatting requirements in about 1-2 weeks. Please do not upload the final materials and make any revisions until you receive this additional information from us.

Thank you again for your interest in Nature Cell Biology. Please do not hesitate to contact me if you have any questions.

Sincerely,

Melina

Melina Casadio, PhD
Senior Editor, Nature Cell Biology
Consulting Editor, Nature Structural & Molecular Biology
ORCID ID: <https://orcid.org/0000-0003-2389-2243>

Reviewer #1 (Remarks to the Author):

The revised study has added a significant amount of new data that address the major concerns of the previous review round. A significant question was whether addition of exogenous fatty acids could functionally replace loss of TAG lipolysis. This has been investigated with ¹³C-oleic acid pulse chase experiments, showing that free oleic acid does not functionally replace TAG stores in yeast. Whether lipids from TAG were channeled into cardiolipin was also further investigated. It was observed that CRD1-KO yeast did not mobilize TAG stores following dox treatment, suggesting yeast somehow sense cardiolipin (and/or mito homeostasis) to coordinate TAG mobilization. The differential dependence on Dga1 and Lro1 was also examined, showing that their loss impact mitochondrial stress differentially (with Dga1 loss having the larger effect in the growth conditions shown).

Overall the revised manuscript displays significant new data, new experiments refine the working model, and is it overall an important study for dissemination to the lipid metabolism community.

Reviewer #2 (Remarks to the Author):

The authors have done a very good job in responding to key comments from the initial review. The discussion regarding different methods and the correlation analysis helped to confirm method concordance.

This reviewer does not have additional comments

Reviewer #3 (Remarks to the Author):

In their revised manuscript, Baker et al. addressed my criticism and provide additional data to support their conclusions. I appreciate the extensive work and believe that it further strengthens the proposed model. This work provides valuable insights into mechanisms that enable mitochondrial recovery from damage, a topic of significant interest to the scientific community. I therefore support the publication of this manuscript in Nature Cell Biology.

Version 2:

Decision Letter:

Dear Dr Pagliarini,

I am pleased to inform you that your manuscript, "Triacylglycerol mobilization underpins mitochondrial stress recovery", has now been accepted for publication in Nature Cell Biology.

Please note that *Nature Cell Biology* is a Transformative Journal (TJ). Authors may publish their research with us through the traditional subscription access route or make their paper immediately open access through payment of an article-processing charge (APC). Authors will not be required to make a final decision about access to their article until it has been accepted. <https://www.springernature.com/gp/open-research/transformative-journals> Find out more about Transformative Journals

Authors may need to take specific actions to achieve [compliance with funder and institutional open access mandates](https://www.springernature.com/gp/open-research/funding/policy-compliance-faqs). If your research is supported by a funder that requires immediate open access (e.g. according to [Plan S principles](https://www.springernature.com/gp/open-research/plan-s-compliance)) then you should select the gold OA route, and we will direct you to the compliant route where possible. For authors selecting the subscription publication route, the journal's standard licensing terms will need to be accepted, including [self-archiving policies](https://www.springernature.com/gp/open-research/policies/journal-policies). Those licensing terms will supersede any other terms that the author or any third party may assert apply to any version of the manuscript.

If you have not already done so, we strongly recommend that you upload the step-by-step protocols used in this manuscript to protocols.io (<https://protocols.io>), an open online resource that allows researchers to share their detailed experimental know-how. All uploaded protocols are made freely available and are assigned DOIs for ease of citation. Protocols and Nature Portfolio journal papers in which they are used can be linked to one another, and this link is clearly and prominently visible in the online versions of both. Authors who performed the specific experiments can act as primary authors for the Protocol as they will be best placed to share the methodology details, but the Corresponding Author of the present research paper should be included as one of the authors. By uploading your Protocols onto protocols.io, you are enabling researchers to more readily reproduce or adapt the methodology you use, as well as increasing the visibility of your protocols and papers. You can also establish a dedicated workspace to collect your lab Protocols. Further information can be found at <https://www.protocols.io/help/publish-articles>.

Nature Cell Biology encourages authors presenting evidence for cell, biological, molecular, and genetic interactions to consider communicating these findings using Biofactoid (<https://biofactoid.org/>). This tool helps users share a searchable representation of interactions (e.g. binding, gene expression, post-translational modification) between genes, gene products, or chemicals. Information added to Biofactoid, with author attribution, is shared on social media and public databases, such as Pathway Commons, where it can be discovered and analyzed in the context of a large and growing corpus of knowledge.

With kind regards,

Melina Casadio, PhD
Senior Editor, Nature Cell Biology
Consulting Editor, Nature Structural & Molecular Biology
ORCID ID: <https://orcid.org/0000-0003-2389-2243>

** Visit the Springer Nature Editorial and Publishing website at http://editorial-jobs.springernature.com?utm_source=ejp_NCB_email&utm_medium=ejp_NCB_email&utm_campaign=ejp_NCB for more information about our career opportunities. If you have any questions please click [here](mailto:editorial.publishing.jobs@springernature.com).

Response to reviewers

We thank the reviewers for their time, insightful comments, and ideas, which we feel have substantially elevated our manuscript and its findings. In response to reviewer requests, the following figure panels have been added to the revised manuscript:

Fig. 2i | Cardiolipin abundance in *crd1Δ* yeast

Fig. 2j | Respiratory growth of untreated *crd1Δ* yeast

Fig. 2k | Respiratory growth of WT and *crd1Δ* yeast treated with DOX

Fig. 2l | TAG abundance in *crd1Δ* yeast treated with doxycycline

Fig. 3k | Schematic of new [¹³C₁]-oleic acid pulse chase labeling experiment

Fig. 3l | Abundance of heavy [¹³C₃]-triolein in WT and *GDP-PLN1* overexpressing cells after pulse

Fig. 3m | Abundance of heavy [¹³C₃]-triolein in WT and *GDP-PLN1* overexpressing cells during chase

Fig. 3n | Respiratory growth of WT and *GDP-PLN1* overexpressing cells treated with DOX

Fig. 4a | Redone diagram of TAG catabolism in yeast

Fig. 4b | Nile Red fluorescence in WT, *tg1ΔΔΔ*, *atg14Δ*, and *ldo45Δldo16Δ* yeast after acute DOX treatment

Fig. 5e | Updated growth figure combining WT and *ATGL*^{KO} HAP1 cells grown in galactose with DOX

Fig. 5g | Population doublings for WT, *ATGL*^{KO}, and *CPT2*^{KO} HAP1 cells grown in galactose with DOX

ED Fig. 2e | Abundance heatmap for all tail lengths and saturation levels of TAGs in DOX treated cells

ED Fig. 2f | Heatmap with TAG abundance for all detected tail lengths and saturation levels in *rrf1* cells

ED Fig. 2g | Heatmap with TAG abundance for all detected tail lengths and saturation levels in *hsp10* cells

ED Fig. 2h | BODIPY stained WT and *hsp10* cells grown in respiratory media

ED Fig. 2i | Nile Red fluorescence for WT and *hsp10* cells grown in respiratory media

ED Fig. 2n | Cardiolipin and phosphatidylglycerol abundance in *crd1Δ* yeast

ED Fig. 3c | *PLN1* mRNA abundance in WT and group 3 strains

ED Fig. 3e | Growth assay of WT and *GDP-PLN1* overexpressing cells in respiratory media

ED Fig. 3f | Growth assay of WT and *GDP-PLN1* overexpressing cells in respiratory media treated with DOX

ED Fig. 4a | Abundance of TAGs in *ldo45Δldo16Δ* yeast treated with DOX

ED Fig. 4b | Abundance of TAGs in *atg14Δ* yeast treated with DOX

ED Fig. 5c | Resp. growth of DOX treated WT, *dga1Δ*, *lro1Δ*, *dga1Δlro1Δ* and *dga1Δlro1Δare1Δare2Δ* cells

ED Fig. 5f | mtDNA abundance in WT and *tg1ΔΔΔ* cells treated with DOX in respiratory conditions

ED Fig. 5g | [¹³C₃]-triolein abundance in WT and *tg1ΔΔΔ* cells supplemented with oleic acid

ED Fig. 5i | Respiratory growth of WT, and *tg1ΔΔΔ* treated with DOX supplemented with oleic acid

In addition to above, we have provided the following figures to address reviewer concerns within the rebuttal:

Rebuttal Figure 6 | Abundance of ergosterol ester species in WT and Group 3 strains

Rebuttal Figure 7 | Correlation of original multiomic screen with follow up proteomic and lipidomic experiments

Rebuttal Figure 8 | Abundance of peroxisomal proteins in DOX treated yeast

Rebuttal Figure 9 | Abundance of phosphatidic acid (PA) species in Group 3 strains

Rebuttal Figure 16 | Abundance of light species of CL, PE, and PC after [¹³C₁]-oleic acid pulse

While we have improved the manuscript in many ways, we have made four major additions to the revised edition:

- 1.) Based on the comments from **R1**, we have explored whether cardiolipin biosynthesis is required for efficient recovery from mitochondrial dysfunction. We found that a *crd1Δ* strain, which is unable to synthesize cardiolipin, failed to mobilize triacylglycerol species and were unable to efficiently recover from mitochondrial dysfunction. This data further strengthens our conclusion that mobilized acyl-chains from TAG species are shuttled into cardiolipin biosynthesis to enable cellular recovery.
- 2.) Also based on the comments from **R1**, we generated more extensive lipophagy-specific deletion mutants to test the role of lipophagy in TAG mobilization during mitochondrial dysfunction. We found that deleting either *atg14* or *lso45/16* had no effect on TAG mobilization in response to doxycycline treatment. This data further supports our conclusion that TAG lipolysis through the Tgl3-5p proteins is the primary source of TAG mobilization
- 3.) Based on the comments from **R2**, we updated the reporting of our metabolite measurements to match the standards in the field. We also examined if there was a shift in the level of saturation or total chain length in TAG species during recovery from mitochondrial dysfunction. We found that, overall, almost all TAG species were mobilized following mitochondrial dysfunction. However, there were some highly unsaturated TAG species that remained either unchanged or increased.
- 4.) Based on the comments from **R3**, we developed a novel [¹³C₁]-oleic acid pulse-chase experimental design to acquire direct evidence that Pln1p is involved in the mobilization of TAG species. By using an inhibitor of TAG biosynthesis, we were able to show that over-expression of *PLN1* significantly inhibited TAG mobilization during mitochondrial dysfunction.

Point-by-point response to all reviewer comments:

Reviewers' Comments:

Reviewer #1:

Remarks to the Author:

This is an interesting study dissecting how budding yeast mount responses to mitochondrial stress. Omics based screening of multiple yeast stains with defects in mitochondrial homeostasis identified triglyceride synthesis as an important pathway for stress response. TG mobilization via Tgl lipases was found to be important for mitochondrial stress response. This lipid mobilization is proposed to fuel mitochondrial biogenesis (specifically cardiolipin synthesis), rather than other lipids or bioenergetics at peroxisomes, to enable adaptation and survival.

Overall, this is an interesting and informative study that links triglyceride stores and lipid droplets to mitochondrial stress adaptation. The study nicely combines large-scale omics screening with pathway analysis and functional outputs. The implication that lipids mobilized from TAG are channeled into cardiolipin synthesis is intriguing but requires additional experiments to more fully test. There are also additional control experiments needed to allow some of the conclusions to be stated.

We would like to thank the reviewer for their careful consideration and assessment of our manuscript as well as recognizing the strengths of our work.

General concerns:

1) A key question is whether TAG is a necessary lipid species for the DOX survival response. Would providing exogenous fatty acids directly (without their storage as TAG) functionally replace TG stores? For example, does feeding the *tgl*-triple deletion yeast strain oleic acid during DOX treatment rescue similarly to yeast that have *Tgls*?

We thank the reviewer for this suggestion. We too are interested in whether oleic acid supplementation can bypass TAG lipolysis via the *Tgl3-5p* ligases, and we feel that it would bolster the findings of our manuscript. To explore this idea, we cultured wild-type and *tgl* $\Delta\Delta\Delta$ yeast in the presence or absence of heavy [$^{13}\text{C}_1$]-oleic acid and measured their ability to recover from doxycycline treatment. We began testing multiple concentrations of oleic acid and chose a concentration of 0.0002% (v/v) as it was able to produce strong labeling of measured [$^{13}\text{C}_3$]-Triolein (**Rebuttal Fig 1a**). This concentration is 500-fold lower than what is commonly used for loading lipid droplets in yeast^{1,2}. However, after prolonged exposure at higher concentrations of oleic acid (.1% and 0.002% v/v), we observed a significant growth defect in wild-type yeast in respiratory media without DOX treatment (data not shown). Despite the oleic acid treatment, *tgl* $\Delta\Delta\Delta$ yeast still exhibited a significant growth defect at the later time point when treated with doxycycline and were not significantly different from yeast without supplementation (**Rebuttal Fig 1b**). This suggests that free oleic acid cannot bypass integration into TAGs for mitochondrial stress recovery. We have added this data into the updated manuscript (lines 260-263, line 351).

Figure R1

Rebuttal Figure 1. a, Normalized relative abundance of heavy [$^{13}\text{C}_3$]-Triolein (m+3) in WT and *tgl* $\Delta\Delta\Delta$ cells treated with [$^{13}\text{C}_1$]-oleic acid or ethanol control. Samples were collected after 24 hour growth and normalized to the WT untreated sample. **b**, Growth assay of WT and *tgl* $\Delta\Delta\Delta$ grown in doxycycline treated with [$^{13}\text{C}_1$]-oleic acid or ethanol control. Growth was determined by the optical density (600 nm wavelength) of cells after 24 hours (Timepoint E) or 48 hours (Timepoint L) in 50 ml YEP media containing 3% (w/v) glycerol and 0.1% (w/v) glucose). For all experiments n=3 independent biological replicates for each condition.

2) *Cardiolipin* is proposed to be the major mitochondrial lipid species that TAG derived lipids are channeled into. Further investigation of how loss of *Crd1p* impacts DOX and mito stress survival would strengthen this claim.

We agree with the reviewer that expanding our investigation into how loss of cardiolipin synthase (*Crd1p*), impacts mitochondrial stress recovery would greatly strengthen our claims. To address this, we generated a *CRD1* knockout yeast strain (*crd1* Δ) and characterized its growth under doxycycline treatment. Consistent with previous characterization of *CRD1*³, deletion of *CRD1* resulted in the total loss of cardiolipin in cells both treated and untreated with doxycycline (**Rebuttal Fig 2a-b**). Likewise, consistent with past reports on *crd1* Δ , we observed the loss of cardiolipin occurred concurrently with a sharp increase in the abundance of the cardiolipin precursor phosphatidylglycerol (PG) (**Rebuttal Fig 2b**). Although this strain did display a slight growth defect in glycerol, the increase in PG presumably allows the *crd1* Δ strain to maintain respiratory growth (**Rebuttal Fig 2c**). The increase in PG did not, however, enable this strain to efficiently recover from doxycycline treatment like its wild-type counterpart (**Rebuttal Fig 2d**). This data further suggests that cardiolipin is essential for recovery to mitochondrial stress and that the increase in cardiolipin we observe in our Group 3 yeast strains is an adaptive response that promotes efficient recovery. Interestingly, we found that in addition to their lack of recovery, *crd1* Δ yeast did not mobilize TAG stores in response to doxycycline

treatment (**Rebuttal Fig 2e**). This may suggest that the signal for TAG mobilization requires cardiolipin. Determining the signal which initiates TAG mobilization and the role cardiolipin has in this process will require additional experimentation and is an active area of investigation. This data and text have been added to the revised manuscript (lines 161-175)

Figure R2

Rebuttal Figure 2. **a**, Normalized abundance of the four most abundant species of cardiolipin in wild-type and *crd1Δ* strains, untreated and treated with doxycycline at the early timepoint. Cardiolipin species are denoted by color. Abundance is given as the relative peak area normalized to the average untreated WT values (n=3 independent biological replicates). **b**, Relative lipid abundance versus statistical significance for *crd1Δ* yeast. Cardiolipin (CL) species are colored in red and phosphatidylglycerol (PG) species in pink. **c**, **d**, Growth assay of WT and *crd1Δ* yeast strains. Cells were treated with either DMSO (**c**) vehicle control or DOX (**d**). Growth was determined by the final optical density (600 nm wavelength) of cells after 24 hours (c, d, E timepoint) in 50 ml YPG respiratory media containing 3% (w/v) glycerol and 0.1% (w/v) glucose) and again at 48 hours of growth (d, L timepoint). **e**, Relative lipid abundance versus statistical significance for *crd1Δ* yeast treated with doxycycline. Quantified TAG species are colored in light blue with TAG species that are significantly changed (|FC| > 0.7, $p < 0.05$) highlighted in dark blue. All other lipid species are colored in grey.

3) The TAG synthesis enzyme *Dga1p* was found to be important for yeast to survive DOX treatment (ED, Fig 5). However there are two TAG synthesis enzymes in yeast, *Dga1* and *Lro1p*. Is *Lro1p* similarly as important as *Dga1p*? What occurs in the double *dga1/lro1* KO, or in yeast that lack all four neutral lipid synthesis enzymes (*dga1/lro1/are1/are2*) that cannot form droplets at all?

While we initially focused on *Dga1p* as the primary diacylglycerol acyltransferase (DGAT) enzyme involved in generating TAGs for mitochondrial stress recovery, we agree with the reviewer that it would be both interesting and important to test the essentiality of both TAG synthesis enzymes, *Dga1p* and *Lro1p*. To address this concern, we generated a *LRO1* knockout yeast strain (*lro1Δ*) as well as a *dga1Δ lro1Δ* double knockout strain in our W303 background. Concurrently, we also generated a *dga1Δ lro1Δ are1Δ are2Δ* quadruple knockout strain previously shown to lack all lipid droplets⁴. We then cultured these strains and tested their ability to efficiently recover from mitochondrial stress via treatment with doxycycline. Consistent with our previous results, any strain lacking *DGA1* (*dga1Δ*, *dga1Δ lro1Δ*, *dga1Δ lro1Δ are1Δ are2Δ*) was unable to efficiently recover from mitochondrial stress (**Rebuttal Fig 3a**). The *lro1Δ* strain, however, grew and recovered at an equivalent rate to the wild-type strain. This result further indicates that *Dga1p* is the primary and only essential TAG synthesis enzyme required for mitochondrial stress recovery under our growth conditions. Nevertheless, we cannot rule out the possibility that under other growth conditions, such as those in which *Lro1* has been shown to be more important for TAG biosynthesis⁵, that the protein may play a more significant role. We have added this data to the revised manuscript (lines 243-245).

Figure R3

Rebuttal Figure 3. a, Growth assay of WT, *dga1Δ*, *lro1Δ*, *dga1Δlro1Δ* double deletion and *dga1Δlro1Δare1Δare2Δ* quadruple deletion ($\Delta\Delta\Delta\Delta$) yeast treated with DOX (n=3 independent biological replicates for each condition). Growth was determined by the optical density (600 nm wavelength) of cells after 24 hours (Timepoint E) or 48 hours (Timepoint L) in 50 ml YEP media containing 3% (w/v) glycerol and 0.1% (w/v) glucose.

4) *Tgl* lipases are proposed to be the major TAG mobilization pathway versus micro-lipophagy at the yeast vacuole, which was blocked by *Atg1p* and *Atg15p* deletion. Both these *Atg* deletion may have pleiotropic effects since *Atg1p* also governs general macroautophagy, and *Atg15p* is a vacuolar lipase and its loss may perturb general vacuole function. To further investigate micro-lipophagy in this pathway, depleting factors more specifically linked to micro-lipophagy like *Atg14p*, *Vac8p*, or the *Ldo16/45p* proteins would be necessary.

We initially chose the *atg1Δ* and *atg15Δ* deletions to show that not only was micro-lipophagy not required for mitochondrial stress recovery, but general autophagy was also not required. However, we agree with the reviewer that, because of the pleiotropic effects, these strains were perhaps not appropriate for testing the essentiality of yeast micro-lipophagy in isolation. To address this concern, we generated the three recommended knockout strains *atg14Δ*, *vac8Δ* and *ldo16/45Δ* (*ldoΔΔ*) in the W303 background. Initial characterization of the *vac8Δ* strain revealed that it displayed a severe growth defect in the absence of stress conditions and was therefore discontinued for analysis (data not shown). Experimentation was continued with the *atg14Δ* and *ldoΔΔ* deletion strains. Similar to the *atg1Δ* and *atg15Δ* deletions (**Initial manuscript Fig 4b**), *atg14Δ* and *ldoΔΔ* deletion strains showed decreased TAG levels after acute doxycycline treatment as demonstrated by Nile Red staining and LC-MS/MS lipidomics (**Rebuttal Fig 4a-e**). This contrasted with the triacylglyceride lipase triple mutant (*tg/ΔΔΔ*), which was again unable to mobilize TAG species in response to mitochondrial stress (**Rebuttal Fig 4a,c**). This data further supports our claim that lipolysis through the *Tgl3-5p* lipases is the primary form of TAG mobilization during recovery from mitochondrial stress and that micro-lipophagy is not essential to recover from these stress conditions. We have added these new data to the revised manuscript (lines 219-225) as well as an updated schematic that we feel better represents yeast micro-lipophagy and the roles of *Atg14p* and *Lro45p/Lro16p* (**Rebuttal Fig 4f**).

Figure R4

Rebuttal Figure 4. **a**, Relative fluorescent signal from Nile Red staining of WT, *tgl* triple deletion (*tgl3Δtgl4Δtgl5Δ*, *tglΔΔΔ*), *atg14Δ*, and *ldo45/16Δ* (*ldoΔΔ*) yeast ($n=3$ independent biological replicates measured in duplicate for each condition). Cells were treated with either DOX or a DMSO vehicle control. Error bars represent standard deviation. ** $P < 0.01$, n.s. = not significant, two-sided Student's t-test. **b, c, d, e**, Relative lipid abundance versus statistical significance for WT (**b**) *tglΔΔΔ* (**c**) *atg14Δ* (**d**), and *ldo45/16Δ* (**e**) yeast. For **b-e** quantified TAG species are colored in light blue with TAG species that are significantly changed ($|FC| > 0.7$, $p < 0.05$) highlighted in dark blue. All other lipid species are colored in grey. **f**, Schematic illustrating the pathways for TAG mobilization in yeast. TAGs stored in lipid droplets are accessed through lipolysis using the Tgl lipases (top), or micro-lipophagy which requires the Ldo45p and Atg14 proteins.

Reviewer #2:

Remarks to the Author:

This is a well-written manuscript describing triglyceride changes under mitochondrial stress in yeast.

We would like to thank the reviewer for this comment. We greatly appreciate the time given to reading the initial manuscript.

It appears that overall TG species were mobilized during stress and under all studied conditions, but was there also a shift of fatty acyl chain length or total degrees of unsaturation?

The reviewer is correct in their statement that overall TAG species were mobilized under all Group 3 stress conditions. In addition, we agree that it would be important to examine the chain lengths and saturation levels of the TAG species to determine if remodeling occurred as well. To address this comment, we examined the relative changes in TAG abundance for different chain lengths and saturation levels in each of the Group 3 strains. We found that, in general, there is an overall decrease in almost all TAG species in all three of the group 3 strains (**Rebuttal Fig 5a-c**). However, there does seem to be a relative increase in TAG species with lower levels of saturation, particularly in the sicker *rrf1* and *hsp10* strains. As changes in lipid saturation can affect mitochondrial membranes⁶, particularly membrane curvature⁷, this may represent a targeted shift by the cell to decrease acyl-chain saturation, however, more work is required to conclusive statements about this observation. We have added this data to the revised manuscript (lines 133-135).

Figure R5

Rebuttal Figure 5. a, b, c, Heatmaps detailing the relative abundance of TAG species in DOX treated (a), *rrf1* (b), and *hsp10* (c) yeast compared to wildtype cells harvested at the early time point. Saturation and length of TAG species is given by the number of double bonds or total number of carbons in the acyl-tails respectively.

The S-lens conditions are very high for the lipidomic analysis. An s-lens of 90 will cause significant source fragmentation which could cause misidentification of lipids. This could be observed across many lipid classes. Was this presence of source fragments accounted for? Were ergosterol esters identified in the lipids? This is a class of lipids that would be highly altered based on the source conditions and the ester group could easily fragment. There is some research on the role of ergosterol esters in stress response, so identification of them could be important, however this stress response could be more heat derived and may be out of the scope of this work.

We agree with the reviewer that in-source fragmentation of lipids is an issue that must be addressed. Lipidex, the software tool we used to search and identify lipids, accounts for this in-source fragmentation. The process is such that co-eluting identified fragment ions are removed during an initial filtering step⁸. During the initial screen, ergosterol-esters (EEs) were not included in the search libraries. However, we agree that they represent an intriguing molecule class to measure both from a biological and technical standpoint. We therefore generated a new spectral library for common EEs and re-searched our raw data files for them. Re-searching the data files resulted in the identification of 4 individual EEs, although two of these (16:1 and 18:1) had multiple peaks with slightly varying retention times and cannot be unambiguously identified without standards. Of the two that were identified, both displayed a similar pattern in which the EE abundance was slightly higher at the earlier timepoint and then either returned to wild-type levels or was further decreased at the later timepoint (**Rebuttal Fig 6a-b**). While this observation may be biologically relevant, future experiments are required with analytical standards to both correctly identify the EE compounds and determine if these increases are indeed meaningful.

Figure R6

Rebuttal Figure 6. a, b, Relative abundance of the ergosterol esters 16:0 (a) and 20:3 (b) in wildtype and Group 3 strains at either the early (E) or later (L) timepoint. Abundance is given as the relative peak area normalized to the average WT values (n=3 independent biological replicates).

It is strange that lipids and proteomic were conducted with 2 different methods each. Was there a reason for the change in methods? What was the overall correlation? Did the methods find the same number of lipids and proteins?

The use of two different methods for the proteomics and lipidomics was a consequence of multiple factors. First, from the time of the original screen to the time of the follow up data, the laboratory physically moved from the University of Wisconsin-Madison to Washington University in St. Louis. With this move, new instruments and LC systems were purchased for the completion of the project. These instruments required alternative methods that were optimized specifically for them. Secondly, the original extraction method for the screen was designed to isolate all three -omes from a single sample. For many of the follow-up experiments, we were only interested in investigating a single class of molecules and therefore reverted to a simpler, more traditional extraction method for that individual class. We found that for both the proteomics and lipidomics, there were similar amounts of molecules identified and high positive correlation between the methods (**Rebuttal Fig 7a-c**).

Figure R7

Rebuttal Figure 7. a, b, Correlation of intensities for identified proteins (a) or lipids (b) from the initial screen and follow up experiments. Wildtype (left) and doxycycline treated (right) yeast were compared as they were the only conditions overlapping between the two datasets. Pearson coefficient values were used to calculate correlation and given as R².

Please add to the text the level of identification confidence based on the metabolomics standard initiative. Metabolites were identified by MS/MS primarily since authentic standards were not analyzed. There was very little discussion of any metabolite changes which is a bit surprising since metabolites should be altered under stress especially ATP and other nucleotide phosphates. It might be that the extraction method did not retain this class of compounds.

We have now added the level of identification for each metabolite measured in the supplemental table. The reviewer is correct that most metabolites were identified using MS/MS spectra (level 2), however, we also used analytical standards for 39 of the measured metabolites during method development (level 1). While many of the observations in this metabolite dataset are interesting, we elected to not highlight them as to keep the manuscript focused on the main story involving the increase in lipid mobilization. However, future projects and manuscripts are currently underway examining some of these observations.

There are a few errors in the methods where the superscript was not added correctly for AGC target values.

We are thankful to the reviewer for identifying these errors. The values have now been corrected to superscript in the updated form of the manuscript.

Reviewer #3:

Remarks to the Author:

In their manuscript entitled “Triacylglycerol mobilization underpins mitochondrial stress recovery” Baker et al. investigated mechanisms that aid recovery from mitochondrial respiration stress. The authors used a multiomic approach to compare the response and recovery of various yeast strains and conditions that disturb mitochondrial functions. They have identified few mutants/drugs that exhibit a growth defect when grown in non-fermentable carbon source. These mutants/drugs could not undergo diauxic shift in the same timeframe as wild type but could eventually recover and grow. This study explores the cellular requirements for the recovery of these respiratory defective mutants. The work demonstrated that TAG catabolism and CL biosynthesis are key to the respiration and growth recovery of the mutants. Finally, the authors provide evidence that a similar mechanism exist in mammals.

This is a very interesting study that includes an extensive amount of data and broad range of methods to support the hypotheses. It is highly relevant to mitochondrial diseases with decreased oxphos capacity. The quality of the data is excellent and the analyses are very convincing. The manuscript is well-written, but the story is difficult to follow in some of the sections and is often missing details in the main text and figure legends. The take-home messages are novel but they are not always clearly delivered. For example, not enough emphasis is placed on the important message that TAG conversion to CL is required for mitochondrial protein upregulation and therefore for all aspects of mitochondrial biogenesis. Overall, this study will be of interest for the broad scientific community if the following comments could be addressed.

We thank the reviewer for their time and thoughtful consideration of our manuscript. We are glad that they view the data in such a high regard and agree with their sentiment that this work is highly relevant to mitochondrial disease. We have addressed their comments below.

Comments:

*“To investigate how *S. cerevisiae* respond to and recover from certain mitochondrial stress”- I am not sure how accurate this statement is. It might be easier to the reader if this is presented in a more straightforward way with the question of how cells compensate for suboptimal mitochondrial oxphos capacity. Since the study focuses on the ability to adapt to conditions in which mitochondrial respiration is essential, I think it will be beneficial for the reader to present the question as such.*

We agree with the reviewer that classifying the yeast strains as undergoing mitochondrial stress is slightly vague and not entirely straightforward. However, we feel that mitochondrial stress is the only all-encompassing terminology that applies to all 13 experimental strains. Certainly, “*suboptimal mitochondrial oxphos capacity*” would apply to the group 3 strains that were the main focus of follow-up studies; however, many of the other strains, such as those in group 1, have no discernable OxPhos phenotype. Furthermore, we feel that reducing the presentation of the screen layout to an individual term, such as reduced OxPhos capacity, undersells the potential for finding novel biology involved in the recovery of other factors, such as disruptions of mitochondrial proteostasis or mitochondrial redox stress, both of which are present in strains within our dataset. As such, while we value any efforts to clarify our message, we feel that this particular recommendation would be an oversimplification that risks being scientifically inaccurate.

Similarly- “Collectively, these data demonstrate that Group 3 strains mount a successful cellular response to mitochondrial stress distinct from the currently appreciated response pathways.” I agree with this statement but this is too early to conclude this before showing the rest of the data. It can still be that these strains are identical to wild type but just slow to react.

We agree with the reviewer that the subsequent data in the manuscript further support our model that the Group 3 strains are undergoing a so-far unappreciated cellular response pathway. We also agree that the term “demonstrates” may be too strong of language to use at this point in the manuscript. However, we feel that the observations of changes in mitochondrial protein abundance, which have already been established in the manuscript, are enough to first suggest that the sick strains are undergoing a distinct stress response. This conclusion is first based on the observation that the stressed strains have a lower abundance of mitochondrial proteins at the earlier timepoint compared to the WT (**Initial Submission Fig 1f, ED Fig 1h**). The mitochondrial protein abundance in these strains is then sharply increased post-recovery at the later timepoint (**Initial Submission Fig 1g, ED Fig 1i**). We believe these data supports a conclusion that these cells are not identical to wild type but are instead undergoing a change over time that must be triggered by a cellular response pathway. Since we already established that more appreciated pathways of mitochondrial stress recovery were not activated (**Initial Submission Fig ED Fig 1l-n**), it suggests that they are undergoing a novel response, however, the subsequent data is needed to fully make these statements. We have changed the language in the manuscript to reflect this distinction (lines 124-125).

*Fig 1h might suggest that the *ssc1* mutant has no or very low amounts of mtDNA. It can explain why this strain could not recover and grow on glycerol. This should be tested. This result also raises an important question – why this mutant does not affect TAGs. I am not sure this question was attempted to be addressed and it should be at least referred to in the discussion. Is it a direct effect of mtDNA loss or maybe just a complete block of respiration and inability to make enough ATP to execute a response?*

The reviewer is correct that the *ssc1* mutant strain has no detectable levels of mitochondrial DNA (mtDNA) (**Initial Submission Fig 1h**), and this undoubtedly contributes to its lack of recovery. Like many mutant strains undergoing severe mitochondrial stress, this strain will rapidly lose its mtDNA and become respiration incompetent. With this rapid loss of mtDNA in the *ssc1* strain, we are unable to test its growth if it had mtDNA.

As a consequence of this, we cannot determine if it is the lack of mtDNA or the lack of OxPhos/ATP generation that results in a non-recoverable phenotype.

Why the *ssc1* strains does not attempt to mobilize TAGs when it has only slightly elevated levels of stress compared to the other strains is an intriguing question that we are actively pursuing and is one of the fundamental observations in this manuscript. Since we are unable to test whether it is a direct effect of mtDNA loss or a severe decrease in OxPhos (due to the complications of separating these phenotypes explained above), we are unable to determine exactly why this occurs. We agree that these are important distinctions though and have now expanded upon both points in the discussion (lines 360-368).

Fig 1h- The mutants seem to have more mtDNA. This result seems to be in line with the authors' hypothesis that the mutants have higher rates of mitochondrial biogenesis and therefore higher mitochondrial mass. This should be referred to in the manuscript. The authors could also test the mitochondrial mass in at least some of the mutants in addition to DOX treatment.

The reviewer is correct that at the later time point the mutants have higher levels of mtDNA compared to their earlier counterparts. We agree that this increase supports our hypothesis that the mutant strains have an increased amount of mitochondrial biogenesis at the later timepoint. We have further highlighted this observation in the manuscript (lines 114-115). While we agree that it would be ideal to directly measure the mitochondrial mass in the other strains, the “rainbow” strains of yeast used in the initial manuscript were unable to undergo the genetic manipulation required to generate the *rrf1* and *hsp10* mutations as they already contain the gene cassettes used for Cas9 integration. We believe that the increase in mtDNA, the increase in mitochondrial proteins abundance, and the increase in cardiolipin that were measured in these two strains are enough to support this conclusion.

Line 118- “Moreover, expression of defined retrograde stress response pathway markers that may aid recovery were not induced in any of these strains (Extended Data Fig. 1m,n), perhaps due to our selected parental yeast strain (W303)” What were the growth conditions in this analysis? Early or late collections? What does it mean “due to our selected parental yeast strain”?

This analysis was performed on yeast grown in respiration media (0.1% glucose, 3% glycerol) at the early timepoint. All yeast strains in our manuscript were grown in conditions that require respiration. As many of the studies on the retrograde response pathway have examined the pathway in fermentative conditions (2% glucose), we had previously listed these growth conditions as a potential reason for not observing a robust retrograde response, although this has been removed in the revised manuscript. We mentioned the W303 parental yeast strain as it has previously been shown to not exhibit a robust retrograde response under mitochondrial stress⁹. While we only cited this paper in the initial submission, we have now expanded the text about it to make it clearer to the reader (line 123).

How do the authors explain the low levels of cit2 in the mutants compared to wt?

Increase in *CIT2* expression during mitochondrial dysfunction is commonly observed during induction of the retrograde response^{10,11}. We do not observe induction of the retrograde response, potentially for the reasons listed above, and see lower levels of Cit2p in our strains. The lower levels of Cit2p are consistent with a reduction of multiple peroxisomal proteins in our sick strains (**Rebuttal Fig 8a**). Using confocal fluorescence imaging, we also observed a decrease in total peroxisomal volume fraction in DOX-treated cells at the earlier timepoint that recovered at the later timepoint (**Rebuttal Fig8b**). We do not know the exact mechanism behind these observed changes in peroxisome abundance, but they may suggest that peroxisomal biogenesis is linked to the same or similar response pathway to mitochondrial biogenesis under these conditions.

Figure R8

Rebuttal Figure 8. a, Relative protein abundances for doxycycline treated yeast collected at the early timepoint compared to WT versus statistical significance (n=3 independent biological replicates for both samples). Non-mitochondrial proteins are colored in light grey, mitochondrial proteins are colored in dark grey. Mitochondrial proteins that are significantly changed (|FC| > 0.7, p < 0.05) are highlighted in black. (b) Florescent microscopy derived cellular volume fractions of peroxisomes for cells treated with either vehicle control or DOX either pre- or post- recovery from mitochondrial stress.

Fig 2- Did the lipid analysis include phosphatic acid (PA)? If yes it would be helpful to add the results given that PA is a precursor of CL.

Phosphatidic acid (PA) was measured in our lipidomic analysis; however, we only made measurements on whole cell PA and do not have measurements for mitochondrial PA abundance. While we did detect modest increases in some of the species of PA that we detected (**Rebuttal Fig 9a**), we do not think we can make any conclusions on its effect on cardiolipin biosynthesis, since mitochondrial PA is the pool of PA used as a precursor for cardiolipin biosynthesis and is only a small fraction of the total PA found throughout the cell¹².

Figure R9

Rebuttal Figure 9. a, b, c, Heatmaps detailing the relative abundance of Phosphatidic acid (PA) species in DOX treated (a), *rrf1* (b), and *hsp10* (c) yeast compared to wildtype cells harvested at the early time point. Saturation and length of PA species is given by the number of double bonds or total number of carbons in the acyl-tails respectively. White values were not detected.

Fig 2B- The authors should include *hsp10* BODIPY imaging in the figure.

We agree with the reviewer that alternative methods of measuring TAGs in the *hsp10* strain would be valuable. To address this concern, we performed confocal microscopy on WT and *hsp10* yeast. While we did not detect a noticeable difference in the BODIPY stained strains (**Rebuttal Fig 10a**), we believe that the BODIPY stain is not quantitative enough to observe the more modest decrease in TAGs seen in the *hsp10* strain by LC-MS/MS (**Initial manuscript Fig 2a**). To further examine *hsp10* TAG levels and complete the observations we made with the other group 3 strains, we also performed Nile Red staining on *hsp10* yeast and measured the fluorescent signal. Using this more quantitative method, we were able to see a significant decrease in Nile Red signal in the *hsp10* yeast. The observations have been added to the revised manuscript (ED Fig. 2).

Figure R10

Rebuttal Figure 10. a, Representative confocal microscopy images of BODIPY (green) stained wild-type or *hsp10* yeast strains. Images were taken at 60X magnification. Scale bar represents 5 μ m. **b**, Relative fluorescent signal from Nile Red staining of WT or *hsp10* yeast strains harvested at the early timepoint.

Fig 3 h-k- I am not sure the authors can conclude that *Pln1* is important for TAG mobilization without direct evidence especially since it was previously shown to mediate TAG biogenesis. Another interpretation of the results is that *Pln1* overexpression increases TAG biogenesis and changes the balance of stored versus free TAG (less TAG is available for CL biosynthesis). *Pln1* deletion might have no impact due to compensation by other factors.

We agree with the reviewer that we previously did not have direct evidence to support *Pln1p* being involved in TAG mobilization and that this direct evidence is critical for making these conclusions. To address these concerns, we performed a modified ¹³C oleic acid pulse-chase experiment to directly measure the rates of TAG biogenesis and mobilization in wild-type and *PLN1* overexpressing cells. As in our previous heavy labelling experiments, we cultured yeast in respiratory media and treated them with ¹³C oleic acid for one hour before harvesting and measuring the abundance of newly synthesized heavy ¹³C-labelled triolein (**Rebuttal Fig 11a**). Overexpression of *PLN1* did not have a significant effect on the abundance of heavy-¹³C-labelled TAGs, suggesting that it does not drastically increase the rates of TAG biosynthesis (**Rebuttal Fig 11b**). Following the one-hour labelling with ¹³C oleic acid, we next treated the strains with cerulenin, a potent inhibitor of TAG biosynthesis¹³, along with doxycycline to induce TAG mobilization, and collected yeast after one, three, and five hours to measure the rate at which the heavy ¹³C triolein was mobilized (**Rebuttal Fig 11a**). By the three-hour timepoint, TAG levels in wildtype yeast were already trending lower than the *Pln1* overexpressing strain. This difference widened at 5 hours with wildtype yeast having significantly lower levels of heavy ¹³C TAG (**Rebuttal Fig 11c**). This experiment provides direct evidence that overexpression of *Pln1* inhibits TAG mobilization while not changing TAG biosynthesis under these conditions. We have now added this text along with the data to the revised manuscript (lines 199-215).

Figure R11

Rebuttal Figure 11. **a**, Schematic for acute DOX pulse-chase labeling experiment. WT and *GPD-pln1* overexpressing cells were inoculated and incubated for 23 hours before being treated for 1 hour with [¹³C₁]-oleic acid. This was followed addition of cerulenin and doxycycline to inhibit TAG biosynthesis and induce mobilization. Samples were collected a 1-, 3- and 5- hour timepoints post DOX addition. **b**, Normalized relative abundance of heavy [¹³C₃]-Triolein (m+3) in WT and *GPD-pln1* overexpressing cells pulsed with [¹³C₁]-oleic acid for one hour. **c**, Normalized relative abundance of heavy [¹³C₃]-Triolein (m+3) in WT and *GPD-pln1* overexpressing cells after addition of cerulenin and doxycycline at 1, 3, and 5 hours post addition. Abundances were normalized to the total [¹³C₃]-Triolein (m+3) abundance at T₀ (**b**).

Fig 3k – please add growth curves for this experiment

To address this comment and facilitate comparison with all other growth assays in the manuscript, we first established an overexpression system that can be used in YEP media by swapping the *URA3* cassette with a G418 resistance marker in the expression vector. We then measured the growth of WT, *tg1ΔΔΔ* and, *PLN1*-overexpression cells either untreated or treated with doxycycline over a period of 24 or 48 hours in YEP media containing 0.1% (w/v) glucose, 3% (w/v) glycerol. While the untreated strains had a negligible difference in growth in the untreated condition (**Rebuttal Fig 12a**), both the *tg1ΔΔΔ* and *PLN1*-overexpression strains had a significant growth defect when treated with DOX (**Rebuttal Fig 12b**). Interestingly, the strains had a similar growth defect under DOX treatment consistent with them having similar defects in TAG mobilization. This data has been added into the revised manuscript (Fig 3n, ED Fig 3e,f).

Figure R12

Rebuttal Figure 12. **a**, Growth assays of WT, *tgl* triple deletion (*tgl3Δtgl4Δtgl5Δ*, *tg1ΔΔΔ*), and *GPD-pln1* overexpressing yeast. Growth was measured by optical density (600 nm wavelength) of a 50 ml culture in YEP media containing 0.1% (w/v) glucose, 3% (w/v) glycerol.

Fig 3h- Are the transcription levels of *Pln1* different between *wt* and DOX/group 3 mutants?

To address this comment, we measure *PLN1* transcript levels using qPCR. Consistent with our proteomics measurements (**Initial Manuscript ED Fig 3b**), the DOX treated strain had a slight but significant reduction in the abundance of *PLN1* mRNA, whereas there was no difference in the *rrf1* strain and a slight increase in the *hsp10* strain (**Rebuttal Fig 13a**). This data, combined with our proteomic data, suggest that overall changes in *Pln1* abundance are minor compared to the overall reduction in TAGs and that the correlation analysis would still be the most reliable way to link *Pln1* to the TAG mobilization phenotype. We have added this data to the revised manuscript (line 192).

Figure R13

Fig 5 e-f – This is a very important result that positions TAG catabolism upstream of mitochondrial gene upregulation. The authors should test whether TAG catabolism is also necessary for mtDNA replication using the same strains and the assay used in fig 1h.

While we believe that our data support a model that puts TAG mobilization upstream of efficient mitochondrial biogenesis and recovery from mitochondrial stress, we do not believe that this necessarily puts it upstream of mitochondrial gene expression at a transcriptional level or upstream of the regulation of mtDNA replication. To determine the effect of TAG mobilization on mtDNA levels under mitochondrial stress, we used our qPCR method to measure relative mtDNA in wildtype and *tg/ΔΔΔ* yeast strains treated with doxycycline. We found that *tg/ΔΔΔ* yeast had identical levels of mtDNA compared to wildtype cells in untreated and treated conditions at both timepoints (**Rebuttal Fig. 14a**). This data, combined with our data on mitochondrial protein abundance, lead us to favor a model where TAG mobilization is upstream of mitochondrial biogenesis by supplying the required lipids for membrane expansion/cardiolipin biosynthesis. Without this, mitochondria remain dysfunctional and newly synthesized mitochondrial proteins are rapidly turned over as they are translated. We have expanded upon this in the discussion of the revised manuscript (lines 342-345).

Figure R14

Fig 6g and h- I assume that galactose medium was used here? It would be helpful to add the parental wild type strain to this graph (in the presence and absence of DOX) to verify that the untreated KO mutants do not have a growth defect. If there is a defect the authors should confirm that the combination of mutants with DOX is not additive.

The reviewer is correct that these cells were grown in galactose media. To aid the reader in comparing the growth rates of the wildtype and ATGL knockout cells, we combined them into a single graph (**Rebuttal Fig 15a**). We then used a previously published method for calculating synthetic growth defects¹⁴, measuring the doubling times of all cell lines either untreated or treated with doxycycline after five days of growth to determine if the modest growth phenotype seen in the ATGL knockout cells was additive with the doxycycline treatment.

We found that the combination of ATGL deletion and doxycycline treatment was not additive, and these cells were synthetic lethal (**Rebuttal Fig 15b**). In contrast, the CPT2 knockout cells had no difference in doubling times compared to the wildtype (**Rebuttal Fig 15b**) We have added these panels to the revised version of Fig. 6.

Figure R15

Rebuttal Figure 15. a, Growth assay of WT (blue) and ATGL knockout (green) HAP1 cells treated with DOX (dark shade) or DMSO vehicle control (light shade). Cells were plated into IMDM glucose containing media and grown for 24 hours before being swapped into DMEM galactose media (Day 0) to measure respiratory growth over 4-6 days. n=3 independent biological replicates. **b**, Population doublings from each of the strains as calculated from (a) and **Initial Manuscript Fig. 6h**.

Minor comments:

*It would be helpful to add references for the mutants used in this study. For example, to provide context to the claim that constitutive overexpression of *cox4* and *mDhfr* disrupts mitochondrial proteostasis.*

While we initially listed the references to each of the mutants used in the study in Table 1 (Initial Manuscript Table 1), we have now added more to the text to make this clearer to the reader where to find these citations (lines 86-87).

*Fig. 1 and S1 present data from 2 strains that are not mentioned in the text: *hsp78* and *yme1**

The *hsp78* Δ strain and the *yme1*^{K327E} were both in the group of strains that grew similarly to wild type and had few changes throughout all of our measurements (Group 1). These strains are included in the descriptions within Table 1.

Fig 2g- the authors should discuss the decrease in PE and PC

We also thought that the decrease in heavy labeled PE and PC was interesting in these cells. We went back through the data and found that while the heavy labeled species of these lipids were decreased (**Initial Manuscript Fig 2g**), the light species of these same lipids were increased (**Rebuttal Fig 16a**). This could suggest that while nascent cardiolipin is being synthesized though the lipolysis of heavy TAGs, nascent PE and PC are coming from other sources such as de novo FA biosynthesis. However, more work will have to be done to make such conclusions.

Figure R16

Rebuttal Figure 16. a, Normalized abundance of light 72:4 CL (Red), 36:2 PC (Purple), and 36:2 PE (Orange). Abundance is given as the relative peak area normalized to the average WT values (n=3 independent biological replicates). WT cells were inoculated and incubated for 23 hours and treated for 1 hour with [¹³C₁]-oleic acid. This was followed by a 3-hour acute DOX treatment and collection.

The figures and figure legends are often missing details that could help evaluate and interpret the data and the conclusion. Moreover, some small details are missing. A few examples:

Fig 1B- What is WTV?

Fig 1B and C- it would be helpful to add a color key to the legend

Fig 1 legend –

“Mitochondrial proteins that are significantly changed ($|FC| > 0.7$, $p < 0.05$) (g) are highlighted in black.” Why is this statement relevant only to g? black labels are depicted in fig f too.

Also, in g and f- the color key in the graph is unclear. What is “L” from the left of the black circle? This is relevant to other figures.

Please specify the conditions used in each experiment. For example, what were the conditions in figure 3i? how long were the cells grown in glycerol, how long was the DOX treatment for?

Fig 3J and k- typo in the legend, replace f and g with j and k

Lines 206- Isn't dga1 included in Fig 5a as well?

Fig 5f –f is missing from the figure.

Sup table 2- It would be helpful to include gene names in the table.

We thank the reviewer for providing such careful overview of the text and figures of our manuscript. We have corrected all these oversights in the main text and figures as well as significantly updated our text and legends to give more information and make them clearer for the reader.

References

1. Gao, Q. *et al.* Pet10p is a yeast perilipin that stabilizes lipid droplets and promotes their assembly. *J Cell Biol* **216**, 3199–3217 (2017).
2. Rogers, S. *et al.* Triglyceride lipolysis triggers liquid crystalline phases in lipid droplets and alters the LD proteome. *J. Cell Biol.* **221**, e202205053 (2022).
3. Jiang, F. *et al.* Absence of Cardiolipin in the *crd1* Null Mutant Results in Decreased Mitochondrial Membrane Potential and Reduced Mitochondrial Function*. *J. Biol. Chem.* **275**, 22387–22394 (2000).
4. Sandager, L. *et al.* Storage Lipid Synthesis Is Non-essential in Yeast*. *J. Biol. Chem.* **277**, 6478–6482 (2002).
5. Oelkers, P. *et al.* A Lecithin Cholesterol Acyltransferase-like Gene Mediates Diacylglycerol Esterification in Yeast*. *J. Biol. Chem.* **275**, 15609–15612 (2000).
6. Romanauska, A. & Köhler, A. Lipid saturation controls nuclear envelope function. *Nat. Cell Biol.* **25**, 1290–1302 (2023).
7. Baumgart, T., Hess, S. T. & Webb, W. W. Imaging coexisting fluid domains in biomembrane models coupling curvature and line tension. *Nature* **425**, 821–824 (2003).
8. Hutchins, P. D., Russell, J. D. & Coon, J. J. LipiDex: An Integrated Software Package for High-Confidence Lipid Identification. *Cell Syst.* **6**, 621-625.e5 (2018).
9. Kirchman, P. A., Kim, S., Lai, C.-Y. & Jazwinski, S. M. Interorganelle Signaling Is a Determinant of Longevity in *Saccharomyces cerevisiae*. *Genetics* **152**, 179–190 (1999).
10. Liao, X., Small, W. C., Srere, P. A. & Butow, R. A. Intramitochondrial Functions Regulate Nonmitochondrial Citrate Synthase (CIT2) Expression in *Saccharomyces cerevisiae*. *Mol. Cell. Biol.* **11**, 38–46 (1991).
11. Liao, X. & Butow, R. A. RTG1 and RTG2: Two yeast genes required for a novel path of communication from mitochondria to the nucleus. *Cell* **72**, 61–71 (1993).
12. Horvath, S. E. & Daum, G. Lipids of mitochondria. *Prog. Lipid Res.* **52**, 590–614 (2013).
13. Kurat, C. F. *et al.* Obese Yeast: Triglyceride Lipolysis Is Functionally Conserved from Mammals to Yeast*. *J Biol Chem* **281**, 491–500 (2006).
14. To, T.-L. *et al.* A Compendium of Genetic Modifiers of Mitochondrial Dysfunction Reveals Intra-organelle Buffering. *Cell* **179**, 1222-1238.e17 (2019).